# Tight Stability Bounds for Robust Distributed Learning:
# Byzantine Failures Hurt Generalization More than Data Poisoning

**Thomas Boudou** [1]   **Batiste Le Bars** [2]   **Nirupam Gupta** [3]   **Aurélien Bellet** [1]

## Abstract

Robust distributed learning algorithms aim to maintain reliable performance despite the presence of misbehaving workers. Such misbehaviors are commonly modeled as *Byzantine failures*, allowing arbitrarily corrupted communication, or as *data poisoning*, a weaker form of corruption restricted to local training data. While prior work shows similar optimization guarantees for both models, an important question remains: *How do these threat models impact generalization?* We show, for the first time, a fundamental gap in generalization guarantees between the two threat models: Byzantine failures yield strictly worse rates than those achievable under data poisoning. Our findings are based upon a tight algorithmic stability analysis of robust distributed learning. Specifically, with $f$ out of $n$ workers misbehaving, we prove that: *(i)* under data poisoning, the uniform algorithmic stability of a robust distributed learning algorithm degrades by an additive factor $\Theta\left(\frac{f}{n-f}\right)$; whereas *(ii)* under Byzantine failures, the degradation factor is $\Omega\left(\sqrt{\frac{f}{n-2f}}\right)$.

## 1. Introduction

With the proliferation of large-scale data and pervasive connectivity, distributed learning has evolved into a fundamental paradigm in modern AI (Kairouz et al., 2021). From federated smartphones to geo-distributed datacenters, algorithms now orchestrate millions of workers toward shared objectives. Yet, unlike the idealized setting of perfectly cooperative workers, real-world systems face a range of adversarial misbehaviors, broadly categorized under the umbrella of Byzantine failures (Guerraoui et al., 2024). Encompassing everything from intermittent data corruption and hardware failure to sophisticated attacks, these failures threaten the reliability of distributed learning algorithms.

Robust distributed learning aims to maintain strong learning guarantees despite the presence of such misbehaviors. The prevailing approach is to model a fraction of workers as corrupted, either Byzantine—able to send arbitrarily corrupted updates—or poisoned, where corruption is limited to local training data. While Farhadkhani et al. (2024b) surprisingly showed that both threat models yield similar empirical risk guarantees, experiments consistently observe significantly worse generalization—i.e., higher risk on unseen data—under Byzantine failures (cf. Allouah et al., 2023a; Karimireddy et al., 2022, and additional discussion in Appendix I). Whether this gap reflects an inherent difference in adversarial power or only the suboptimality of practical attacks remains unknown. We address this open problem with the first theoretical analysis of the generalization gap between Byzantine failures and data poisoning.

To this end, we use algorithmic stability (Bousquet & Elisseeff, 2002) to quantify generalization resilience against both forms of attacks. In particular, uniform stability—which measures an algorithm's sensitivity to the replacement of a single training example—offers a worst-case, distribution-free, and algorithm-centric perspective on generalization. This stands in contrast to approaches based on uniform convergence (Vapnik, 1998), PAC-Bayes (McAllester, 1999), or mutual information (Russo & Zou, 2019), which provide less direct insight into algorithm-specific behavior. This makes stability well-suited for addressing the open question under consideration, as it enables a focused analysis of the potential harm posed by different threat models on the generalization performance of robust learning algorithms.

**Summary of our contributions.** We formalize a unified framework to analyze generalization error of robust distributed learning under different threat models (Section 2). Our results rigorously expose a significant gap in the generalization guarantees between Byzantine failures and data poisoning. Specifically, for convex objective we prove the

[1]PreMeDICaL team, Inria, Idesp, Inserm, Université de Montpellier, Montpellier, France [2]Univ. Lille, Inria, CNRS, Centrale Lille, UMR 9189, CRIStAL, F-59000 Lille, France [3]Department of Computer Science, University of Copenhagen, Copenhagen, Danemark. Correspondence to: Thomas Boudou <thomas.boudou@inria.fr>.

*Proceedings of the 43$^{rd}$ International Conference on Machine Learning*, Seoul, South Korea. PMLR 306, 2026. Copyright 2026 by the author(s).

| Regime | Byzantine failures | Data poisoning |
|---|---|---|
| $f < \frac{n}{3}$ | $\mathcal{O}(\sqrt{\frac{f}{n-f}})$ | $\Theta(\frac{f}{n-f})$ |
| $\frac{n}{3} \leq f < \frac{n}{2+\nu}$ | $\Theta(\sqrt{\frac{f}{n-f}})$ | $\Theta(\frac{f}{n-f})$ |
| $f \sim \frac{n}{2}$ | $\Omega(\sqrt{\frac{f}{n-2f}})$ | $\Theta(\frac{f}{n-f})$ |

*Table 1.* Uniform stability overhead in different threat models for convex loss functions. Here, $\mathcal{O}(\cdot)$ and $\Omega(\cdot)$ denote upper and lower bounds up to absolute constant factors.

following: (1) When $f$ workers among $n$ are Byzantine, the uniform algorithmic stability of robust distributed (S)GD with SMEA—an optimal high-dimensional robust aggregation rule—degrades with an additive factor $\Omega\left(\sqrt{\frac{f}{n-2f}}\right)$ for $f \geq \frac{n}{3}$ (Section 3.2). (2) This factor improves to $\Theta\left(\frac{f}{n-f}\right)$ when we restrict the misbehavior to poisoned data (Section 3.1). Table 1 summarizes our results. Lastly, we show that the above difference yields a fundamental generalization gap between the two threat models (Section 4) by establishing a proportional relationship between stability and generalization (Lemma 4.1). This latter result may also be of independent interest, as it identifies a setting in which stability exactly characterizes generalization, enabling lower bounds on generalization via stability arguments.

## 2. Preliminaries

**Problem setting.** We consider a distributed setup with a central server and $n$ workers, where up to $f$ (with $f < \frac{n}{2}$) may be subject to either *Byzantine failures* or *data poisoning*, as defined below. We refer to such workers as *misbehaving*, unless we need to distinguish between the two attack types. Their identities are unknown to the server. While the actual number of misbehaving workers may be less than $f$, we assume the worst-case scenario where exactly $f$ workers are misbehaving. The remaining *honest* workers form the set $\mathcal{H} \subseteq [n] := \{1, \ldots, n\}$, with $|\mathcal{H}| = n - f$. Each honest worker $i \in \mathcal{H}$ holds a local dataset $\mathcal{D}_i = \{z^{(i,1)}, \ldots, z^{(i,m)}\}$ composed of $m$ i.i.d. data points (or samples) from an input space $\mathcal{Z}$ drawn from a distribution $p_i$. We denote $\mathcal{S} = \cup_{i \in \mathcal{H}} \mathcal{D}_i$. Given a parameter vector $\theta \in \Theta \subset \mathbb{R}^d$ representing the model, a data point $z \in \mathcal{Z}$ incurs a differentiable loss defined by a real-valued function $\ell(\theta; z)$. The goal is to minimize the *population* risk over the honest workers,

$$R_{\mathcal{H}}(\theta) = \frac{1}{|\mathcal{H}|} \sum_{i \in \mathcal{H}} \mathbb{E}_{z \sim p_i}[\ell(\theta, z)].$$

As we only have access to a finite number of samples from each distribution, the above objective can only be solved approximately by minimizing the *empirical* risk over the honest workers, with the added challenge that the honest subgroup is unknown,

$$\widehat{R}_{\mathcal{H}}(\theta) = \frac{1}{|\mathcal{H}|} \sum_{i \in \mathcal{H}} \widehat{R}_i(\theta) = \frac{1}{|\mathcal{H}|} \sum_{i \in \mathcal{H}} \frac{1}{m} \sum_{z \in \mathcal{D}_i} \ell(\theta, z).$$

The expected excess population risk of a distributed learning algorithm's output $\mathcal{A}(\mathcal{S})$ can be decomposed into the generalization and optimization errors (e.g., Hardt et al., 2016, Section 5) as follows

$$\mathbb{E}\left[R_{\mathcal{H}}(\mathcal{A}(\mathcal{S})) - \inf_{\theta \in \Theta} R_{\mathcal{H}}(\theta)\right] \leq \mathbb{E}\left[R_{\mathcal{H}}(\mathcal{A}(\mathcal{S}))\right.$$
$$\left. - \widehat{R}_{\mathcal{H}}(\mathcal{A}(\mathcal{S}))\right] + \mathbb{E}\left[\widehat{R}_{\mathcal{H}}(\mathcal{A}(\mathcal{S})) - \inf_{\theta \in \Theta} \widehat{R}_{\mathcal{H}}(\theta)\right] \quad (1)$$

Prior work on robust distributed learning has focused primarily on the optimization error, i.e., the second term on the right-hand side. In contrast, our goal is to study the generalization error, corresponding to the first term. Formally, we define the notion of *robustness* as follows.

**Definition 2.1.** An algorithm is said to be $(f, \rho, \text{statistical})$-resilient if it outputs a parameter $\hat{\theta}$ such that

$$\mathbb{E}_{\mathcal{A}}\left[R_{\mathcal{H}}(\hat{\theta}) - \inf_{\theta \in \Theta} R_{\mathcal{H}}(\theta)\right] \leq \rho.$$

We say $(f, \rho, \text{empirical})$-resilient when we only consider empirical risk.

Note that the (standard) notion of $(f, \rho, \text{empirical})$-resilience is impossible in general for any $\rho$ when $f \geq n/2$ (Liu et al., 2021), justifying our assumption $f < n/2$.

**Threat models.** We analyze this resilience property under two standard threat models widely studied in the literature (Farhadkhani et al., 2024a; Blanchard et al., 2017).

*Byzantine failures.* In this threat model, misbehaving workers can act arbitrarily: they may deviate from the prescribed algorithm, collude, and send adversarial updates to the server while having access to all information exchanged between honest workers and the server. This model follows the classical notion of Byzantine failures in distributed systems introduced in the seminal work of Lamport et al. (1982).

*Data poisoning.* In this restricted threat model, misbehaving workers follow the prescribed algorithm but may corrupt their local training data *before* execution. Specifically, for each misbehaving worker $i \notin \mathcal{H}$, a local dataset $\mathcal{D}_i \in \mathcal{Z}^m$ is adversarially constructed with full knowledge of the honest workers' data distributions $\{p_i, \ i \in \mathcal{H}\}$.

**Background on robust distributed optimization.** Robust distributed optimization algorithms aim to achieve $(f, \rho, \text{empirical})$-resilience. They are typically adaptations of standard first-order iterative optimization algorithms like gradient descent (GD) or stochastic gradient descent (SGD)

(SGD). These algorithms are made resilient by replacing the server-side averaging operator by a robust aggregation rule $F : (\mathbb{R}^d)^n \to \mathbb{R}^d$ (Guerraoui et al., 2024). Formally, given a learning rate $\gamma$, the parameter $\theta_t$ at iteration $t \in \{0, \ldots, T-1\}$ is updated as follows

$$\theta_{t+1} = G_\gamma^F(\theta_t) := \theta_t - \gamma F(g_t^{(1)}, \ldots, g_t^{(n)}), \quad (2)$$

where $g_t^{(i)}$ denotes the update sent by worker $i$ at iteration $t$. For an honest worker $i \in \mathcal{H}$, we have $g_t^{(i)} = \nabla \widehat{R}_i(\theta_t)$ under GD, or $g_t^{(i)} = \nabla \ell(\theta_t; z_t^{(i)})$ under SGD, with $z_t^{(i)}$ uniformly sampled from the local dataset $\mathcal{D}_i$. For a misbehaving worker $i \notin \mathcal{H}$, $g_t^{(i)}$ is either an arbitrary vector in $\mathbb{R}^d$ (Byzantine failures) or a gradient computed on corrupted data (data poisoning).

Recent advances in robust distributed optimization have identified key properties sufficient for a robust aggregation rule to ensure $(f, \rho, \text{empirical})$-resilience (Allouah et al., 2023a;b). The following definition encompasses a wide range of aggregation rules and has been shown to yield tight resilience guarantees across a variety of practical distributed learning settings. We denote by $\|\cdot\|_2$ the Euclidean norm.

**Definition 2.2.** Let $n \geq 1$, $0 \leq f < n/2$ and $\kappa \geq 0$. An aggregation rule $F$ is said $(f, \kappa)$-robust if for any $g_1, \ldots, g_n \in \mathbb{R}^d$, any set $S \subset [n]$ of size $n - f$, with $\overline{g}_S = \frac{1}{|S|}\sum_{i \in S} g_i$ and $\Sigma_S = \frac{1}{|S|}\sum_{i \in S}(g_i - \overline{g}_S)(g_i - \overline{g}_S)^\mathsf{T}$, we have

$$\|F(g_1, \ldots, g_n) - \overline{g}_S\|_2^2 \leq \kappa \|\Sigma_S\|_{\text{sp}}.$$

Here, $\|\cdot\|_{\text{sp}}$ denotes the spectral norm (i.e., the largest eigenvalue), $f$ and $\kappa$ are referred to as the *robustness parameter* and *robustness coefficient* of $F$, respectively.

Any aggregation rule $F$ that is $(f, \mathcal{O}(f/n))$-robust achieves optimal $(f, \rho, \text{empirical})$-resilience. Specifically, iterative methods employing such a rule attain an optimization error that matches the information-theoretic lower bound for first-order optimization in the presence of $f$ misbehaving workers (Farhadkhani et al., 2024b). One such optimal rule is SMEA, introduced by Allouah et al. (2023b).

**Definition 2.3.** Given $f, n \in \mathbb{N}$, $f < n/2$ and vectors $g_1, \ldots, g_n \in \mathbb{R}^d$, the smallest maximum eigenvalue averaging aggregation rule (SMEA) outputs the average of the $n - f$ gradients that are most directionally consistent, discarding outliers based on covariance,

$$\overline{g}_{S^*} = \frac{1}{|S^*|}\sum_{i \in S^*} g_i, \quad \text{with} \quad S^* \in \operatorname*{argmin}_{\substack{S \subseteq \{1, \ldots, n\}, \\ |S| = n-f}} \|\Sigma_S\|_{\text{sp}}.$$

However, the above results focus on the optimization error, overlooking the impact of robust aggregation rules on the generalization error—the first term in the right-hand side

of (1). To address this gap, we provide a rigorous analysis of the generalization guarantees of robust distributed learning, focusing on the SMEA aggregation rule as it yields optimal optimization guarantees.

**Stability under misbehaving workers.** We leverage the algorithmic stability framework to bound the generalization error of robust distributed learning algorithms. We do so via uniform stability, which measures how sensitive a learning algorithm is to changes in its training data. This approach was introduced by Vapnik & Chervonenkis (1974); Rogers & Wagner (1978); Devroye & Wagner (1979) and later popularized by Bousquet & Elisseeff (2002); Shalev-Shwartz et al. (2010); Hardt et al. (2016). We revisit this framework to study generalization in robust distributed learning, where the change is only in the honest workers' data.

**Definition 2.4.** Consider $n$ workers, with $f$ misbehaving and $n - f$ honest, each holding $m$ local samples. A distributed algorithm $\mathcal{A}$ is said $\varepsilon$-uniformly stable if, for all honest datasets $\mathcal{S}, \mathcal{S}' \in \mathcal{Z}^{(n-f)m}$ that are *neighboring*—i.e., differing in at most one sample—we have

$$\sup_{z \in \mathcal{Z}} \mathbb{E}[\ell(\mathcal{A}(\mathcal{S}); z) - \ell(\mathcal{A}(\mathcal{S}'); z)] \leq \varepsilon,$$

where the expectation is over the randomness of $\mathcal{A}$.

This property leads to the following bound on the generalization error, proved in Appendix A.

**Proposition 2.5.** *If $\mathcal{A}$ is $\varepsilon$-uniformly stable, then*

$$|\mathbb{E}_{\mathcal{S},\mathcal{A}}[R_\mathcal{H}(\mathcal{A}(\mathcal{S})) - \widehat{R}_\mathcal{H}(\mathcal{A}(\mathcal{S}))]| \leq \varepsilon.$$

The link between uniform stability and generalization can be further strengthened to yield high-probability generalization bounds, as discussed in Appendix H. We can now formalize a relationship between empirical resilience and statistical resilience through stability: any algorithm that is $(f, \rho, \text{empirical})$-resilient and $\varepsilon$-uniformly stable is $(f, \rho + \varepsilon, \text{statistical})$-resilient. Here, $\varepsilon$ quantifies the effect of misbehaving workers on generalization, and is the central focus of this paper.

## 3. (In)stability of Robust Distributed Learning

In this section, we first derive upper bounds on the stability of robust distributed (S)GD under general $(f, \kappa)$-robust aggregation rules. We next analyze SMEA to derive lower bounds under Byzantine failures, as well as tight upper and lower bounds for the special case of data poisoning. Our results show a fundamental gap in stability between the two threat models. Definitions of function regularity are given in Appendix B.

### 3.1. The Case of Byzantine Failures

**Upper bounds.** We start by deriving uniform stability upper bounds for convex optimization. Proofs are deferred to Appendix C.2.

**Theorem 3.1.** *Consider the setting described in Section 2 under Byzantine failures. Let $\mathcal{A} \in \{\mathrm{GD}, \mathrm{SGD}\}$ with a $(f, \kappa)$-robust aggregation rule $F$. Suppose $\forall z \in \mathcal{Z}, \ell(\cdot; z)$ $C$-Lipschitz and $L$-smooth, and $\mathcal{A}$ is run for $T \in \mathbb{N}^*$ iterations with $\gamma \leq \frac{1}{L}$.*

*(i) If $\ell(\cdot; z)$ is convex $\forall z \in \mathcal{Z}$, then the uniform stability of $\mathcal{A}$ is upper bounded by*

$$2\gamma C^2 T \left( \sqrt{\kappa} + \frac{1}{(n-f)m} \right). \tag{3}$$

*(ii) If $\ell(\cdot; z)$ is $\mu$-strongly convex $\forall z \in \mathcal{Z}$, then the uniform stability of $\mathcal{A}$ is upper bounded by*

$$\frac{2C^2}{\mu} \left( \sqrt{\kappa} + \frac{1}{(n-f)m} \right). \tag{4}$$

*Proof sketch.* Denote $\{\theta_t\}_{t \in [T]}$ and $\{\theta'_t\}_{t \in [T]}$ the coupled optimization trajectories resulting from two neighboring datasets $\mathcal{S}, \mathcal{S}'$. In the context of stability analysis via parameter sensitivity, we track how the parameters diverge along these optimization trajectories. To do so, we decompose the analysis by introducing an intermediate comparison between the robust update $G_\gamma^F$ and the averaging over honest workers update $G_\gamma^{\mathcal{A}}$. This enables us to leverage either the regularity of the loss function or the robustness property of the aggregation rule. This approach is motivated by the fact that Byzantine vectors cannot be assumed to exhibit regularities when comparing them. By adding and subtracting $G_\gamma^{\mathcal{A}}(\theta_t)$ and $G_\gamma^{\mathcal{A}'}(\theta'_t)$, the triangle inequality yields

$$\mathbb{E}_{\mathcal{A}} \| G_\gamma^F(\theta_t) - G_\gamma^{F'}(\theta'_t) \|_2 \leq A + B + B',$$

where $A = \mathbb{E}_{\mathcal{A}} \| G_\gamma^{\mathcal{A}}(\theta_t) - G_\gamma^{\mathcal{A}'}(\theta'_t) \|_2$, $B = \mathbb{E} \| G_\gamma^F(\theta_t) - G_\gamma^{\mathcal{A}}(\theta_t) \|_2$, and $B' = \mathbb{E} \| G_\gamma^{F'}(\theta'_t) - G_\gamma^{\mathcal{A}'}(\theta'_t) \|_2$. We then bound $A$ using standard stability arguments (Hardt et al., 2016), and $B, B'$ with the $(f, \kappa)$-robust property, the Jensen's inequality and boundedness of honest gradients,

$$B + B' \leq 2\gamma \mathbb{E}_{\mathcal{A}} \sqrt{\kappa \| \Sigma_{\mathcal{H}, t} \|_{\mathrm{sp}}} \leq 2\gamma \sqrt{\kappa \mathbb{E}_{\mathcal{A}} \| \Sigma_{\mathcal{H}, t} \|_{\mathrm{sp}}}$$
$$\leq 2\gamma \sqrt{\kappa} C. \quad \square$$

When there are no Byzantine workers ($f = 0$, hence $\kappa = 0$), both (3) and (4) reduce to the classical stability bounds established in Hardt et al. (2016), known to be tight (Zhang et al., 2022). However, when $f > 0$, the analysis reveals a degradation by an additive term of order $\mathcal{O}(\sqrt{\kappa})$. In the convex setting, this degradation accumulates over the $T$ iterations, whereas in the strongly convex case, it appears as a one-time term independent of $T$.

**Lower bounds.** Since $\kappa \geq \frac{f}{n-2f}$ (Allouah et al., 2023a, Proposition 6), the additive term in Theorem 3.1 is lower bounded by $\sqrt{\frac{f}{n-2f}}$. We show that this bound is tight by deriving the following lower bound, proved in Appendix C.3.

**Theorem 3.2.** *Consider the setting in Section 2 under Byzantine failures and assume $\frac{n}{3} \leq f < \frac{n}{2}$. Let $\mathcal{A} \in \{\mathrm{GD}, \mathrm{SGD}\}$, with SMEA. Suppose $\mathcal{A}$ is run for $T \in \mathbb{N}^*$ iterations ($T \in \Omega(m)$ for SGD) with learning rate $\gamma$. Then there exist $\ell \in \mathbb{R}^{\Theta \times \mathcal{Z}}$ such that $\forall z \in \mathcal{Z}, \ell(\cdot; z)$ is $C$-Lipschitz, $L$-smooth and convex, and neighboring datasets such that the uniform stability of $\mathcal{A}$ is lower bounded by*

$$\Omega \left( \gamma C^2 T \left( \sqrt{\frac{f}{n-2f}} + 1 + \frac{1}{(n-f)m} \right) \right). \tag{5}$$

*Proof sketch.* Let $C, L \in \mathbb{R}_+^*$ and let $\ell(\theta; z) = z\theta, \theta \in \mathbb{R}$, $z \in [-C, C]$, which is $C$-Lipschitz, $L$-smooth and convex. In this particular case, the SMEA rule amounts to picking the subset of size $n - f$ with smallest variance. Our proof relies on three key elements.

*(i)* We maximize the variance among honest gradients by dividing them into two equal-sized subgroups positioned at the boundary of the Lipschitz constraint. More precisely, $\frac{n-f}{2}$ honest workers have datasets with $m$ identical samples equal to $C$, the remaining $\frac{n-f}{2}$ have samples equal to $-C$. Hence, this set of $n - f$ values has, at each iteration, a variance equal to $C^2$.

*(ii)* When $f \geq \frac{n}{3}$, Byzantine workers can craft updates that cause SMEA to entirely discard one honest subgroup while outweighing the influence of the other. Specifically, if Byzantine workers send $\beta = C \left( 1 + \frac{1}{2} \frac{n-f}{\sqrt{f(n-2f)}} \right)$ at each iteration, then the subgroup consisting of $f \geq \frac{n-f}{2}$ Byzantine workers together with $n - 2f \leq \frac{n-f}{2}$ honest workers sending $C$ forms the subset with the lowest variance, equal to $C^2/4 < C^2$. Consequently, under this configuration, the parameter at step $T$ becomes $\theta_T = -\gamma C T \left( 1 + \frac{1}{2} \sqrt{\frac{f}{n-2f}} \right)$.

*(iii)* Finally, we exploit the fact that Byzantine workers can observe all communications and adapt their behavior accordingly. Specifically, we can define an event that triggers them to switch their communicated value $\beta$ to an arbitrarily large number. For instance, if a single sample from a honest worker is changed from $C$ to $0$—defining a neighboring dataset—the server receives a different average $\frac{m-1}{m} C < C$ that triggers the Byzantine workers. The choice of an arbitrary large value is a matter of convenience, ensuring that the perturbed run diverges in the positive direction at a rate of $\theta'_T = \frac{\gamma C T}{(n-f)m}$. As, in this setting, stability is proportional to $|\theta_T - \theta'_T|$, this concludes the proof sketch. $\square$

Note that, for SMEA, $\sqrt{\kappa} \in \mathcal{O}\left(\sqrt{\frac{f}{n-f}}\left(1 + \frac{f}{n-2f}\right)\right)$ (Al-louah et al., 2023b, Proposition 5.1). Plugging this in our upper-bound and considering the regime $n/3 \leq f \leq n/(2+\nu)$ for some constant $\nu > 0$, our upper and lower bounds become tight at a rate of $\Theta\left(\sqrt{\frac{f}{n-f}}\right)$, providing a precise characterization of stability under Byzantine failures. For $f < n/3$, we lack a strong lower bound; nonetheless, we investigate this regime numerically in Section 4 (and discuss it in Appendix I), yielding valuable insights into the transition from few to many misbehaving workers.

Crucially, the lower bound above cannot be replicated under data poisoning, due to points *(ii)* and *(iii)* in the proof sketch. Unlike Byzantine workers, who can send arbitrary and adaptive updates, poisoned workers are constrained by the loss function's regularity and by being committed to a fixed dataset before the algorithm begins. The next section formally proves a stability gap between the two threat models.

### 3.2. The Case of Data Poisoning

Unlike our previous bounds, which account for the arbitrariness of Byzantine vectors, we now leverage the regularity of poisoned gradients in data poisoning to derive tighter stability bounds for robust distributed learning with SMEA. The proof sketches below further elucidate the key distinctions between the threat models. Formal proofs are in Appendices D.1 to D.3.

**Upper bounds.** We start by proving a tighter upper bound under data poisoning.

**Theorem 3.3.** *Consider the setting described in Section 2 under data poisoning. Let $\mathcal{A} \in \{\mathrm{GD}, \mathrm{SGD}\}$, with* SMEA. *Suppose $\ell(\cdot; z)$ C-Lipschitz and L-smooth $\forall z \in \mathcal{Z}$, and $\mathcal{A}$ is run for $T \in \mathbb{N}^*$ iterations with $\gamma \leq \frac{1}{L}$.*

*(i) If $\ell(\cdot; z)$ is convex $\forall z \in \mathcal{Z}$, then the uniform stability of $\mathcal{A}$ is upper bounded by*

$$2\gamma C^2 T \left(\frac{f}{n-f} + \frac{1}{(n-f)m}\right). \quad (6)$$

*(ii) If $\ell(\cdot; z)$ is $\mu$-strongly convex $\forall z \in \mathcal{Z}$, then the uniform stability of $\mathcal{A}$ is upper bounded by*

$$\frac{2C^2}{\mu}\left(\frac{f}{n-2f} + \frac{1}{(n-2f)m}\right). \quad (7)$$

*Proof sketch.* We consider two runs of (2), with SMEA and GD, on neighboring datasets, initialized with $\theta_0 = \theta'_0$. Denote $\delta_t = \|\theta_t - \theta'_t\|_2$. At each step $t$, SMEA selects subsets of workers $S_t^*$ and $S_t'^*$ of size $n-f$. The key insight is to decompose the update rule based on the intersection $I_t = S_t^* \cap S_t'^*$, where $n - 2f \leq |I_t| \leq n - f$, and to compare the updates as a gradient step on the intersection $I_t$, $G_{I_t}(\theta_t) = \theta_t - \frac{\gamma}{n-f}\sum_{i \in I_t} \nabla \widehat{R}_i(\theta_t)$, plus a perturbation

term from the symmetric difference $\Delta_t = (S_t^* \setminus S_t'^*) \cup (S_t'^* \setminus S_t^*)$, $|\Delta_t| \leq 2f$. The proof then relies on two key elements, where we bound the update (2) difference by

$$
\begin{aligned}
\delta_{t+1} &\leq \|G_{I_t}(\theta_t) - G_{I_t}(\theta'_t)\|_2 \\
&+ \frac{\gamma}{n-f}\|\sum_{i \in S_t^* \setminus S_t'^*} \nabla \widehat{R}_i(\theta_t) - \sum_{i \in S_t'^* \setminus S_t^*} \nabla \widehat{R}_i'(\theta_t)\|_2 \\
&+ \frac{\gamma}{(n-f)m}\|\nabla \ell(\theta_t, z^{(a,b)}) - \nabla \ell(\theta'_t, z_t'^{(a,b)})\|_2, \quad (8)
\end{aligned}
$$

where $a, b \in \mathcal{H} \times \{1, \ldots, m\}$ index the differing samples.

*(i)* Since $I_t$ represents a fraction $\frac{n-2f}{n-f} \leq \rho_t := \frac{|I_t|}{n-f} \leq 1$ of the gradient update, its expansivity (cf. Lemma B.4) $\eta_{I_t}$ relates to the full update expansivity $\eta$ via

$$\eta_{I_t} \leq 1 + \rho_t(\eta - 1).$$

If $\ell$ is convex ($\eta = 1$), then $\eta_{I_t} \leq 1$. If $\ell$ is $\mu$-strongly convex ($\eta = 1 - \gamma\mu$), then $\eta_{I_t} \leq 1 - \gamma\mu\frac{n-2f}{n-f}$.

*(ii)* We bound the divergence terms in (8)—arising from the non-shared indices $\Delta_t$ and the differing samples—using the Lipschitz continuity of $\ell$, $\|\sum_{i \in S_t^* \setminus S_t'^*} \nabla \widehat{R}_i(\theta_t) - \sum_{i \in S_t'^* \setminus S_t^*} \nabla \widehat{R}_i'(\theta_t)\|_2 \leq 2fC$, and $\|\nabla \ell(\theta_t, z^{(a,b)}) - \nabla \ell(\theta'_t, z_t'^{(a,b)})\|_2 \leq 2C$. Combining (8), *(i)* and *(ii)* yields the following recursion

$$\delta_{t+1} \leq \eta_{I_t}\delta_t + 2\gamma C\left(\frac{f}{n-f} + \frac{1}{m(n-f)}\right). \quad (9)$$

We conclude the proof by solving the recursion and invoking the Lipschitz continuity of $\ell$. $\square$

Crucially, the upper bounds derived above cannot be replicated for Byzantine failures, due to points *(i)* and *(ii)* in the proof sketch. The fundamental distinction is that poisoned workers—unlike Byzantine workers—remain constrained by the loss function's regularity. Furthermore, notice that the proof technique generalizes beyond SMEA, yielding the same upper bounds for any rule that averages $n - f$ gradients, independent of the selection mechanism.

**Lower bounds.** Before comparing the above bounds to the Byzantine case, we establish their tightness by deriving matching lower bounds. These lower bounds are established for projected–SGD, which applies a Euclidean projection onto the positive half parameter space after each iteration. While this simplifies the lower bounds proofs (see Appendix D.3), it does not affect the tightness of our result since the upper bounds on uniform stability remain valid for projected–SGD as the projection does not increase the distance between projected points.

**Theorem 3.4.** *Consider the setting of Section 2 under data poisoning. Let $\mathcal{A} \in \{\mathrm{GD}, \mathrm{projected}\text{–}\mathrm{SGD}\}$, with* SMEA, *run for $T \in \mathbb{N}^*$ iterations with $\gamma \leq \frac{1}{L}$.*

(i) *There exists $\ell$ such that $\forall z \in \mathcal{Z}, \ell(\cdot; z)$ is C-Lipschitz, L-smooth and convex, and neighboring datasets such that the uniform stability of $\mathcal{A}$ is lower bounded by*

$$\Omega\left(\gamma C^2 T\left(\frac{f}{n-f} + \frac{1}{(n-f)m}\right)\right). \quad (10)$$

*For* projected–SGD, *the above result assumes the existence of a constant $\tau \geq c > 0$ (for an arbitrary constant c) such that $T \geq \tau m$.*

(ii) *For $\mathcal{A} = $ GD, if $T \geq \frac{\ln(1-c)}{\ln(1-\gamma\mu)}$ with $0 < c < 1$, there exist $\ell$ such that $\forall z \in \mathcal{Z}, \ell(\cdot; z)$ is additionally $\mu$-strongly convex, and neighboring datasets such that the uniform stability of $\mathcal{A}$ is lower bounded by*

$$\Omega\left(\frac{C^2}{\mu}\left(\frac{f}{n-f} + \frac{1}{(n-f)m}\right)\right). \quad (11)$$

*Proof sketch.* For formal proofs, see Appendices D.2 and D.3. We specifically sketch below our construction for $\mathcal{A} = $ GD, which will be used extensively in Section 4. In dimension one, the SMEA rule amounts to picking the subset with the smallest gradient variance. For $n = 3$ and $f = 1$, this reduces to minimizing in $i, j, \|g_i - g_j\|_2^2/4$, so SMEA averages the closest pair of worker gradients. Let $\ell(\theta; z) = z\theta, \theta \in \mathbb{R}, z \in [-C, C]$, and set local datasets to be homogeneous, equal to 0 (worker 1), $-C$ (worker 2), and $C - \delta$ (worker 3, $\delta > 0$). Then $\nabla\widehat{R}_1(\theta) = 0$, $\nabla\widehat{R}_2(\theta) = -C, \nabla\widehat{R}_3(\theta) = C - \delta$, so SMEA favors $\{1, 3\}$, updating the parameter in the negative direction. Since $C - \delta$ can be made arbitrarily close to $C$, a perturbation of worker 1's dataset (e.g., changing a sample from 0 to $-C$, giving $\nabla\widehat{R}_1'(\theta) = -\frac{C}{m}$) makes SMEA favor $\{1, 2\}$, updating the parameter in the positive direction. Hence, parameter divergence is proportional to $\gamma CT$. This reasoning generalizes to any $f$ and $n$ (see Appendix D.2.1), replacing a single sample can cause SMEA to swap $f$ workers among the $n - f$ originally subsampled, yielding stability proportional to $\gamma C^2 T\left(\frac{f}{n-f} + \frac{1}{(n-f)m}\right)$, where the first term reflects worker swapping and the second the perturbed sample. □

This result establishes an unavoidable additional instability relative to the simple averaging rule. Interestingly, if SMEA were co-coercive (cf. Equation 15 in Appendix B), these lower bounds would not hold: one could then directly apply the techniques of Hardt et al. (2016) to obtain a tighter upper bound, effectively removing the second term in (8). This suggests that the loss of co-coercivity is the primary driver of the instability observed in (10)-(11). To our knowledge, no robust aggregation rule preserves this inequality. Identifying such a rule—or proving that none exists—remains an important open question for future work to further improve stability guarantees under data poisoning.

**Fundamental gap with Byzantine bounds.** We now compare the bounds obtained for the two threat models. Crucially, the additional instability term in the Byzantine upper bounds (3) and (4) decreases from $\mathcal{O}(\sqrt{\kappa})$—which for SMEA evaluates to $\mathcal{O}(\sqrt{\frac{f}{n-f}}(1 + \frac{f}{n-2f}))$ (Allouah et al., 2023b, Proposition 5.1)—down to $\mathcal{O}(\frac{f}{n-f})$ in the data poisoning case (6) and (7). The upper bound (6) is thus an order of magnitude smaller than the lower bound $\Omega(\sqrt{\frac{f}{n-2f}})$ (5) obtained for $f \geq \frac{n}{3}$, establishing a fundamental stability gap. As $f$ approaches its maximal value $\frac{n}{2}$, the gap widens significantly, highlighting the stability contrast between the two threat models. Table 1 summarizes our results.

We present further results for smooth *nonconvex* learning in Appendix E, along with a discussion on relaxing other regularity assumptions in Appendix I. Taken together, these findings support the view that the stability gap between Byzantine failures and data poisoning is not limited to our baseline assumptions.

In the next section, we show that this stability gap drives a corresponding gap in generalization error, revealing a fundamental difference in the ability of the two threat models to degrade performance on unseen data.

## 4. Generalization Gap

While we proved tight stability bounds for robust distributed learning in Section 3, it remains unclear how the uncovered stability gap translates into generalization error under the two threat models. To address this, we focus on the smooth convex setting with SMEA and construct a data distribution where the generalization error is not merely upper bounded by uniform stability but is in fact *proportional* to it, independently of the threat model.

**Lemma 4.1.** *Consider the setting described in Section 2, with $m = 1$, regardless of the assumed threat model. There exist $\ell \in \mathbb{R}^{\Theta \times \mathcal{Z}}$ such that $\forall z \in \mathcal{Z}, \ell(\cdot; z)$ is C-Lipschitz, L-smooth and convex, data distributions $\{p_i\}_{i \in \mathcal{H}}$ over $\mathcal{Z}$, such that for any distributed algorithm $\mathcal{A}$, we have*

$$\left|\mathbb{E}_{\mathcal{A}, \mathcal{S} \sim \otimes_{i \in \mathcal{H}}\left(p_i^{\otimes m}\right)}\left[R_{\mathcal{H}}(\mathcal{A}(\mathcal{S})) - \widehat{R}_{\mathcal{H}}(\mathcal{A}(\mathcal{S}))\right]\right|$$
$$= \frac{1}{4(n-f)} \sup_{z \in \mathcal{Z}} \mathbb{E}_{\mathcal{A}}\left[\ell(\mathcal{A}(S); z) - \ell(\mathcal{A}(S'); z)\right], \quad (12)$$

*where the dependence on the threat model arises solely through the stability term.*

Lemma 4.1 establishes a direct equality between the generalization gap and algorithmic stability, confirming that the threat model's impact is captured by the latter—a result that may be of independent interest for deriving lower bounds on generalization using stability tools. Notably, This construction unifies the lower-bound settings (i.e., honest

datasets and loss function) shared by both worst-case data poisoning and Byzantine attack scenarios. Therefore, our findings reveal a fundamental separation: there exist data distributions and Byzantine attacks whose generalization error is strictly larger (in terms of convergence rate) than that of the worst-case data poisoning attack, as demonstrated by the following theorem, proved in Appendix F.1.

**Theorem 4.2.** *Consider the setting of Lemma 4.1, where $\frac{n}{3} \leq f < \frac{n}{2}$, $\mathcal{A} \in \{\text{GD}, \text{SGD}\}$ with SMEA. Then, there exist $\ell \in \mathbb{R}^{\Theta \times \mathcal{Z}}$ such that $\forall z \in \mathcal{Z}, \ell(\cdot; z)$ is C-Lipschitz, L-smooth and convex, data distributions $\{p_i\}_{i \in \mathcal{H}}$ over $\mathcal{Z}$, and a Byzantine attack such that, for any data poisoning attack, we have*

$$\frac{\mathcal{E}_{gen}^{\text{byz}}}{\mathcal{E}_{gen}^{\text{pois}}} \in \Omega\left(\frac{n-f}{\sqrt{f(n-2f)}}\right), \qquad (13)$$

*where $\mathcal{E}_{gen}^{\text{byz}}$ and $\mathcal{E}_{gen}^{\text{pois}}$ denote generalization errors under the Byzantine and data poisoning attacks, respectively.*

Theorem 4.2 quantifies the separation between the threat models, proving that the generalization error under Byzantine attacks asymptotically dominates that of data poisoning. Specifically, the factor $\frac{n-f}{\sqrt{f(n-2f)}}$ grows from a minimum of 2, when $f = \frac{n}{3}$, to exceed $\frac{\sqrt{n}}{2}$ as $f$ approaches the breakdown point ($f = \frac{n}{2} - 1$, or $\frac{n-1}{2}$). Below, we complement our theoretical results with numerical experiments that validate the analysis and provide further insights. Then, we discuss the broader intuition underlying the emergence of this generalization gap between the two threat models.

**Numerical validation.** We numerically instantiate the construction used in proving the stability lower bound under data poisoning, which coincides with the Byzantine case when $f \geq \frac{n}{3}$ (see Appendices C.3 and D.2.1 for further details), with $m = 1$. Under this configuration, we can use Lemma 4.1 as demonstrated in Theorem 4.2. For our experiments, we fix the learning rate and the Lipschitz coefficient $\gamma = C = 1$, the number of epochs $T = 5$, the number of total workers $n = 15$ and vary the number of misbehaving workers $f$ from 1 to 7. As expected, we observe that under data poisoning, our theoretical upper bound (scaled by the factor $\frac{1}{4(n-f)}$ from Equation 12) closely matches the empirically measured generalization error (Figure 1), confirming the practical tightness of our analysis in this setting, including the constant factors.

Next, we evaluate the system's performance under the Byzantine attack described in the proof of the stability lower bound in Theorem 3.2. In particular, for each $f$, we fix $\mathcal{H}$ to maximize honest gradients' variance. The Byzantine workers then accelerate the divergence of the parameters, either in the positive or negative direction, triggered by a specific event that depends on the observed communication. Numerically, this is achieved by adaptively crafting updates

that remain within the aggregation rule's selection range (see Appendix F.2 for details). We observe that this Byzantine attack results in a significantly higher generalization error that grows with the number of Byzantine workers. Importantly, when $f \geq \frac{n}{3}$, the attack produces an error greater than the lower bound reported in Theorem 3.2. This was expected as our data distribution produces datasets matching the worst-case one crafted for the Byzantine lower bound. The gap between the two curves illustrates that our attack optimizes the numerical constant in the lower bound.

Interestingly, even when $f < \frac{n}{3}$, Byzantine failures continue to induce substantially greater generalization error than worst-case data poisoning, emphasizing that the gap persists even with a moderate fraction of misbehaving workers. In fact, as $f < \frac{n}{3}$ decreases, it becomes increasingly challenging (but not impossible) to simultaneously maintain maximal honest gradients' variance and amplify the influence of Byzantine workers. In the absence of an order-of-magnitude larger lower bound under Byzantine failures in the regime $f < n/3$, it is natural to ask whether the generalization gap also extends to this low-to-moderate fraction of misbehaving workers. While our numerical experiments support this intuition, the open question remains whether the gap is merely a constant factor or truly an order of magnitude.

In summary, our numerical experiment demonstrates that Byzantine failures can have a significantly greater generalization error than worst-case data poisoning across all regimes, from low to high fractions of misbehaving workers. Remarkably, this also confirms the fundamental generalization gap between Byzantine and data poisoning threat models, as highlighted above in Theorem 4.2.

**Why threat model differences affect generalization, but not optimization?** In contrast to Farhadkhani et al. (2024b), who show similar optimization error guarantees for both threat models, our work reveals a gap in generalization error. The key intuition is as follows. Optimization analysis focuses on how accurately the descent direction is estimated at each step, whereas generalization analysis examines the algorithm's sensitivity to individual training samples, independent of how informative the estimated direction is for optimization. In the worst case, both corruptions similarly impair optimization direction estimation. However, as discussed in Section 3, stability analysis benefits from the additional regularity and non-adaptive nature of data poisoning.

Specifically, optimization error analysis relies on the smoothness of the empirical loss over honest workers (e.g., Theorem 1 in Allouah et al., 2023a) to control its decrease along the optimization trajectory, typically via terms like $\widehat{R}_{\mathcal{H}}(\theta_{t+1}) - \widehat{R}_{\mathcal{H}}(\theta_t)$ or $\|\nabla \widehat{R}_{\mathcal{H}}(\theta_t)\|_2^2$. This smoothness-based control applies equally to both threat models, as the regularity of corrupted gradients is not invoked. In contrast, stability analysis aims to bound the difference

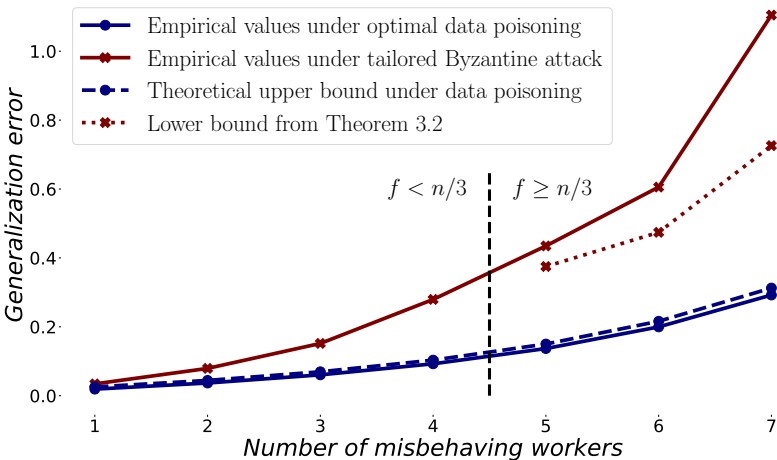

*Figure 1.* Generalization Error Under Optimal Poisoning And Tailored Byzantine Attacks.

$\ell(\theta_t, z) - \ell(\theta'_t, z)$ between the losses along two optimization trajectories $\theta_t$ and $\theta'_t$ produced by neighboring datasets. Here, regularities can be exploited under data poisoning—where corrupted updates remain gradients of a smooth loss function (cf. proof of Theorem 3.3)—but not under Byzantine failures, which require an intermediate comparison step (cf. proof of Theorem 3.1).

Furthermore, prior work (Allouah et al., 2023a) shows that for the optimization error, $(f, \kappa)$-robustness with $\kappa \in \mathcal{O}(f/n)$ suffices to achieve an upper bound under Byzantine failures that matches the lower bound for poisoned data (Allouah et al., 2023a, Theorem 1 & Proposition 1). Hence, this bound cannot be improved using the regularity of corrupted gradients.

## 5. Related Work

Minimizing empirical risk over honest workers in the presence of Byzantine workers has been extensively studied (Guerraoui et al., 2018; Karimireddy et al., 2021; 2022; Allouah et al., 2023a; Gorbunov et al., 2023; Allouah et al., 2023b), leading to robust aggregation schemes enabling distributed SGD to attain optimal error even under data heterogeneity. A key contribution of this line of work is the formal analysis of aggregation rules previously studied under i.i.d. assumptions. However, these analyses focus on optimization error and offer limited insight into the generalization performance.

Empirical studies (Baruch et al., 2019; Xie et al., 2020; Shejwalkar & Houmansadr, 2021; Allen-Zhu et al., 2021; Allouah et al., 2023a) have shown that the generalization error under Byzantine attacks is often worse than under data poisoning, but offer no formal explanation for this gap. These findings stand in contrast to theoretical analyses

proving comparable statistical error rates across the two threat models (Alistarh et al., 2018; Zhu et al., 2023; Yin et al., 2018). However, these theoretical results assume an i.i.d. data setting, where all honest workers sample from the same distribution, and do not account for data heterogeneity common in distributed learning. We bridge this gap between empirical observations and theoretical results by developing the first theoretical framework to analyze the generalization error of robust first-order iterative methods under heterogeneous data and the two considered threat models. Our findings apply to a much larger class of loss functions, and indeed show that Byzantine attacks can be more harmful to generalization than data poisoning.

A prior work from Farhadkhani et al. (2022b), which claims an equivalence between the two threat models in the context of PAC learning, considers only the classic non-robust distributed SGD algorithm. Their analysis does not apply to robust distributed GD or SGD, as classic averaging has no optimization guarantees in presence of misbehaving workers. Another work from Farhadkhani et al. (2024b), which analyzes the relationship between data poisoning and Byzantine failures for robust distributed SGD, considers a streaming setting in which honest workers acquire i.i.d. training samples from their respective data-generating distributions at each iteration. While they show that, in that setting, the two threat models yield comparable population risk (asymptotically), their results and proof techniques do not apply to our more pragmatic setting wherein the training datasets across the honest workers are sampled *a priori*.

Ye & Ling (2025) study uniform stability of robust decentralized SGD under Byzantine failures. However, their analysis uses a diameter-based robustness definition for aggregation rules. This definition leads to robustness parameters that exhibit suboptimal dependence on the problem dimen-

sion or the number of honest agents, as highlighted in Ye et al. (2024, Table II) and further discussed in Allouah et al. (2023a, Section 8.3). Consequently, this robustness notion yields suboptimal empirical resilience (cf. Farhadkhani et al., 2022a; Allouah et al., 2023a), in contrast to covariance-based definitions, such as $(f, \kappa)$-robustness, that we consider. Moreover, Ye & Ling (2025) does not provide any insights into the data poisoning threat model.

## 6. Conclusion

We have rigorously proved, for the first time, a fundamental gap in the generalizability of robust distributed learning under two central threat models: Byzantine failures and data poisoning. Specifically, we have shown that Byzantine failures can cause strictly worse generalization than data poisoning—even when both attacks are optimally designed. Our results explain prior empirical observations overlooked by optimization analyses, revealing that guarantees differ across the two threat models. This insight opens new avenues to better understand robust distributed learning and adapt robustness properties to generalization. In particular, incorporating stability into the design of robust aggregation rules could help achieve not only strong optimization guarantees but also lower population risk.

**Practical implications.** Our results highlight the relevance of cryptographic tools—particularly *zero-knowledge proofs* (ZKPs) (Goldwasser et al., 1989; Thaler, 2022)—for distributed learning. ZKPs can prevent Byzantine failures by verifying that each worker's input data lies within an appropriate range and that their model updates remain consistent with their initially committed dataset, all without revealing the data itself (Sabater et al., 2022; Abbaszadeh et al., 2024; Shamsabadi et al., 2024). However, they do not guard against data poisoning, which stems from malicious or corrupted local data. Therefore, cryptographic safeguards must be paired with robust learning algorithms that can tolerate compromised training data to ensure end-to-end resilience.

**Future work.** Promising directions include: *(i)* Find a robust aggregation rule preserving the loss function's co-coercivity, or prove this is impossible (cf. discussion after Theorem 3.4); *(ii)* Go beyond worst-case analysis via data-dependent refinements, such as on-average stability (Lei & Ying, 2020; Sun et al., 2024; Ye et al., 2025); *(iii)* Extend results to the *locally-poisonous* setting (Farhadkhani et al., 2024b); *(iv)* Investigate whether the trilemma between robustness, privacy and optimization error (Allouah et al., 2023b) also holds for generalization error; *(v)* Quantify the stability of other first-order algorithms, including momentum and communication-efficient methods (cf. Appendix I).

## Impact Statement

This paper presents work whose goal is to advance the field of machine learning. There are many potential societal consequences of our work, none of which we feel must be specifically highlighted here.

## Acknowledgements

The work of Thomas Boudou and Aurélien Bellet is supported by grant ANR 22-PECY-0002 IPOP (Interdisciplinary Project on Privacy) project of the Cybersecurity PEPR.

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

## Organization of the appendix

The supplementary materials are organized as follows.

## A. Stability Implies Generalization Under Byzantine Failures or Data Poisoning

In this section, we focus on the formal link between uniform stability and generalization under Byzantine failures and data poisoning. The proof follows classic arguments from Bousquet & Elisseeff (2002).

**Proposition 2.5.** Consider the setting described in Section 2, under either Byzantine failures or data poisoning. let $\mathcal{A}$ an $\varepsilon$-*uniformly stable* (randomized) distributed algorithm. Then,

$$|\mathbb{E}_{\mathcal{S},\mathcal{A}}[R_{\mathcal{H}}(\mathcal{A}(\mathcal{S})) - \widehat{R}_{\mathcal{H}}(\mathcal{A}(\mathcal{S}))]| \leq \varepsilon.$$

*Proof.* Recall that $\mathcal{S}$ denotes the collective dataset of honest workers. Let $\mathcal{S}' = \{z'^{(i,j)}, i \in \mathcal{H}, j \in \{1, \ldots, m\}\}$ be another independently sampled dataset for honest workers where for all $i \in \mathcal{H}$, $\mathcal{D}'_i = \{z'^{(i,1)}, \ldots, z'^{(i,m)}\}$ is composed of $m$ i.i.d. data points (or samples) drawn from distribution $p_i$. Let $\mathcal{S}^{(i,j)}$ be a neighboring dataset of $\mathcal{S}$ with only a single differing data sample $z'^{(i,j)}$ at index $(i,j) \in \mathcal{H} \times \{1, \ldots, m\}$. Then,

$$\mathbb{E}_{\mathcal{S},\mathcal{A}}[\widehat{R}_{\mathcal{H}}(\mathcal{A}(\mathcal{S}))] = \mathbb{E}_{\mathcal{S},\mathcal{S}',\mathcal{A}}\left[\frac{1}{|\mathcal{H}|}\sum_{i \in \mathcal{H}}\frac{1}{m}\sum_{j=1}^{m}\ell(\mathcal{A}(\mathcal{S}^{(i,j)}), z'^{(i,j)})\right]$$

$$= \mathbb{E}_{\mathcal{S},\mathcal{S}',\mathcal{A}}\left[\frac{1}{|\mathcal{H}|}\sum_{i \in \mathcal{H}}\frac{1}{m}\sum_{j=1}^{m}\ell(\mathcal{A}(\mathcal{S}), z'^{(i,j)})\right] + \delta$$

$$= \mathbb{E}_{\mathcal{S},\mathcal{A}}[R_{\mathcal{H}}(\mathcal{A}(\mathcal{S}))] + \delta,$$

with

$$\delta = \mathbb{E}_{\mathcal{S},\mathcal{S}'}\left[\frac{1}{|\mathcal{H}|}\sum_{i \in \mathcal{H}}\frac{1}{m}\sum_{j=1}^{m}\mathbb{E}_{\mathcal{A}}\left[\ell(\mathcal{A}(\mathcal{S}^{(i,j)}), z'^{(i,j)}) - \ell(\mathcal{A}(\mathcal{S}), z'^{(i,j)})\right]\right]$$

$$\leq \mathbb{E}_{\mathcal{S}}\mathbb{E}_{\mathcal{S}'}\left[\frac{1}{|\mathcal{H}|}\sum_{i \in \mathcal{H}}\frac{1}{m}\sum_{j=1}^{m}\sup_{z \in \mathcal{Z}}\mathbb{E}_{\mathcal{A}}\left[\ell(\mathcal{A}(\mathcal{S}^{(i,j)}), z) - \ell(\mathcal{A}(\mathcal{S}), z)\right]\right] \leq \varepsilon$$

Substituting from the above equation proves the lemma. □

## B. Regularity Assumptions and Expansivity Results

To analyze the stability of iterative optimization algorithms, we typically rely on regularity assumptions on the loss function. Below, we state standard regularity assumptions used throughout the paper, along with classical results that follow from them.

**Definition B.1** (Regularity assumptions). Let $\ell : \Theta \to \mathbb{R}$ differentiable, $\Theta \subset \mathbb{R}^d$.

*(i)* $\ell$ is $C$-Lipschitz continuous (or $C$-Lipschitz) if there exists $C > 0$ such that,

$$\forall u, v \in \Theta, \quad |\ell(u) - \ell(v)| \leq C\|u - v\|_2.$$

This property is equivalent to the norm of the gradient of $\ell$ being uniformly bounded by $C$.

*(ii)* $\ell$ is $L$-Lipschitz smooth (or $L$-smooth) if its gradient is $L$-Lipschitz. This is equivalent to the following smoothness inequality

$$\forall u, v \in \Theta, \quad \ell(u) \leq \ell(v) + \langle \nabla\ell(v), u - v \rangle + \frac{L}{2}\|u - v\|_2^2. \tag{14}$$

*(iii)* $\ell$ is convex if and only if, $\forall u, v \in \Theta$, $\langle \nabla\ell(u) - \nabla\ell(v), u - v \rangle \geq 0$. Moreover it is $\mu$-strongly convex if there exists $\mu > 0$ such that,

$$\forall u, v \in \Theta, \quad \ell(u) \geq \ell(v) + \langle \nabla\ell(v), u - v \rangle + \frac{\mu}{2}\|u - v\|_2^2.$$

We note that for a strongly convex function to have bounded gradients, it must be defined on a convex compact set, which then implies boundedness of both the loss and the gradient. To ensure this, we can either penalize the problem or restrict the parameter domain and apply an Euclidean projection at every step. Throughout the paper, we will tacitly assume this when referring to strongly convex functions, which does not limit the applicability of our analysis since the projection does not increase the distance between projected points.

Notably, convex and $L$-smooth functions satisfy an important property known as co-coercivity (Bach, 2024, Proposition 5.4).

**Lemma B.2.** *Let $\ell : \Theta \to \mathbb{R}$ differentiable, $\Theta \subset \mathbb{R}^d$. If $\ell$ is a convex and $L$-smooth function, then it satisfies the co-coercivity inequality,*

$$\forall u, v \in \Theta, \quad \langle \nabla\ell(u) - \nabla\ell(v), u - v \rangle \geq \frac{1}{L}\|\nabla\ell(u) - \nabla\ell(v)\|_2^2. \tag{15}$$

We focus on robust variants of distributed gradient descent and stochastic gradient descent. Below we recall properties of the one-step update. Let $\mathcal{A} \in \{\text{GD}, \text{SGD}\}$, $\ell : \Theta \to \mathbb{R}$ differentiable, $\gamma > 0$, and we denote

$$G_\gamma^{\mathcal{A}}(\theta) := \theta - \gamma\frac{1}{|\mathcal{H}|}\sum_{i \in \mathcal{H}} g^{(i)}.$$

Here, for each $i \in \mathcal{H}$, $g^{(i)}$ denotes the exact gradient $\nabla\ell(\theta)$ in the case of GD, or an unbiased estimate thereof—effectively computed with a sample from the local dataset—in the case of SGD. We recall the expansiveness definition for an update rule from Hardt et al. (2016, Definition 2.3).

**Definition B.3.** An update rule $G : \Theta \to \Theta$ is said to be $\eta$-expansive if for all $\theta, \omega \in \Theta$,

$$\|G(\theta) - G(\omega)\|_2 \leq \eta\|\theta - \omega\|_2.$$

Additionally, we rely on a direct adaptation of Hardt et al. (2016, Lemma 3.6) for the distributed context.

**Lemma B.4.** *Let $\mathcal{A} \in \{\text{GD}, \text{SGD}\}$ and $\gamma > 0$. Assume $\forall z \in \mathcal{Z}, \ell(\cdot, z) : \Theta \to \mathbb{R}$ is $L$-smooth, Then $G_\gamma^{\mathcal{A}}$ is $\eta_{G_\gamma^{\mathcal{A}}}$-expansive, with the following expansive coefficient,*

*(i)* $\eta_{G_\gamma^{\mathcal{A}}} = (1 + \gamma L)$.

*(ii)* *Assume in addition that $\ell$ is convex. Then, for any $\gamma \leq \frac{2}{L}$, $\eta_{G_\gamma^{\mathcal{A}}} = 1$.*

*(iii)* *Assume in addition that $\ell$ is $\mu$-strongly convex. Then, for any $\gamma \leq \frac{2}{\mu+L}$, $\eta_{G_\gamma^{\mathcal{A}}} = (1 - \gamma\frac{L\mu}{\mu+L})$. For $\gamma \leq \frac{1}{L}$, since we have $L \geq \mu$, we can simplify the contractive constant to $\eta_{G_\gamma^{\mathcal{A}}} = (1 - \gamma\mu)$.*

# C. Deferred Proofs from Section 3.1

We first establish supporting results in Appendix C.1, then prove Theorem 3.1.

## C.1. Expectation Bounds for Honest Gradients' Empirical Spectral Norm

We derive bounds on the expected spectral norm of the empirical covariance matrix of the honest workers' gradients under the setting of bounded gradient assumption (i.e., Lipschitz-continuity of the loss function).

**Lemma C.1.** *Consider the setting described in Section 2 under Byzantine failures. Assume the loss function $C$-Lipschitz. Let $\mathcal{A} \in \{\mathrm{GD}, \mathrm{SGD}\}$, with a $(f, \kappa)$-robust aggregation rule, for $T \in \mathbb{N}^*$ iterations. We have, for every $t \in \{1, \dots, T-1\}$,*

$$\mathbb{E}_{\mathcal{A}} \|\Sigma_{\mathcal{H},t}\|_{\mathrm{sp}} = \mathbb{E}_{\mathcal{A}}[\lambda_{\max}(\frac{1}{|\mathcal{H}|} \sum_{i \in \mathcal{H}} (g_t^{(i)} - \overline{g}_t)(g_t^{(i)} - \overline{g}_t)^{\mathsf{T}})] \leq C^2.$$

*Proof.* For the first inequality, we directly use the boundedness of gradients as follows,

$$\lambda_{\max}(\frac{1}{|\mathcal{H}|} \sum_{i \in \mathcal{H}} (g_t^{(i)} - \overline{g}_t)(g_t^{(i)} - \overline{g}_t)^{\mathsf{T}}) = \sup_{\|v\|_2 \leq 1} \frac{1}{|\mathcal{H}|} \sum_{i \in \mathcal{H}} \langle v, g_t^{(i)} - \overline{g}_t \rangle^2$$

$$\leq \sup_{\|v\|_2 \leq 1} \frac{1}{|\mathcal{H}|} \sum_{i \in \mathcal{H}} \langle v, g_t^{(i)} \rangle^2 \leq \frac{1}{|\mathcal{H}|} \sum_{i \in \mathcal{H}} \|g_t^{(i)}\|_2^2 \leq C^2. \quad \square$$

*Remark* C.2. In Appendix G, we present example results derived under refined assumptions—namely, bounded heterogeneity and bounded variance—in place of the more general bounded gradients condition (i.e., Lipschitz-continuity of the loss function). These assumptions are more commonly used in the optimization literature (as opposed to the generalization literature), and while they often yield tighter bounds, they do not provide additional conceptual insights within the scope of our discussion.

## C.2. Proof of Theorem 3.1

We begin with a unified *proof sketch* that outlines the key ideas to bound stability under Byzantine failures. We then prove Theorem 3.1.

**Proof sketch.** Let denote $\{\theta_t\}_{t \in \{0,\dots,T-1\}}$ and $\{\theta'_t\}_{t \in \{0,\dots,T-1\}}$ the optimization trajectories resulting from two neighboring datasets $\mathcal{S}, \mathcal{S}'$, for $T$ iterations of $\mathcal{A} \in \{\mathrm{GD}, \mathrm{SGD}\}$ with $F$ a $(f, \kappa)$-robust aggregation rule, and learning rate schedule $\{\gamma_t\}_{t \in \{0,\dots,T-1\}}$. In the context of the uniform algorithmic stability analysis via parameter sensitivity, we track how the parameters diverge along these optimization trajectories. To do so, we decompose the analysis by introducing an intermediate comparison between the robust update $G_{\gamma_t}^F$ and the averaging over honest workers update $G_{\gamma_t}^{\mathcal{A}}$. This enables us to leverage either the regularity of the loss function or the robustness property of the aggregation rule. This approach is primarily motivated by the fact that Byzantine vectors cannot be assumed to exhibit regularities when comparing them.

$$\mathbb{E}_{\mathcal{A}} \left[ \|G_{\gamma_t}^F(\theta_t) - G_{\gamma_t}^{F\prime}(\theta'_t)\|_2 \right] = \mathbb{E}_{\mathcal{A}}[\|G_{\gamma_t}^F(\theta_t) - G_{\gamma_t}^{\mathcal{A}}(\theta_t) + G_{\gamma_t}^{\mathcal{A}}(\theta_t) - G_{\gamma_t}^{\mathcal{A}\prime}(\theta'_t) + G_{\gamma_t}^{\mathcal{A}\prime}(\theta'_t) - G_{\gamma_t}^{F\prime}(\theta'_t)\|_2]$$

$$\leq \underbrace{\mathbb{E}_{\mathcal{A}} \|G_{\gamma_t}^{\mathcal{A}}(\theta_t) - G_{\gamma_t}^{\mathcal{A}\prime}(\theta'_t)\|_2}_{A} + \underbrace{\mathbb{E}_{\mathcal{A}} \left[ \|G_{\gamma_t}^F(\theta_t) - G_{\gamma_t}^{\mathcal{A}}(\theta_t)\|_2 + \|G_{\gamma_t}^{F\prime}(\theta'_t) - G_{\gamma_t}^{\mathcal{A}\prime}(\theta'_t)\|_2 \right]}_{B}.$$

We then bound the term $A$ with classical stability tools from Hardt et al. (2016), and the term $B$ with the $(f, \kappa)$-robust property, the Jensen inequality and the Lemma C.1,

$$B \leq 2\gamma_t \mathbb{E}_{\mathcal{A}} \sqrt{\kappa \|\Sigma_{\mathcal{H},t}\|_{\mathrm{sp}}} \leq 2\gamma_t \sqrt{\kappa \mathbb{E}_{\mathcal{A}} \|\Sigma_{\mathcal{H},t}\|_{\mathrm{sp}}} \leq 2\gamma_t \sqrt{\kappa} C.$$

**Theorem 3.1.** Consider the setting described in Section 2 under Byzantine failures. Let $\mathcal{A} \in \{\mathrm{GD}, \mathrm{SGD}\}$, with $F$ a $(f, \kappa)$-robust aggregation rule, for $T \in \mathbb{N}^*$ iterations. Suppose $\forall z \in \mathcal{Z}$, $\ell(\cdot; z)$ $C$-Lipschitz and $L$-smooth.

*(i)* Assume $\forall z \in \mathcal{Z}$, $\ell(\cdot; z)$ convex, with constant learning rate $\gamma \leq \frac{2}{L}$. Then, for any neighboring datasets $\mathcal{S}, \mathcal{S}'$, we have

$$\sup_{z \in \mathcal{Z}} \mathbb{E}_{\mathcal{A}}[|\ell(\mathcal{A}(\mathcal{S}); z) - \ell(\mathcal{A}(\mathcal{S}'); z)|] \leq 2\gamma C^2 T \left( \frac{1}{(n-f)m} + \sqrt{\kappa} \right).$$

*(ii)* Assume $\forall z \in \mathcal{Z}$, $\ell(\cdot; z)$ $\mu$-strongly convex, with $\gamma \leq \frac{1}{L}$. Then, for any neighboring datasets $\mathcal{S}, \mathcal{S}'$, we have

$$\sup_{z \in \mathcal{Z}} \mathbb{E}_{\mathcal{A}}[|\ell(\mathcal{A}(\mathcal{S}); z) - \ell(\mathcal{A}(\mathcal{S}'); z)|] \leq \frac{2C^2}{\mu} \left( \frac{1}{(n-f)m} + \sqrt{\kappa} \right).$$

*Proof.* Let first reason about $\mathcal{A} = \text{GD}$. For each $t \in \{0, \ldots, T-1\}$, following the proof sketch in C.2, we bound the term $\|G^{\text{GD}}_{\gamma_t}(\theta_t) - G^{\text{GD}\prime}_{\gamma_t}(\theta'_t)\|_2$ where $G^{\text{GD}}_{\gamma_t}(\theta_t) = \theta_t - \gamma_t \nabla \widehat{R}_{\mathcal{H}}(\theta_t)$. Let $a, b \in \mathcal{H} \times \{1, \ldots, m\}$ denote the indices of the differing samples. For $t \in \{0, \ldots, T-1\}$, we have,

$$\|G^{\text{GD}}_{\gamma_t}(\theta_t) - G^{\text{GD}\prime}_{\gamma_t}(\theta'_t)\|_2 = \|G^{\text{GD}}_{\gamma_t}(\theta_t) - G^{\text{GD}}_{\gamma_t}(\theta'_t)\|_2 + \frac{\gamma_t}{|\mathcal{H}|m} \|\nabla\ell(\theta'_t; z'^{(a,b)}) - \nabla\ell(\theta'_t; z^{(a,b)})\|_2$$

$$\leq \eta_{G^{\text{GD}}_{\gamma_t}} \|\theta_t - \theta'_t\|_2 + \frac{2\gamma_t C}{(n-f)m},$$

where $\eta^{\text{GD}}_{G_{\gamma_t}}$ denotes the expansive coefficient, which depends on the regularity properties of the loss function (cf. Lemma B.4). Let define $\sigma^F_{\text{GD}} = 2C\left(\frac{1}{(n-f)m} + \sqrt{\kappa}\right)$, we then resolve the recursion by incorporating the impact of the Byzantine failures as demonstrated in the previous proof sketch.

*(i)* If the loss function is Lipschitz, smooth, and convex, then for all $t \in \{0, \ldots, T-1\}$,

$$\|\theta_{t+1} - \theta'_{t+1}\|_2 \leq \|\theta_t - \theta'_t\|_2 + \gamma\sigma^F_{\text{GD}}.$$

To conclude the proof, we assume a constant learning rate $\gamma \leq \frac{1}{L}$ and sum the inequality over $t \in 0, \ldots, T-1$, noting that $\theta_0 = \theta'_0$. We then invoke the Lipschitz continuity assumption.

*(ii)* If the loss function is Lipschitz, smooth, and strongly convex, then for all $t \in \{0, \ldots, T-1\}$,

$$\|\theta_{t+1} - \theta'_{t+1}\|_2 \leq (1 - \gamma\mu)\|\theta_t - \theta'_t\|_2 + \gamma\sigma^F_{\text{GD}}.$$

Considering $\gamma \leq \frac{1}{L}$ and summing over $t \in \{0, \ldots, T-1\}$, with $\theta_0 = \theta'_0$, yields

$$\|\theta_T - \theta'_T\|_2 \leq \gamma\sigma^F_{\text{GD}} \sum_{t=0}^{T-1} (1 - \gamma\mu)^t = \frac{\sigma^F_{\text{GD}}}{\mu} \left[ 1 - (1 - \gamma\mu)^T \right] \leq \frac{\sigma^F_{\text{GD}}}{\mu}.$$

We conclude the derivation of the bound by invoking the Lipschitz continuity assumption.

For $\mathcal{A} = \text{SGD}$, the same recursion holds for the expected distance $\mathbb{E}_{\mathcal{A}}\|\theta_t - \theta'_t\|_2$ for $t \in \{1, \ldots, T\}$, where $G^{\text{SGD}}_{\gamma_t}(\theta_t) = \theta_t - \gamma_t \frac{1}{|\mathcal{H}|} \sum_{i \in \mathcal{H}} g^{(i)}_t$, and for each $i \in \mathcal{H}$, $g^{(i)}_t = \nabla\ell(\theta_t; z^{(i, J^{(i)}_t)}_t)$ with $J^{(i)}_t$ sampled uniformly from $\{1, \ldots, m\}$. In fact, at each iteration, a single sample is selected uniformly at random from each honest worker. Consequently, with probability $(1 - \frac{1}{m})$, the differing samples are not selected,

$$\|G^{\text{SGD}}_{\gamma_t}(\theta_t) - G^{\text{SGD}\prime}_{\gamma_t}(\theta'_t)\|_2 = \|G^{\text{SGD}}_{\gamma_t}(\theta_t) - G^{\text{SGD}}_{\gamma_t}(\theta'_t)\|_2 \leq \eta_{G^{\text{SGD}}_{\gamma_t}} \|\theta_t - \theta'_t\|_2,$$

and with probability $\frac{1}{m}$, the differing samples are selected,

$$\|G^{\text{SGD}}_{\gamma_t}(\theta_t) - G^{\text{SGD}\prime}_{\gamma_t}(\theta'_t)\|_2 = \|G^{\text{SGD}}_{\gamma_t}(\theta_t) - G^{\text{SGD}}_{\gamma_t}(\theta'_t)\|_2$$

$$+ \frac{\gamma_t}{|\mathcal{H}|} \|\nabla\ell(\theta'_t; z'^{(a,b)}) - \nabla\ell(\theta'_t; z^{(a,b)})\|_2$$

$$\leq \eta_{G^{\text{SGD}}_{\gamma_t}} \|\theta_t - \theta'_t\|_2 + \frac{2\gamma_t C}{|\mathcal{H}|}.$$

We thus obtain the following bound

$$\mathbb{E}_{\mathcal{A}}[\|G^{\text{SGD}}_{\gamma_t}(\theta_t) - G^{\text{SGD}\prime}_{\gamma_t}(\theta'_t)\|_2] \leq \eta_{G^{\text{SGD}}_{\gamma_t}} \mathbb{E}_{\mathcal{A}}[\|\theta_t - \theta'_t\|_2] + \frac{2\gamma_t C}{(n-f)m}.$$

Let define $\sigma_{\text{SGD}}^F = 2C\left(\frac{1}{(n-f)m} + \sqrt{\kappa}\right)$. We then resolve the recursion analogously as in the GD case above, by incorporating the impact of the Byzantine failures as demonstrated in the previous proof sketch. This yields

$$\mathbb{E}_{\mathcal{A}}[\|\theta_{t+1} - \theta'_{t+1}\|_2] \leq \eta_{G_{\gamma_t}^{\text{SGD}}} \mathbb{E}_{\mathcal{A}}[\|\theta_t - \theta'_t\|_2] + \gamma_t \sigma_{\text{SGD}}^F, \tag{16}$$

which conclude the proof. $\qquad\square$

## C.3. Proof of Theorem 3.2

**Theorem 3.2.** Consider the setting described in Section 2 under Byzantine failures. Let $\mathcal{A} = \text{GD}$, with SMEA. Suppose $\mathcal{A}$ is run for $T \in \mathbb{N}^*$ iterations with the learning rate $\gamma$. Then for both regime stated below, there exist $\ell \in \mathbb{R}^{\Theta \times \mathcal{Z}}$ such that $\forall z \in \mathcal{Z}, \ell(\cdot; z)$ is $C$-Lipschitz, $L$-smooth and convex, and a pair of neighboring datasets such that the uniform stability is lower bounded by

*(i)* in the regime of moderate to high fractions of misbehaving workers $\frac{n}{3} \leq f < \frac{n}{2}$,

$$\Omega\left(\gamma C^2 T \left(\sqrt{\frac{f}{n-2f}} + 1 + \frac{1}{(n-f)m}\right)\right).$$

*(ii)* in the regime of low to moderate fractions of misbehaving workers $f < \frac{n}{3}$,

$$\Omega\left(\gamma C^2 T \frac{f}{n-2f}\right).$$

*(iii)* instead, if we use $\mathcal{A} = \text{SGD}$, and assume there exists $k > 0$ such that $\mathcal{A}$ is run for $T \geq km$ iterations with the learning rate $\gamma$ and $\frac{n}{3} \leq f < \frac{n}{2}$. Then,

$$\Omega\left(\gamma C^2 T \left(\sqrt{\frac{f}{n-2f}} + 1 + \frac{1}{(n-f)m}\right)\right).$$

*Proof.* The lower bound construction relies on two key elements. First, we maximize the variance among the honest gradients by dividing them into two equal subgroups positioned at the boundary of the Lipschitz constraint. Second, in the regime with a high proportion of Byzantine workers, this setup enables the adversary to completely discard one of the honest subgroups and even outweigh the influence of the other. The extent to which the adversary can bias the aggregate away from the true direction (which is zero by construction) depends on the robustness of the aggregation rule.

In what follows, we provide the proof for the case $d = 1$. The result extends naturally to higher dimensions by embedding the one-dimensional construction into $\mathbb{R}^d$. We arbitrarily refer to one trajectory as *unperturbed*, with parameter $\theta_t$, and the other as *perturbed*, with $\theta'_t$. Let $C, L \in \mathbb{R}_+^*$, we define the following loss function

$$\text{For } \theta \in \mathbb{R}, \ z \in [-C, C], \quad \ell(\theta; z) = z\theta, \quad \ell'(\theta; z) = z, \quad \ell''(\theta; z) = 0.$$

By construction, the loss function is convex, $C$-Lipschitz and $L$-smooth. Let $m, n, f \in \mathbb{N}^*$ such that $\frac{n}{3} \leq f < \frac{n}{2}$. Let $\gamma$ the learning rate and assume $\theta_0 = \theta'_0 = 0$.

***(i.A)If $n - f$ is even and $f \geq \frac{n}{3}$.*** We define the following pair of neighboring datasets.

$$S = \{D_1, \ldots, D_{n-f}\}, \quad S' = S \setminus \{D_1\} \cup \{D_1 \setminus \{z^{(1,1)}\} \cup \{z'^{(1,1)}\}\}$$
$$\text{with } \forall i \in \{1, \ldots, n-f\}, \quad D_i = \{z^{(i,1)}, \ldots, z^{(i,m)}\}$$
$$\text{and } \forall j \in \{1, \ldots, m\}, \quad i \in \mathcal{H}_+ = \{1, \ldots, \frac{n-f}{2}\}, |\mathcal{H}_+| = \frac{n-f}{2} : z^{(i,j)} = C \text{ and } z'^{(1,1)} = 0.$$
$$i \in \mathcal{H}_- = \{\frac{n-f}{2} + 1, \ldots, n-f\}, |\mathcal{H}_-| = \frac{n-f}{2} : z^{(i,j)} = -C.$$

For $t \in \{0, \ldots, T-1\}$, we have for $i \in \mathcal{H}_-$, $g_t^{(i)} = g_t^{(i)\prime} = -C$; for $i \in \mathcal{H}_+ \setminus \{1\}$, $g_t^{(i)} = g_t^{(i)\prime} = C$; and for $i = 1$, $g_t^{(1)} = C$ and $g_t^{(1)\prime} = \frac{m-1}{m}C < C$. We consider $f$ Byzantine workers, and let denote $\mathcal{B} = \{n-f+1, \ldots, n\}$. Since they can observe all communications, they may adapt their strategy based on the message sent by worker 1. We define their behavior in Algorithm 1.

---

**Algorithm 1** Byzantine Worker Behavior

    **if** $g_1 < C$ **then**
        **return** an arbitrarily large value (e.g., $n\beta$)
    **else**
        **return** $\beta$ (to be specified below)
    **end if**

---

In the first case, the subset selected by the SMEA aggregation rule consists entirely of honest workers. Consequently, we have $\theta_T' = \frac{\gamma C T}{m(n-f)}$. In the second case, we choose

$$\beta = \left(1 + \frac{1}{2}\frac{n-f}{\sqrt{f(n-2f)}}\right)C,$$

such that the SMEA aggregation rule selects the subset $\mathrm{Select}_{n-2f}(\mathcal{H}_+) \cup \mathcal{B}$, where $\mathrm{Select}_k(E)$ denote any subset of size $k$ from a set $E$, i.e., all the Byzantine workers together with $n-2f$ workers from $\mathcal{H}_+$. This is made possible because:

- $\mathrm{Var}(\{g_t^{(i)}\}_{i \in \mathcal{H}}) = C^2$, and any subset $\mathrm{Select}_a(\mathcal{H}_-) \cup \mathrm{Select}_b(\mathcal{H}_+) \cup \mathrm{Select}_{n-f-a-b}(\mathcal{B})$, with $0 < a < \frac{n-f}{2}$, $0 \leq b < \frac{n-f}{2}$ yields higher variance.

- As $f \geq n/3$, $f \geq \frac{n-f}{2} \geq n-2f$. Hence, either $f = \frac{n-f}{2}$ and

$$\mathcal{H}_+ \cup \mathrm{Select}_{\frac{n-3f}{2}}(\mathcal{B}) = \mathrm{Select}_{n-2f}(\mathcal{H}_+) \cup \mathcal{B} = \mathcal{H}_+ \cup \mathcal{B},$$

  or $f > \frac{n-f}{2}$ and $\mathrm{Var}(\{g_t^{(i)}\}_{i \in \mathcal{H}_+ \cup \mathrm{Select}_{\frac{n-3f}{2}}(\mathcal{B})}) > \mathrm{Var}(\{g_t^{(i)}\}_{i \in \mathrm{Select}_{n-2f}(\mathcal{H}_+) \cup \mathcal{B}})$. Hence, the aggregation rule will always pick a larger subgroup over a smaller one (it minimizes the variance magnitude).

- $\mathrm{Var}(\{g_t^{(i)}\}_{i \in \mathrm{Select}_{n-2f}(\mathcal{H}_+) \cup \mathcal{B}}) = \frac{C^2}{4} < C^2 = \mathrm{Var}(\{g_t^{(i)}\}_{i \in \mathcal{H}})$. In fact,

$$
\begin{aligned}
\mathrm{Var}(\{g_t^{(i)}\}_{i \in \mathrm{Select}_{n-2f}(\mathcal{H}_+) \cup \mathcal{B}}) &= \frac{f}{n-f}\left[\beta - \frac{1}{n-f}(f\beta + (n-2f)C)\right]^2 \\
&\quad + \frac{n-2f}{n-f}\left[C - \frac{1}{n-f}(f\beta + (n-2f)C)\right]^2 \\
&= \frac{f(n-2f)}{(n-f)^2}(\beta - C)^2 \\
&= \frac{f(n-2f)}{(n-f)^2}C^2\left(1 + \frac{1}{2}\frac{n-f}{\sqrt{f(n-2f)}} - 1\right)^2 \\
&= \frac{C^2}{4}.
\end{aligned}
$$

This reasoning applies to all iterations because the gradients, by construction, stay constant during the entire optimization process. As a result,

$$\theta_T = \theta_0 - \gamma T \frac{1}{n-f} \sum_{i \in \text{Select}_{n-2f}(\mathcal{H}_+) \cup \mathcal{B}} g_0^{(i)}$$

$$= -\frac{\gamma T}{n-f} \left( fC \left( 1 + \frac{1}{2} \frac{n-f}{\sqrt{f(n-2f)}} \right) + (n-2f)C \right)$$

$$= -\gamma T C \left( 1 + \frac{1}{2} \sqrt{\frac{f}{n-2f}} \right).$$

Hence,

$$\sup_{z \in [-C,C]} |\ell(\theta_T; z) - \ell(\theta_T'; z)| = \sup_{z \in [-C,C]} z \left( |\theta_T - \theta_T'| \right)$$

$$= \gamma C^2 T \left( 1 + \frac{1}{2} \sqrt{\frac{f}{n-2f}} + \frac{1}{(n-f)m} \right) \in \Omega \left( \gamma C^2 T \left( \sqrt{\frac{f}{n-2f}} + \frac{1}{(n-f)m} \right) \right).$$

***(i.B)*** **If $n - f$ is odd and $f \geq \frac{n}{3}$.** The same construction works with a slight adaptation. That is, we define the following pair of neighboring datasets.

$$S = \{D_1, \ldots, D_{n-f}\}, \quad S' = S \setminus \{D_1\} \cup \{D_1 \setminus \{z^{(1,1)}\} \cup \{z'^{(1,1)}\}\}$$

$$\text{with } \forall i \in \{1, \ldots, n-f\}, \quad D_i = \{z^{(i,1)}, \ldots, z^{(i,m)}\}$$

$$\text{and } \forall j \in \{1, \ldots, m\}, \quad i \in \mathcal{H}_+ = \{1, \ldots, \frac{n-f-1}{2}\}, |\mathcal{H}_+| = \frac{n-f-1}{2}: z^{(i,j)} = C \text{ and } z'^{(1,1)} = 0.$$

$$i \in \mathcal{H}_- = \{\frac{n-f-1}{2} + 1, \ldots, n-f-1\}, |\mathcal{H}_-| = \frac{n-f-1}{2}: z^{(i,j)} = -C.$$

$$z^{(n-f,j)} = 0.$$

Then,

$$\text{Var}(\{g_t^{(i)}\}_{i \in \mathcal{H}}) = (1 - \frac{1}{n-f})C^2 > \frac{C^2}{3} \text{ as } n - f > \frac{n}{2} \geq \frac{3}{2}.$$

The remainder of the proof when $n - f$ is odd proceeds identically to the previous case when $n - f$ is even.

***(ii)*** **In the regime $f < n/3$.** For values restricted to $z \in [-C, C]$, variance is maximized by placing all points at the endpoints $\pm C$, with counts as balanced as possible. Thus we take

$$k = \left\lceil \frac{n-f}{2} \right\rceil \quad \text{points at } + C, \qquad \left\lfloor \frac{n-f}{2} \right\rfloor = n - f - k \quad \text{points at } - C.$$

We have two cases, if $n - f$ is even then the variance equals $C^2$, else it equals $C^2 \left( 1 - \frac{1}{(n-f)^2} \right)$. Because $f < n/3$ we have $n - 2f > (n-f)/2$. Hence, we would like to consider the subset $S$ composed by $f$ times $\beta$, $k$ times $C$ and $n - 2f - k$ times $-C$. We compute the competing set variance by the identity $\text{Var}(X) = \mathbb{E}[X^2] - \mathbb{E}[X]^2$,

$$\text{Var}(S) = \frac{(n-2f)C^2 + f\beta^2}{n-f} - \frac{\left( (2k - n + 2f)C + f\beta \right)^2}{(n-f)^2}.$$

Requiring $\text{Var}(S) \leq \text{Var}(\mathcal{H}) = C^2 \left( 1 - \frac{(2k-n+f)^2}{(n-f)^2} \right)$ yields the quadratic inequality in $\beta$

$$(n-2f)\beta^2 + 2(n-2f-2k)C\beta - (4k - n + 2f)C^2 \leq 0.$$

The discriminant simplifies exactly to $16C^2k^2$, so the two roots are

$$\beta_- = -C, \qquad \beta_+ = \frac{4k - (n - 2f)}{n - 2f}C = \left(\frac{4k}{n - 2f} - 1\right)C.$$

Since $(n - 2f) > 0$, the admissible $\beta$ lie between the two roots. Moreover, since $\frac{2k}{n-2f} = \frac{2\left\lceil\frac{n-f}{2}\right\rceil}{n-2f} \geq \frac{n-f}{n-2f}$, we have $\frac{4k}{n-2f} - 1 \geq 2\frac{n-f}{n-2f} - 1 = \frac{2f}{n-2f} + 1 > 1$. Next, we have

$$\theta_T = -\gamma CT \frac{2k - n + 2f + \frac{4fk}{n-2f} - f}{n - f} = -\gamma CT\left(\frac{4fk}{(n-f)(n-2f)} + \frac{2k - (n-f)}{n - f}\right),$$

with $2k \geq n - f$. Hence, similarly to the previous cases, we have $\theta_T' > 0$, resulting in

$$\sup_{z\in[-C,C]} |\ell(\theta_T; z) - \ell(\theta_T'; z)| = \sup_{z\in[-C,C]} z\left(|\theta_T - \theta_T'|\right) \geq \gamma C^2 T \frac{2f}{n - 2f}.$$

*(iii)* **For** SGD.  We prove a similar lower bound as for GD. Because $T \geq km$, this ensures that the differing sample has non-negligeable probability to be drawn at least once. As the differing sample might not be drawn immediately, we have to adapt Algorithm 2 so that the attacker's trigger depends on all previously observed gradients. The proof follows from Step (i.A) and (i.B), adapting the Byzantine behavior as follows.

---

**Algorithm 2** Byzantine Worker Behavior

---

**if** $\forall s \in [t], g_s^{(1)} < C$ **then**
    **return** an arbitrarily large value (e.g., $n\beta$)
**else**
    **return** $\beta$
**end if**

---

Under the condition $T \geq km$, the probability of sampling the differing sample is lower bounded by a constant $(1 - e^{-k})$. This allows us to recover the same divergence as in Step (i.A) over a constant fraction of the $T$ iterations, yielding the desired lower bound. Indeed, let the even $E = \{$The differing sample is selected at least once during the first $T/2$ iterations$\}$. Then,

$$\sup_{z\in[-C,C]} \mathbb{E}_{\mathcal{A}}|\ell(\theta_T; z) - \ell(\theta_T'; z)| \geq \sup_{z\in[-C,C]} \mathbb{E}_{\mathcal{A}}\left[|\ell(\theta_T; z) - \ell(\theta_T'; z)| \mid E\right]\mathbb{P}(E)$$

$$\geq (1 - e^{\frac{k}{2}})\gamma C^2(T - \frac{T}{2})\left(1 + \frac{1}{2}\sqrt{\frac{f}{n - 2f}} + \frac{1}{(n - f)m}\right)$$

$$\in \Omega\left(\gamma C^2 T\left(\sqrt{\frac{f}{n - 2f}} + \frac{1}{(n - f)m}\right)\right).$$

$\square$

# D. Deferred Proofs from Section 3.2

In the following section, we establish uniform algorithmic upper bounds and lower bounds under data poisoning, leveraging the structure of the SMEA robust aggregation rule.

## D.1. Proof of Theorem 3.3

**Theorem 3.3.** Consider the setting described in Section 2 under data poisoning. Let $\mathcal{A} \in \{\text{GD}, \text{SGD}\}$, with SMEA, for $T \in \mathbb{N}^*$ iterations. Suppose $\forall z \in \mathcal{Z}, \ell(\cdot; z)$ $C$-Lipschitz and $L$-smooth.

*(i)* Assume $\forall z \in \mathcal{Z}$, $\ell(\cdot; z)$ convex, with constant learning rate $\gamma \leq \frac{2}{L}$. Then, for any neighboring datasets $\mathcal{S}, \mathcal{S}'$, we have

$$\sup_{z \in \mathcal{Z}} \mathbb{E}_{\mathcal{A}}[|\ell(\mathcal{A}(\mathcal{S}); z) - \ell(\mathcal{A}(\mathcal{S}'); z)|] \leq 2\gamma C^2 T \left( \frac{f}{n-f} + \frac{1}{(n-f)m} \right).$$

*(ii)* Assume $\forall z \in \mathcal{Z}$, $\ell(\cdot; z)$ $\mu$-strongly convex, with $\gamma \leq \frac{1}{L}$. Then, for any neighboring datasets $\mathcal{S}, \mathcal{S}'$, we have

$$\sup_{z \in \mathcal{Z}} \mathbb{E}_{\mathcal{A}}[|\ell(\mathcal{A}(\mathcal{S}); z) - \ell(\mathcal{A}(\mathcal{S}'); z)|] \leq \frac{2C^2}{\mu} \left( \frac{f}{n-2f} + \frac{1}{(n-2f)m} \right).$$

*Proof.* Let first reason about $\mathcal{A} = \text{GD}$. Let denote $\{\theta_t\}_{t \in \{0,\dots,T-1\}}$ and $\{\theta'_t\}_{t \in \{0,\dots,T-1\}}$ the optimization trajectories resulting from two neighboring datasets $\mathcal{S}, \mathcal{S}'$, for $T$ iterations of GD with aggregation rule SMEA and learning rate schedule $\{\gamma_t\}_{t \in \{0,\dots,T-1\}}$. The proof proceeds, by analyzing our robust aggregation as a classical gradient descent, albeit with a random data-dependent subset selection at each iteration. Notably, the poisoned vectors are treated as true gradients, although computed on arbitrary data. For $t \in \{0, \dots, T-1\}$, let denote,

$$S_t^* \in \underset{S \subseteq \{1,\dots,n\}, |S|=n-f}{\arg\min} \lambda_{\max} \left( \frac{1}{|S|} \sum_{i \in S} \left( g_t^{(i)} - \overline{g}_S \right) \left( g_t^{(i)} - \overline{g}_S \right)^\mathsf{T} \right)$$

with $\overline{g}_S = \frac{1}{|S|} \sum_{i \in S} g_t^{(i)}$, where $g_t^{(i)} = \nabla \widehat{R}_i(\theta_t)$ and the prime notation $(\cdot)'$ denotes the corresponding quantities for $\mathcal{S}'$. At each step $t \in \{0, \dots, T-1\}$, the intersection and differences of the two subsets $S_t^*$ and $S_t^{*\prime}$ verifies the following properties

$$n - 2f \leq |S_t^* \cap S_t^{\prime*}| \leq n - f \quad \text{and} \quad |S_t^* \setminus S_t^{\prime*}| + |S_t^{\prime*} \setminus S_t^*| \leq 2f. \tag{17}$$

Let denote

$$G^{\text{GD}}_{\gamma_t|_{S_t^* \cap S_t^{\prime*}}}(\theta_t) = \theta_t - \frac{\gamma_t}{n-f} \sum_{i \in S_t^* \cap S_t^{\prime*}} \nabla \widehat{R}_i(\theta_t), \tag{18}$$

and

$$G^{\text{GD}}_{\gamma_t|_{S_t^* \cap S_t^{\prime*}}}(\theta'_t) = \theta'_t - \frac{\gamma_t}{n-f} \sum_{i \in S_t^* \cap S_t^{\prime*}} \nabla \widehat{R}_i(\theta'_t).$$

Let $i \notin \mathcal{H}$, the misbehaving vectors are gradients computed on the true loss function. This enables us to exploit their regularities if $i \in S_t^* \cap S_t^{\prime*}$ or their boundedness if $i \in S_t^* \setminus S_t^{\prime*} \cup S_t^{\prime*} \setminus S_t^*$. This is to be contrasted with the previously studied Byzantine failures—where an intermediate comparison was introduced—as we adopt a more direct bounding technique when subject to data poisoning. Let $a, b \in \mathcal{H} \times \{1, \dots, m\}$ the indices of the differing samples. Using these relations, we derive the following bound on $\delta_{t+1} = \|\theta_{t+1} - \theta'_{t+1}\|_2$

$$\begin{aligned}
\delta_{t+1} \leq \quad & \|G^{\text{GD}}_{\gamma_t|_{S_t^* \cap S_t^{\prime*}}}(\theta_t) - G^{\text{GD}\prime}_{\gamma_t|_{S_t^* \cap S_t^{\prime*}}}(\theta'_t)\|_2 + \frac{\gamma_t}{n-f} \| \sum_{i \in S_t^* \setminus S_t^{\prime*}} \nabla \widehat{R}_i(\theta_t) - \sum_{i \in S_t^{\prime*} \setminus S_t^*} \nabla \widehat{R}_i(\theta'_t) \|_2 \\
\leq \quad & \|G^{\text{GD}}_{\gamma_t|_{S_t^* \cap S_t^{\prime*}}}(\theta_t) - G^{\text{GD}}_{\gamma_t|_{S_t^* \cap S_t^{\prime*}}}(\theta'_t)\|_2 + \frac{\gamma_t}{(n-f)m} \|\nabla \ell(\theta_t, z^{(a,b)}) - \nabla \ell(\theta'_t, z_t^{\prime(a,b)})\|_2 \\
& + \frac{\gamma_t}{n-f} \| \sum_{i \in S_t^* \setminus S_t^{\prime*}} \nabla \ell(\theta_t, z_t^{(i)}) - \sum_{i \in S_t^{\prime*} \setminus S_t^*} \nabla \ell(\theta'_t, z_t^{\prime(i)}) \|_2 \\
\leq \quad & \|G^{\text{GD}}_{\gamma_t|_{S_t^* \cap S_t^{\prime*}}}(\theta_t) - G^{\text{GD}}_{\gamma_t|_{S_t^* \cap S_t^{\prime*}}}(\theta'_t)\|_2 + 2\gamma_t C \frac{f + \frac{1}{m}}{n-f}.
\end{aligned}$$

Hence, by Lemma B.4, we obtain the following recursion

$$\delta_{t+1} \leq \eta_{G^{\text{GD}}_{\gamma_t|_{S_t^* \cap S_t^{\prime*}}}} \delta_t + 2\gamma_t \frac{f + \frac{1}{m}}{n-f} C, \tag{19}$$

Moreover, denote $\rho = \frac{|S_t^* \cap S_t^{\prime*}|}{n-f}$, where $0 < \frac{n-2f}{n-f} \leq \rho \leq 1$ by (17). We aim to relate the expansivity of $G^{\text{GD}}_{\gamma_t|_{S_t^* \cap S_t^{\prime*}}}$, denoted $\eta_{G^{\text{GD}}_{\gamma_t|_{S_t^* \cap S_t^{\prime*}}}}$, to the expansivity of $G^{\text{GD}}_{\gamma_t}$, denoted $\eta_{G^{\text{GD}}_{\gamma_t}}$. Fist, note that we we can write (18) as follows

$$G^{\text{GD}}_{\gamma_t|_{S_t^* \cap S_t^{\prime*}}}(\theta) = (1-\rho)\theta + \rho \left( \theta - \frac{\gamma_t}{|S_t^* \cap S_t^{\prime*}|} \sum_{i \in S_t^* \cap S_t^{\prime*}} \nabla \widehat{R}_i(\theta) \right). \tag{20}$$

The term in the parenthesis of (20) corresponds to the standard gradient descent step averaged over the set $S_t^* \cap S_t'^*$. Hence, by Lemma B.4, its expansive coefficient is $\eta_{G_{\gamma_t}^{\mathrm{GD}}}$. By the triangular inequality, we then have

$$\|G_{\gamma_t|S_t^* \cap S_t'^*}^{\mathrm{GD}}(\theta) - G_{\gamma_t|S_t^* \cap S_t'^*}^{\mathrm{GD}}(\theta')\|_2 \le (1-\rho)\|\theta - \theta'\|_2 + \rho\eta_{G_{\gamma_t}^{\mathrm{GD}}}\|\theta - \theta'\|_2 = \big(1 + \rho(\eta_{G_{\gamma_t}^{\mathrm{GD}}} - 1)\big)\|\theta - \theta'\|_2.$$

That is,

$$\eta_{G_{\gamma_t|S_t^* \cap S_t'^*}^{\mathrm{GD}}} \le \big(1 + \rho(\eta_{G_{\gamma_t}^{\mathrm{GD}}} - 1)\big). \tag{21}$$

Now, assume there exists $\alpha \ge 0$ such that $\eta_{G_{\gamma_t}^{\mathrm{GD}}} = 1 + \alpha$. then, by (21) and (17), we have $\eta_{G_{\gamma_t|S_t^* \cap S_t'^*}^{\mathrm{GD}}} \le 1 + \rho\alpha \le 1 + \alpha = \eta_{G_{\gamma_t}^{\mathrm{GD}}}$. Finally, assume there exists $0 < \alpha < 1$ such that $\eta_{G_{\gamma_t}^{\mathrm{GD}}} = 1 - \alpha$, then, by (21) and (17), we have $\eta_{G_{\gamma_t|S_t^* \cap S_t'^*}^{\mathrm{GD}}} \le 1 - \alpha\rho \le 1 - \alpha\rho \le 1 - \alpha\frac{n-2f}{n-f}$. The precise value of the expansivity coefficient depends on the regularity assumption on the loss function. We then solve the recursions.

(i) Using (19) and (21) in the smooth and convex case, we have, for $t \in \{0, \ldots, T-1\}$,

$$\delta_{t+1} \le \delta_t + 2\gamma C\frac{f + \frac{1}{m}}{n - f}.$$

We finish the proof by considering a constant learning rate $\gamma \le \frac{1}{L}$ and by summing over $t \in \{0, \ldots, T-1\}$ with $\theta_0 = \theta_0'$. We then invoke the Lipschitz continuity assumption.

(ii) Using (19) and (21) in the smooth and strongly convex case, we have for $t \in \{0, \ldots, T-1\}$,

$$\delta_{t+1} \le (1 - \gamma\mu\frac{n-2f}{n-f})\delta_t + 2\gamma C\frac{f + \frac{1}{m}}{n - f}.$$

Considering $\gamma \le \frac{1}{L}$ and summing over $t \in \{0, \ldots, T-1\}$, with $\theta_0 = \theta_0'$, yields

$$\delta_T \le 2\gamma C\frac{f + \frac{1}{m}}{n - f}\sum_{t=0}^{T-1}\left(1 - \gamma\mu\frac{n-2f}{n-f}\right)^t = 2\gamma C\frac{f + \frac{1}{m}}{n - f}\frac{1}{\gamma\mu\frac{n-2f}{n-f}}\left[1 - \left(1 - \gamma\mu\frac{n-2f}{n-f}\right)^T\right] \le \frac{2C}{\mu}\frac{f + \frac{1}{m}}{n - 2f}.$$

We conclude the derivation of the bound by invoking the Lipschitz continuity assumption.

For $\mathcal{A} = \mathrm{SGD}$, the same recursion holds for the expected distance $\delta_t = \mathbb{E}_{\mathcal{A}}\|\theta_t - \theta_t'\|_2$ for $t \in \{1, \ldots, T\}$. We use similar notation as above, where, for $t \in \{1, \ldots, T\}$, $G_{\gamma_t}^{\mathrm{SGD}}(\theta_t) = \theta_t - \gamma_t\frac{1}{|\mathcal{H}|}\sum_{i \in \mathcal{H}}g_t^{(i)}$, for each $i \in \mathcal{H}$, $g_t^{(i)} = \nabla\ell(\theta_t; z_t^{(i, J_t^{(i)})})$ with $J_t^{(i)}$ sampled uniformly from $\{1, \ldots, m\}$, and for each $i \notin \mathcal{H}$, $g_t^{(i)}$ is a gradient computed on poisoned data. In fact, at each iteration, a single sample is selected uniformly at random from each worker. Consequently, with probability $1 - \frac{1}{m}$,

$$\delta_{t+1} \le \|G_{\gamma_t|S_t^* \cap S_t'^*}^{\mathrm{SGD}}(\theta_t) - G_{\gamma_t|S_t^* \cap S_t'^*}^{\mathrm{SGD}}(\theta_t')\|_2 + \frac{\gamma_t}{n - f}\|\sum_{i \in S_t^* \setminus S_t'^*}g_t^{(i)} - \sum_{i \in S_t'^* \setminus S_t^*}g_t'^{(i)}\|_2$$

$$\le \|G_{\gamma_t|S_t^* \cap S_t'^*}^{\mathrm{SGD}}(\theta_t) - G_{\gamma_t|S_t^* \cap S_t'^*}^{\mathrm{SGD}}(\theta_t')\|_2 + 2\gamma_t C\frac{f}{n - f},$$

and with probability $\frac{1}{m}$,

$$\delta_{t+1} \le \|G_{\gamma_t|S_t^* \cap S_t'^*}^{\mathrm{SGD}}(\theta_t) - G_{\gamma_t|S_t^* \cap S_t'^*}^{\mathrm{SGD}'}(\theta_t')\|_2 + \frac{\gamma_t}{n - f}\|\sum_{i \in S_t^* \setminus S_t'^*}g_t^{(i)} - \sum_{i \in S_t'^* \setminus S_t^*}g_t'^{(i)}\|_2$$

$$\le \|G_{\gamma_t|S_t^* \cap S_t'^*}^{\mathrm{SGD}}(\theta_t) - G_{\gamma_t|S_t^* \cap S_t'^*}^{\mathrm{SGD}}(\theta_t')\|_2 + \frac{\gamma_t}{n - f}\|\nabla\ell(\theta_t, z^{(a,b)}) - \nabla\ell(\theta_t', z_t'^{(a,b)})\|_2$$

$$+ \frac{\gamma_t}{n - f}\|\sum_{i \in S_t^* \setminus S_t'^*}g_t^{(i)} - \sum_{i \in S_t'^* \setminus S_t^*}g_t'^{(i)}\|_2$$

$$\leq \quad \|G^{\text{SGD}}_{\gamma_t|_{S^*_t \cap S'^*_t}}(\theta_t) - G^{\text{SGD}}_{\gamma_t|_{S^*_t \cap S'^*_t}}(\theta'_t)\|_2 + 2\gamma_t C \frac{f+1}{n-f},$$

where the second inequality corresponds to the worst-case scenario, namely when the index of the worker with the differing data samples lies in the intersection of the two selected subsets of workers, i.e., $a \in S^*_t \cap S'^*_t$. Hence, with similar derivation as in the GD case, we obtain the following recursion

$$\mathbb{E}_{\mathcal{A}} \delta_{t+1} \leq \eta_{G^{\text{SGD}}_{\gamma_t}|_{S^*_t \cap S'^*_t}} \mathbb{E}_{\mathcal{A}} \delta_t + 2\gamma_t \frac{f+\frac{1}{m}}{n-f} C, \tag{22}$$

where $\eta_{G^{\text{SGD}}_{\gamma_t}|_{S^*_t \cap S'^*_t}}$ is explicited in (21), that is $\eta_{G^{\text{SGD}}_{\gamma_t}|_{S^*_t \cap S'^*_t}} \leq \eta_{G^{\text{SGD}}_{\gamma_t}}$ if $\eta_{G^{\text{SGD}}_{\gamma_t}} \geq 1$; or $\eta_{G^{\text{SGD}}_{\gamma_t}|_{S^*_t \cap S'^*_t}} \leq 1 + \frac{n-2f}{n-f}(\eta_{G^{\text{SGD}}_{\gamma_t}} - 1)$ if $\eta_{G^{\text{SGD}}_{\gamma_t}} < 1$. We finish the proof as above, the depending on the regularity of the underlying loss function. $\qquad\square$

*Remark* D.1. Our bounds does not depend on the robustness coefficient $\kappa$ of the SMEA robust aggregation, but solely on its structural characteristic—namely, the sub-sampling of worker vectors. Additionally, our proof does not exploit the randomness in the selection of workers at each step, as this would require assumptions regarding the local distributions of the misbehaving workers to evaluate the probability of selecting the differing data samples' gradients. Interestingly, in practice, one could observe improved stability compared to the setting without misbehaving workers, possibly due to an extended "burn-in period". That is, if the differing samples are not selected by the aggregation rule when first drawn. This phenomenon could be further explored through numerical experiments.

*Remark* D.2. We can actually achieve the same theoretical bound as the non-misbehaving workers framework without the robust aggregation rule. Specifically, by reverting to the classical non-robust mean aggregation, we are able to handle terms with the same proof techniques from Hardt et al. (2016), since data heterogeneity does not manifest in the worst-case uniform algorithmic stability analysis. However, it is important to note that while this approach simplifies the analysis, the population risk may still remain unbounded due to the inherently unbounded optimization error that can arise from mean aggregation when facing an arbitrarily poisoned dataset.

## D.2. Proof of Theorem 3.4 for GD

The following lemma plays a key role in understanding the SMEA aggregation rule, which is essential for the subsequent lower-bound constructions.

**Lemma D.3.** *Let $n, f \in \mathbb{N}^*$, $f < \frac{n}{2}$ and $g_A, g_B, g_C$ three collinear vectors in $\mathbb{R}^d, d \geq 1$. Assume we observe a pool of $n$ vectors, consisting of $\alpha$ copies of $g_A$, $\beta$ copies of $g_B$ and $\gamma$ copies of $g_C$, where $\alpha, \beta, \gamma \in \{0, \ldots, n\}$, $\alpha + \beta + \gamma = n$. We abstract each subset $S \subseteq \{1, \ldots, n\}$, composed of $|S| = n - f$ vectors from our pool of vectors as $S_{a,b,c}$ with $a \in \{0, \ldots, \alpha\}, b \in \{0, \ldots, \beta\}, c \in \{0, \ldots, \gamma\}$, such that $a + b + c = n - f$ and $S$ is composed by $a$ copies of $g_A$, $b$ copies of $g_B$ and $c$ copies of $g_C$. With this notation, we have*

$$S^* \in \underset{\substack{S \subset \{1, \ldots, n\}, \\ |S| = n - f}}{argmin} \lambda_{\max}\left(\frac{1}{|S|} \sum_{i \in S}(g_i - \overline{g}_S)(g_i - \overline{g}_S)^{\mathsf{T}}\right)$$

$$= \frac{1}{(n-f)^2} \underset{\substack{S_{a,b,c} \\ a \in \{0, \ldots, \alpha\}, b \in \{0, \ldots, \beta\}, \\ c \in \{0, \ldots, \gamma\}, a+b+c=n-f}}{argmin} \{ab\|g_A - g_B\|^2_2 + bc\|g_B - g_C\|^2_2 + ac\|g_A - g_C\|^2_2\}.$$

*Let $u$ be a unit vector indicating the line the vectors are colinear to. Since all computations are effectively confined to this one-dimensional subspace, we could equivalently emphasize that*

$$\lambda_{\max}\left(\frac{1}{|S|} \sum_{i \in S}(g_i - \overline{g}_S)(g_i - \overline{g}_S)^{\mathsf{T}}\right) = \frac{1}{|S|} \sum_{i \in S}(g_i^{\mathsf{T}} u - \overline{g}_S^{\mathsf{T}} u)^2 \lambda_{\max}(uu^{\mathsf{T}}) = Var\left(\{g_i^{\mathsf{T}} u\}_{i \in S}\right).$$

*Proof.* Let $S_{a,b,c}$ with $a \in \{0, \ldots, \alpha\}, b \in \{0, \ldots, \beta\}, c \in \{0, \ldots, \gamma\}$, such that $a + b + c = n - f$ and $S$ is composed by $a$ copies of $g_A$, $b$ copies of $g_B$ and c copies of $g_C$. Then $\overline{g}_S = \frac{1}{n-f}(ag_A + bg_B + cg_C)$, $g_A - \overline{g}_S = \frac{b}{n-f}(g_A - g_B) + \frac{c}{n-f}(g_A - g_C)$, $g_B - \overline{g}_S = \frac{a}{n-f}(g_B - g_A) + \frac{c}{n-f}(g_B - g_C)$ and $g_C - \overline{g}_S = \frac{a}{n-f}(g_C - g_A) + \frac{b}{n-f}(g_C - g_B)$.

We already note that the outer product of vectors $g_A$, $g_B$ and $g_C$ commute because they are collinear. We denote $(*) = (n-f)^2 \sum_{i \in S_{a,b,c}} (g_i - \overline{g}_S)(g_i - \overline{g}_S)^\intercal$, by expending and re-arranging the terms, we obtain

$$
\begin{aligned}
(*) = {} & a \times [b^2(g_A - g_B)(g_A - g_B)^\intercal + c^2(g_A - g_C)(g_A - g_C)^\intercal + 2bc(g_A - g_B)(g_A - g_C)^\intercal] \\
& + b \times [a^2(g_B - g_A)(g_B - g_A)^\intercal + c^2(g_B - g_C)(g_B - g_C)^\intercal + 2ac(g_B - g_A)(g_B - g_C)^\intercal] \\
& + c \times [a^2(g_C - g_A)(g_C - g_A)^\intercal + b^2(g_C - g_B)(g_C - g_B)^\intercal + 2ab(g_C - g_A)(g_C - g_B)^\intercal] \\
= {} & (n-f)ab(g_A - g_B)(g_A - g_B)^\intercal - abc(g_A - g_B)(g_A - g_B)^\intercal \\
& + (n-f)ac(g_A - g_C)(g_A - g_C)^\intercal - abc(g_A - g_C)(g_A - g_C)^\intercal \\
& + (n-f)bc(g_B - g_C)(g_B - g_C)^\intercal - abc(g_B - g_C)(g_B - g_C)^\intercal \\
& + 2abc\left[(g_A - g_B)(g_A - g_C)^\intercal + (g_B - g_A)(g_B - g_C)^\intercal + (g_C - g_A)(g_C - g_B)^\intercal\right].
\end{aligned}
$$

We then re-arrange the terms and simplify the result,

$$
\begin{aligned}
(*) = {} & (n-f)ab(g_A - g_B)(g_A - g_B)^\intercal - abc(g_A - g_B)(g_A - g_B)^\intercal \\
& + (n-f)ac(g_A - g_C)(g_A - g_C)^\intercal - abc(g_A - g_C)(g_A - g_C)^\intercal \\
& + (n-f)bc(g_B - g_C)(g_B - g_C)^\intercal - abc(g_B - g_C)(g_B - g_C)^\intercal \\
& + abc\left[(g_A - g_B)(g_A - g_C)^\intercal - (g_A - g_B)(g_B - g_C)^\intercal\right] \\
& + abc\left[(g_A - g_B)(g_A - g_C)^\intercal - (g_A - g_C)(g_C - g_B)^\intercal\right] \\
& + abc\left[(g_B - g_A)(g_B - g_C)^\intercal - (g_C - g_A)(g_B - g_C)^\intercal\right] \\
= {} & (n-f)\left[ab(g_A - g_B)(g_A - g_B)^\intercal + ac(g_A - g_C)(g_A - g_C)^\intercal + bc(g_B - g_C)(g_B - g_C)^\intercal\right].
\end{aligned}
$$

To conclude the proof, we leverage the fact that $g_A, g_B, g_C$ are collinear vectors. Let $u$ be a unit vector representing the line the vectors are colinear to. Since each $g \in \{g_A, g_B, g_C\}$ lies along the line spanned by $u$, we can factor out $u$ and use the fact that $\lambda_{\max}(uu^\intercal) = 1$,

$$
\lambda_{\max}\left(\frac{1}{n-f} \sum_{i \in S_{a,b,c}} (g_i - \overline{g}_S)(g_i - \overline{g}_S)^\intercal\right) = \frac{1}{(n-f)^2}\left(ab\|g_A - g_B\|_2^2 + bc\|g_B - g_C\|_2^2 + ac\|g_A - g_C\|_2^2\right).
$$

$\square$

### D.2.1. CONVEX CASE

**Theorem D.4.** *Consider the setting described in Section 2 under data poisoning. Let $\mathcal{A} = $ GD, with the SMEA. Suppose $\mathcal{A}$ is run for $T \in \mathbb{N}^*$ iterations with $\gamma \leq \frac{2}{L}$. Then there exist $\ell \in \mathbb{R}^{\Theta \times \mathcal{Z}}$ such that $\forall z \in \mathcal{Z}, \ell(\cdot; z)$ is C-Lipschitz, L-smooth and convex, and a pair of neighboring datasets such that the uniform stability is lower bounded by*

$$
\Omega\left(\gamma C^2 T\left(\frac{f}{n-f} + \frac{1}{(n-f)m}\right)\right).
$$

*Proof.* In what follows, we provide the proof for the case $d = 1$. The result extends naturally to higher dimensions by embedding the one-dimensional construction into $\mathbb{R}^d$. An explicit construction for $d = 2$ is presented in the lower-bound proof for SGD, with SMEA, and convex loss function. We arbitrarily refer to one trajectory as *unperturbed*, with parameter $\theta_t$, and the other as *perturbed*, with $\theta_t'$. The proof is structured in the following three parts.

*(i)* First, we specify the loss function and the pair of neighboring datasets used. We then describe the structure of our problem, along with its regularity. Broadly speaking, the setup involves three distinct subgroups of workers. The first subgroup acts as a neutral or pivot subgroup that contains the differing samples, while the other two subgroups compete to be selected by the aggregation rule alongside the neutral subgroup.

*(ii)* We then show that, within our setup, it is possible to favor one subgroup through initialization and the other through perturbation introduced by the differing sample. This property holds inductively throughout the optimization process, ensuring that the unperturbed and perturbed parameters converge to distinct minimizers. As a result, the two trajectories remain divergent at each iteration, introducing an additional expansive term relative to the behavior observed under standard averaging rule.

*(iii)* Finally, we leverage this construction to derive a lower bound on uniform stability, which matches the upper bound established in Appendix D.1.

*(i)* Let $C, L \in \mathbb{R}_+^*$, we define the following loss function,

$$\text{For } \theta \in \mathbb{R}, \ z \in [-C, C], \quad \ell(\theta; z) = z\theta, \quad \ell'(\theta; z) = z, \quad \ell''(\theta; z) = 0.$$

By construction, the loss function is convex, $C$-Lipschitz and $L$-smooth. Let $m, n, f \in \mathbb{N}^*$ such that $f < \frac{n}{2}$. We define

$$0 < \psi := \sqrt{\frac{n - 2f - \frac{2}{m}}{n - 2f + \frac{2}{m}}} < 1,$$

and the following pair of neighboring datasets:

$$
\begin{aligned}
S &= \{D_1, \ldots, D_n\}, \quad S' = S \setminus \{D_1\} \cup \{D_1 \setminus \{z^{(1,1)}\} \cup \{z'^{(1,1)}\}\} \\
&\text{with } \forall i \in \{1, \ldots, n\}, \quad D_i = \{z^{(i,1)}, \ldots, z^{(i,m)}\} \\
&\text{and } \forall j \in \{1, \ldots, m\}, \quad i = 1: z^{(1,j)} = 0, \quad z^{(1,1)} = -C, \text{ and } z'^{(1,1)} = 0. \\
&\quad i \in N = \{2, \ldots, n - 2f\}, |N| = n - 2f - 1: z^{(i,j)} = 0. \\
&\quad i \in E = \{n - 2f + 1, \ldots, n - f\}, |E| = f: z^{(i,j)} = \frac{1 + \psi}{2}C. \\
&\quad i \in F = \{n - f + 1, \ldots, n\}, |F| = f: z^{(i,j)} = -C.
\end{aligned}
$$

Building on this setup, we first characterize the form of the parameters and gradients throughout the training process. We then specify the conditions under which our lower bound holds. Let $\gamma \leq \frac{2}{L}$ the learning rate and assume $\theta_0 = \theta'_0 = 0$. For $t \in \{0, \ldots, T - 1\}, g_t^{(N)} = g_t^{(N)'} = g_t^{(1)'} = 0, g_t^{(F)} = g_t^{(F)'} = -C, g_t^{(E)} = g_t^{(E)'} = \frac{1+\psi}{2}C$, and $g_t^{(1)} = -\frac{C}{m}$.

*(ii)* In what follows, we denote, for a subset $S \subset \{1, \ldots, n\}$ of size $n - f$, and for all $t \in \{0, \ldots, T - 1\}$,

$$\lambda_{\max}^S := \lambda_{\max}\left(\frac{1}{|S|}\sum_{i \in S}(g_t^{(i)} - \bar{g}_S)(g_t^{(i)} - \bar{g}_S)^\intercal\right) = \lambda_{\max}\left(\frac{1}{|S|}\sum_{i \in S}(g_0^{(i)} - \bar{g}_S)(g_0^{(i)} - \bar{g}_S)^\intercal\right)$$

with $\bar{g}_S = \frac{1}{|S|}\sum_{i \in S} g_0^{(i)}$. Recall that SMEA performs averaging over the subset of gradients selected according to the following rule:

$$S_t^* \in \operatorname*{argmin}_{S \subseteq \{1, \ldots, n\}, |S| = n - f} \lambda_{\max}\left(\frac{1}{|S|}\sum_{i \in S}\left(g_t^{(i)} - \bar{g}_S\right)\left(g_t^{(i)} - \bar{g}_S\right)^\intercal\right).$$

We highlight that the gradients are independent of the parameters and remain constant throughout the optimization process, as they are equal to the mean of their local samples. With a slight abuse of notation, we denote, for instance, the set $\{1\} \cup N \cup E$ by $\{1, N, E\}$. To uncover the source of instability, we consider the following two key conditions.

*(C1)* By Lemma D.3, for all $t \in \{0, \ldots, T - 1\}, S_t^{*'} = \{1, N, E\}$ is equivalent to

$$\lambda_{\max}'^{\{1,N,F\}} = \frac{f(n - 2f)}{(n - f)^2}C^2 > \frac{f(n - 2f)}{(n - f)^2}\left(\frac{1 + \psi}{2}\right)^2 C^2 = \lambda_{\max}'^{\{1,N,E\}},$$

which is ensured by construction. In fact, we again recall that the gradients are independent of the parameters and remain constant throughout the optimization process. Next, we demonstrate that any subset composed of a mixture of gradients from $1, N, F$, and $E$ necessarily yields a larger maximum eigenvalue. This is due to the fact that the vectors $g_0^{(F)'}$ and $g_0^{(E)'}$ are collinear but oriented in opposite directions, thereby contributing additively to the overall magnitude. To make this precise, we consider the following two cases.

(1) Let denote $S$ the set were we remove $q < f$ gradients of the subgroup $E$ and add $q$ gradients of the subgroup $F$ from the set $\{1, N, E\}$, we end up comparing the previous quantity to the following, by Lemma D.3,

$$\lambda_{\max}^{\prime S} = \frac{(f-q)(n-2f)}{(n-f)^2}C^2\|g_0^{(E)\prime} - g_0^{(N)\prime}\|_2^2 + \frac{q(n-2f)}{(n-f)^2}\|g_0^{(F)\prime} - g_0^{(N)\prime}\|_2^2$$

$$+ \frac{q(f-q)}{(n-f)^2}\|g_0^{(F)\prime} - g_0^{(E)\prime}\|_2^2$$

$$= \frac{(f-q)(n-2f)}{(n-f)^2}C^2\left(\frac{1+\psi}{2}\right)^2 + \frac{q(n-2f)}{(n-f)^2}C^2 + q(f-q)C^2\left(\frac{3+\psi}{2}\right)^2$$

$$= \lambda_{\max}^{\prime\{1,N,E\}} + \frac{q(n-2f)}{(n-f)^2}C^2\left(1 - \left(\frac{1+\psi}{2}\right)^2\right) + \frac{q(f-q)}{(n-f)^2}C^2\left(\frac{3+\psi}{2}\right)^2$$

$$> \lambda_{\max}^{\prime\{1,N,E\}}.$$

(2) Similarly, let denote $S$ the set were we remove $q < \min(f, n-2f)$ gradients of the subgroup $\{1, N\}$ and add $q$ gradients of the subgroup $F$ from the set $\{1, N, E\}$, we end up comparing the previous quantity to the following, by Lemma D.3,

$$\lambda_{\max}^{\prime S} = \frac{f(n-2f-q)}{(n-f)^2}C^2\|g_0^{(E)\prime} - g_0^{(N)\prime}\|_2^2 + \frac{q(n-2f-q)}{(n-f)^2}\|g_0^{(F)\prime} - g_0^{(N)\prime}\|_2^2$$

$$+ \frac{qf}{(n-f)^2}\|g_0^{(F)\prime} - g_0^{(E)\prime}\|_2^2$$

$$= \lambda_{\max}^{\prime\{1,N,E\}} + \frac{q(n-2f-q)}{(n-f)^2}C^2 + \frac{qf}{(n-f)^2}C^2\left(\left(\frac{3+\psi}{2}\right)^2 - \left(\frac{1+\psi}{2}\right)^2\right)$$

$$> \lambda_{\max}^{\prime\{1,N,E\}}.$$

*(C2)* As shown in $(C1)$, the core tension in the choice of the SMEA aggregation rule ultimately reduces to a comparison between the sets $\{1, N, E\}$ and $\{1, N, F\}$, a consideration that also applies for $(C2)$. By Lemma D.3, for all $t \in \{0, \ldots, T-1\}$,

$$S_t^* = \{1, N, F\} \iff \left(n - 2f - \frac{2}{m}\right)|g_t^{(F)}|^2 < (n-2f)|g_t^{(E)}|^2 + \frac{2}{m}|g_t^{(F)}||g_t^{(E)}|.$$

As $|g_t^{(E)}| < |g_t^{(F)}|$, the above inequality holds if

$$\left(n - 2f - \frac{2}{m}\right)|g_t^{(F)}|^2 < (n - 2f + \frac{2}{m})|g_t^{(E)}|^2,$$

which holds by construction as

$$\psi|g_t^{(F)}| = \psi C < \frac{1+\psi}{2}C = |g_t^{(E)}|.$$

Hence, by construction SMEA always selects $\{1, N, F\}$ for the unperturbed run, and $\{1, N, E\}$ for the perturbed one. Let denote $p = \frac{f+1/m}{n-f}$, as $\theta_0 = \theta_0' = 0$, we then have by the $(C1)$ and $(C2)$ conditions,

$$\theta_T' = -\gamma\sum_{t=0}^{T-1}\frac{1}{n-f}\sum_{i\in\{1,N,E\}}g_t^{(i)\prime} = -\frac{f}{n-f}\gamma\frac{1+\psi}{2}\gamma CT,$$

and

$$\theta_T = -\gamma\sum_{t=0}^{T-1}\frac{1}{n-f}\sum_{i\in\{1,N,F\}}g_t^{(i)} = p\gamma CT.$$

***(iii)*** Since $\ell(\theta_T'; z) \leq 0$, this directly yields the following lower bound on uniform stability,

$$\sup_{z \in [-C, C]} |\ell(\theta_T; z) - \ell(\theta_T'; z)| \geq \ell(\theta_T; C) = C\theta_T = p\gamma C^2 T. \qquad \square$$

### D.2.2. STRONGLY CONVEX CASE

The proof structure with strongly convex objectives closely parallels that of Theorem D.4. We refer to it where appropriate to avoid unnecessary repetition, while explicitly highlighting the key differences.

**Theorem D.5.** *Consider the setting described in Section 2 under data poisoning. Let $\mathcal{A} = \text{GD}$, with the SMEA. Suppose $\mathcal{A}$ is run for $T \in \mathbb{N}^*$ iterations, $T \geq \frac{\ln(1-c)}{\ln(1-\gamma\mu)}$, $0 < c < 1$, with $\gamma \leq \frac{1}{L}$. Then there exist $\ell \in \mathbb{R}^{\Theta \times \mathcal{Z}}$ such that $\forall z \in \mathcal{Z}, \ell(\cdot; z)$ is $C$-Lipschitz, $L$-smooth and $\mu$-stongly convex, and a pair of neighboring datasets such that the uniform stability is lower bounded by*

$$\Omega\left(\frac{C^2}{\mu}\left(\frac{f}{n-f} + \frac{1}{(n-f)m}\right)\right).$$

*Proof.* In what follows, we provide the proof for the case $d = 1$. The result extends naturally to higher dimensions by embedding the one-dimensional construction into $\mathbb{R}^d$. We arbitrarily refer to one trajectory as *unperturbed*, with parameter $\theta_t$, and the other as *perturbed*, with $\theta_t'$. We adopt a reasoning analogous to that used in the lower bound proof in Theorem D.4. As explained in Theorem D.4, we structure our proof into three paragraphs *(i)*, *(ii)* and *(iii)* below.

***(i)*** Let $C, L, \mu \in \mathbb{R}_+^*$, with $\mu \leq L$, we define the following loss function

$$\forall \theta \in \Theta = B(0, \frac{C}{2\mu}), z \in \mathcal{Z} = B(0, \frac{C}{2\mu}),$$

$$\ell(\theta; z) = \frac{\mu}{2}(\theta - z)^2, \quad \ell'(\theta; z) = \mu(\theta - z), \quad \ell''(\theta; z) = \mu.$$

By construction, the loss function is $\mu$-convex, $C$-Lipschitz and $L$-smooth. Let $m, n, f \in \mathbb{N}^*$ such that $f < \frac{n}{2}$. We define,

$$\max\left\{\sqrt{\frac{n - 2(f + \frac{1}{m})}{n - 2(f - \frac{1}{m})}}, 1 - \frac{4}{m(n - 2f)}\right\} < \psi < 1,$$

and the following pair of neighboring datasets:

$$S = \{D_1, \ldots, D_n\}, \quad S' = S \setminus \{D_1\} \cup \{D_1 \setminus \{(x^{(1,1)}, y^{(1,1)})\} \cup \{(x'^{(1,1)}, y'^{(1,1)})\}\}$$
$$\text{with } \forall i \in \{1, \ldots, n\}, \quad D_i = \{(x^{(i,1)}, y^{(i,1)}), \ldots, (x^{(i,m)}, y^{(i,m)})\}$$
$$\text{and } \forall j \in \{1, \ldots, m\}, \quad i = 1: z^{(i,j)} = 0, \quad z^{(1,1)} = \frac{C}{2\mu}, \quad z'^{(1,1)} = 0.$$
$$i \in N = \{2, \ldots, n - 2f\}, |N| = n - 2f - 1: z^{(i,j)} = 0.$$
$$i \in E = \{n - 2f + 1, \ldots, n - f\}, |E| = f: z^{(i,j)} = -\frac{1 + \psi}{2}\frac{C}{2\mu}.$$
$$i \in F = \{n - f + 1, \ldots, n\}, |F| = f: z^{(i,j)} = \frac{C}{2\mu}.$$

Let $\gamma \leq \frac{1}{L}$ the learning rate. Following this setting, we first state the form that take the parameters and the gradients during the training process, then the conditions under which we claim our instability: for $t \in \{0, \ldots, T - 1\}, \theta_0 = \theta_0' = 0$, $g_t^{(N)} = \mu\theta_t'$ $g_t'^{(N)} = g_t'^{(1)} = \mu\theta_t', g_t^{(1)} = (1 - \frac{1}{m})g_t^{(N)} + g_t^{(F)}/m, g_t^{(E)} = \mu\left(\theta_t + \frac{1+\psi}{2}\frac{C}{2\mu}\right), g_t^{(E)'} = \mu\left(\theta_t' + \frac{1+\psi}{2}\frac{C}{2\mu}\right)$, $g_t^{(F)} = \mu\left(\theta_t - \frac{C}{2\mu}\right)$ and $g_t^{(F)'} = \mu\left(\theta_t' - \frac{C}{2\mu}\right)$.

*(ii)* In what follows, we denote, for a subset $S \subset \{1, \ldots, n\}$ of size $n - f$. Recall that SMEA performs averaging over the subset of gradients selected according to the following rule, for all $t \in \{0, \ldots, T - 1\}$,

$$S_t^* \in \underset{S \subseteq \{1, \ldots, n\}, |S| = n - f}{\operatorname{argmin}} \lambda_{\max}\left( \frac{1}{|S|} \sum_{i \in S} \left(g_t^{(i)} - \bar{g}_S\right)\left(g_t^{(i)} - \bar{g}_S\right)^{\mathsf{T}} \right).$$

With a slight abuse of notation, we denote, for instance, the set $\{1\} \cup N \cup E$ by $\{1, N, E\}$. To uncover the source of instability, we consider the following two key conditions.

*(C1)* As shown in the $(C1)$ condition of Theorem D.4, the core tension in the choice of the SMEA aggregation rule ultimately reduces to a comparison between the sets $\{1, N, E\}$ and $\{1, N, F\}$, a consideration that also applies here. By Lemma D.3,

$$S_0^{*\prime} = \{1, N, E\} \underset{\text{SMEA}}{\Longleftrightarrow} |g_0^{(F)\prime}|_2 = \mu \frac{C}{2\mu} > \mu \frac{1 + \psi}{2} \frac{C}{2\mu} = |g_0^{(E)\prime}|_2.$$

Let denote

$$\lambda_{\max,t}^{\{1,N,E\}\prime} := \lambda_{\max}\left( \frac{1}{n - f} \sum_{i \in \{1,N,E\}} (g_t^{(i)\prime} - \bar{g}_S')(g_t^{(i)\prime} - \bar{g}_S')^{\mathsf{T}} \right),$$

and

$$\lambda_{\max,t}^{\{1,N,F\}\prime} := \lambda_{\max}\left( \frac{1}{n - f} \sum_{i \in \{1,N,F\}} (g_t^{(i)\prime} - \bar{g}_S')(g_t^{(i)\prime} - \bar{g}_S')^{\mathsf{T}} \right).$$

We then prove that for all $t \geq 0$, $S_t^{*\prime} = \{1, N, E\}$, and the unperturbed run converges to the minimizer of the subgroup $E$. For this property to be true, we require

$$\lambda_{\max,t}^{\{1,N,E\}\prime} < \lambda_{\max,t}^{\{1,N,F\}\prime}.$$

Again, by Lemma D.3, it is equivalent to

$$\left(\frac{1 + \psi}{2}\right)^2 \frac{C^2}{4} < \frac{C^2}{4},$$

which holds by construction. As the condition is independent of $t$, hence true for all $t$, this conclude the first condition.

*(C2)* By Lemma D.3, $S_0^* = \{1, N, F\} \underset{\text{SMEA and } (C1)}{\Longleftarrow} \psi|g_0^{(F)}| = \psi\mu\frac{C}{2\mu} < \mu\frac{1+\psi}{2}\frac{C}{2\mu} = |g_0^{(E)}|$. We then prove that for all $t \geq 0$, $S_t^* = \{1, N, F\}$, and the perturbed run converges to the minimizer of the subgroup $F$. For this property to be true, we require:

$$\lambda_{\max,t}^{\{1,N,F\}} < \lambda_{\max,t}^{\{1,N,E\}}.$$

Again, by Lemma D.3, it is equivalent to

$$f(n - 2f - 1)\frac{C^2}{4} + f\frac{C^2}{4}\left(1 - \frac{1}{m}\right)^2 + (n - 2f - 1)\frac{C^2}{4m^2}$$
$$< f(n - 2f - 1)\left(\frac{1 + \psi}{2}\right)^2 \frac{C^2}{4} + f\frac{C^2}{4}\left(\frac{1 + \psi}{2} + \frac{1}{m}\right)^2 + (n - 2f - 1)\frac{C^2}{4m^2}.$$

That is,

$$0 < -(n - 2f - 1)\left(\frac{1 - \psi}{2}\right)\left(\frac{3 + \psi}{2}\right) + \left(\frac{3 + \psi}{2}\right)\left(\frac{2}{m} - \frac{1 - \psi}{2}\right),$$

which is equivalent to

$$\psi > \frac{4}{(n - f)m}.$$

As the condition is independent of $t$, hence true for all $t$, this conclude the second condition.

Let denote $p = \frac{f+1/m}{n-f}$. We have proven that, for $t \geq 0$,

$$\text{SMEA}(g_t^{(1)}, \ldots, g_t^{(n)}) = \frac{1}{n-f} \sum_{i \in \{1, N, F\}} g_t^{(i)} = \mu\theta_t - p\frac{C}{2},$$

$$\text{SMEA}(g_t^{(1)\prime}, \ldots, g_t^{(n)\prime}) = \frac{1}{n-f} \sum_{i \in \{1, N, E\}} g_t^{(i)\prime} = \mu\theta_t' + \frac{f}{n-f}\frac{1+\psi}{2}\frac{C}{2}.$$

For $t \geq 0$, $\theta_{t+1} = (1 - \gamma\mu)\theta_t + \frac{\gamma pC}{2}$ and $\theta_{t+1}' = (1 - \gamma)\theta_t' - \frac{\gamma f(1+\psi)C}{4(n-f)}$. Hence, $\theta_T' \leq 0$ and,

$$\theta_T = \frac{pC}{2\mu}\left(1 - (1 - \gamma\mu)^T\right)$$

***(iii)*** Provided that there exist $0 < c < 1$ such that $T \geq \frac{\ln(1-c)}{\ln(1-\gamma\mu)}$, this implies $|\theta_T - \theta_T'| \geq |\theta_T| \in \Omega(\frac{pC}{\mu})$. Finally, let define $\chi = -\text{sign}(\theta_T + \theta_T')$, we have

$$
\begin{aligned}
|\ell(\theta_T; \chi\frac{C}{2\mu}) - \ell(\theta_T'; \chi\frac{C}{2\mu})| &= \frac{\mu}{2}|\left(\theta_T - \chi\frac{C}{2\mu}\right)^2 - \left(\theta_T' - \chi\frac{C}{2\mu}\right)^2| \\
&= \frac{\mu}{2}|\theta_T - \theta_T'||\theta_T + \theta_T' + \text{sign}(\theta_T + \theta_T')\frac{C}{\mu}| \\
&\geq \frac{\mu}{2}|\theta_T|\frac{C}{\mu} \in \Omega(\frac{pC^2}{\mu}). \qquad\qquad \square
\end{aligned}
$$

### D.3. Proof of Theorem 3.4 for SGD

In this section, we derive a lower bound on the uniform stability of distributed $\mathrm{projected-GD}$ with SMEA under data poisoning. The proof structure closely parallels that of Theorem D.4. We refer to it where appropriate to avoid unnecessary repetition, while explicitly highlighting the key differences.

**Theorem D.6.** *Consider the setting described in Section 2 under data poisoning. Let $\mathcal{A} = \mathrm{projected-SGD}$ with SMEA. Suppose $\mathcal{A}$ is run for $T \in \mathbb{N}^*$ iterations with $\gamma \leq \frac{1}{L}$. Assume there exist a constant $\tau \geq c > 0$ for an arbitrary c, such that $T \geq \tau m$, then there exist $\ell \in \mathbb{R}^{\Theta \times \mathcal{Z}}$ such that $\forall z \in \mathcal{Z}, \ell(\cdot; z)$ is C-Lipschitz, L-smooth and convex, and a pair of neighboring datasets such that the uniform stability is lower bounded by*

$$\Omega\left(\gamma C^2 T\left(\frac{f}{n-f} + \frac{1}{(n-f)m}\right)\right).$$

*Proof.* We adopt a reasoning analogous to that used in the lower bound proof in Theorem D.4. However, we can no longer rely on a linear loss function. The reason is that, in the linear case, the gradients are independent of the model parameters, so when the differing samples are not selected, the perturbed and unperturbed executions remain indistinguishable—thus precluding any divergence. We also have to account for the first occurrence of the differing samples, denoted as $T_0 = \inf\{t; J_{t-1}^{(1)} = 1\}$. In fact, we require to ensure the conditions $(C1)$ (that is $S_0^{*\prime} = \{1, N, E\}$) at initialization but we now would like to ensure $(C2)$ when the differing samples are drawn (that is $S_t^* = \{1, N, F\}$ for $t \geq t_0 - 1$, knowing $T_0 = t_0$ for any time $t_0 < T$). To do so, we modify the possible parameters we are searching through our optimization process, making the condition at initialization sufficient to ensure $(C1)$ and $(C2)$ for any $t_0 \in \{0, \ldots, T-1\}$. By defining $\Theta = \{\lambda v; 0 \leq \lambda < \infty\}$ and applying the projection $p_\Theta$ at each step to constrain the parameter within the set of admissible values, we ensure, under condition $(C1)$ that $\theta_t' = 0$ for all $t > 0$ and $\theta_t = 0$ for $t < t_0$. Consequently, the condition $(C2)$ is ensured by construction for any time the differing sample is drawn. While using a projection simplifies the lower bound proof (Appendix D.3), it does not affect the tightness of our result since the upper bound on uniform stability remains valid for $\mathrm{projected-SGD}$ as the projection does not increase the distance between projected points.

As explained in Theorem D.4, we structure our proof into three paragraphs *(i)*, *(ii)* and *(iii)* below.

*(i)* Let $C, L \in \mathbb{R}_+^*$, we define the following dataset and loss.

$$\forall \theta \in \Theta = \{\lambda v; \lambda \in \mathbb{R}\} \subset \mathbb{R}^2, (x, y) \in B(0, \sqrt{L}) \times \mathbb{R},$$

$$\ell(\theta; (x, y)) = \begin{cases} \frac{1}{2}\left(\theta^T x - y\right)^2 & \text{if } |\theta^T x - y|\|x\|_2 \le C \\ C\left(\frac{|\theta^T x - y|}{\|x\|_2} - \frac{C}{2\|x\|_2^2}\right) & \text{otherwise.} \end{cases}$$

$$\nabla \ell(\theta; (x, y)) = \begin{cases} (\theta^T x - y)x & \text{if } |\theta^T x - y|\|x\|_2 \le C \\ C \operatorname{sign}(\theta^T x - y)\frac{x}{\|x\|_2} & \text{otherwise.} \end{cases}$$

$$\nabla^2 \ell(\theta; (x, y)) = \begin{cases} 0 \preceq xx^T \preceq LI_d & \text{if } |\theta^T x - y|\|x\|_2 \le C \\ 0 & \text{otherwise.} \end{cases}$$

By construction, the loss function is convex, $C$-Lipschitz and $L$-smooth. Let $m, n, f \in \mathbb{N}^*$ such that $f < \frac{n}{2}$. We define,

$$0 < \psi = \frac{n - 2f - 2}{n - 2f} < 1,$$

$$\min\{1 - \psi, \gamma L \frac{f}{n - f}\} > \epsilon > 0$$

and the following pair of neighboring datasets:

$$v \in B(0, \sqrt{L}), \quad \|v\|_2^2 = L, \quad \beta = \frac{C}{\sqrt{L}}, \quad \alpha = (1 - \epsilon)\beta, \quad b = \frac{1}{\sqrt{T}}$$

$$S = \{D_1, \ldots, D_n\}, \quad S' = S \setminus \{D_1\} \cup \{D_1 \setminus \{(x^{(1,1)}, y^{(1,1)})\} \cup \{(x'^{(1,1)}, y'^{(1,1)})\}\}$$

$$\text{with } \forall i \in \{1, \ldots, n\}, \quad D_i = \{(x^{(i,1)}, y^{(i,1)}), \ldots, (x^{(i,m)}, y^{(i,m)})\}$$

$$\text{and } \forall j \in \{1, \ldots, m\}, \quad i = 1: (x^{(1,j)}, y^{(1,j)}) = (0, 0), \quad (x^{(1,1)}, y^{(1,1)}) = (bv, \frac{\beta}{b}),$$

$$\text{and } (x'^{(1,1)}, y'^{(1,1)}) = (0, 0).$$

$$i \in N = \{2, \ldots, n - 2f\}, |N| = n - 2f - 1: (x^{(i,j)}, y^{(i,j)}) = (0, 0).$$

$$i \in E = \{n - 2f + 1, \ldots, n - f\}, |E| = f: (x^{(i,j)}, y^{(i,j)}) = (v, -\alpha).$$

$$i \in F = \{n - f + 1, \ldots, n\}, |F| = f: (x^{(i,j)}, y^{(i,j)}) = (bv, \frac{\beta}{b}).$$

Let $\gamma \le \frac{1}{L}$ the learning rate. Following this setup, we first describe the form of the parameters and gradients throughout the training process, and then state the conditions under which we establish our instability result, assuming $\theta_0 = \theta_0' = 0$. For $t \in \{0, \ldots, T - 1\}$, and $\theta_t = \lambda_t v, \theta_t' = \lambda_t' v, \lambda_t, \lambda_t' \in [\lambda_E^* := -\frac{\alpha}{L}, \lambda_F^* := \frac{\beta}{b^2 L}]$,

$$g_t^{(N)} = 0, \qquad \|g_t^{(E)}\|_2 > \psi\|g_t^{(F)}\|_2, \qquad\qquad g_t^{(F)} = -\underbrace{\left(\beta - \lambda_t L b^2\right)}_{>0 \text{ and } < \frac{C}{\sqrt{L}}}v$$

$$g_t^{(N)\prime} = 0, \qquad g_t^{(E)\prime} = \underbrace{\left(\lambda_t' L + \alpha\right)}_{>0 \text{ and } < \frac{C}{\sqrt{L}}}v, \qquad\qquad \|g_t^{(F)\prime}\|_2 > \|g_t^{(E)\prime}\|_2.$$

*(ii)* In what follows, we denote, for a subset $S \subset \{1, \ldots, n\}$ of size $n - f$. Recall that SMEA performs averaging over the subset of gradients selected according to the following rule, for all $t \in \{0, \ldots, T - 1\}$,

$$S_t^* \in \operatorname*{argmin}_{S \subseteq \{1, \ldots, n\}, |S| = n - f} \lambda_{\max}\left(\frac{1}{|S|}\sum_{i \in S}\left(g_t^{(i)} - \bar{g}_S\right)\left(g_t^{(i)} - \bar{g}_S\right)^\top\right).$$

With a slight abuse of notation, we denote, for instance, the set $\{1\} \cup N \cup E$ by $\{1, N, E\}$. To uncover the source of instability, we consider the following two key conditions:

*(C1)* As shown in the $(C1)$ condition of Theorem D.4, the core tension in the choice of the SMEA aggregation rule ultimately reduces to a comparison between the sets $\{1, N, E\}$ and $\{1, N, F\}$, a consideration that also applies here. $S_0^{*'} = \{1, N, E\} \underset{\text{SMEA}}{\Longleftrightarrow} \|g_0^{(F)'}\|_2 = \beta\sqrt{L} > \alpha\sqrt{L} = \|g_0^{(E)'}\|_2$. Hence, $\theta_1' = p_\Theta(-\gamma\frac{f}{n-f}g_0^{(E)}) = p_\Theta(-\gamma\frac{f}{n-f}\alpha v) = 0$. We then prove by induction, that for all $t \geq 0$, $S_t^{*'} = \{1, N, E\}$ and the unperturbed run remains at 0. The perturbed run remains at 0 until the differing samples are drawn.

*(C2)* Let denote

$$\lambda_{\max,t}^{\{1,N,E\}} := \lambda_{\max}\left(\frac{1}{n-f}\sum_{i\in\{1,N,E\}}(g_t^{(i)} - \bar{g}_S)(g_t^{(i)} - \bar{g}_S)^\top\right),$$

and

$$\lambda_{\max,t}^{\{1,N,F\}} := \lambda_{\max}\left(\frac{1}{n-f}\sum_{i\in\{1,N,F\}}(g_t^{(i)} - \bar{g}_S)(g_t^{(i)} - \bar{g}_S)^\top\right).$$

Assuming $T_0 = t_0$ known, $S_{t_0-1}^* = \{1, N, F\} \underset{\text{SMEA and } (C1)}{\Longleftrightarrow} \frac{n-2(f+1)}{n-2}\|g_{t_0-1}^{(F)}\|_2 < \|g_{t_0-1}^{(E)}\|_2$. In fact, by Lemma D.3 with $g_A = g_{t_0-1}^{(E)}$, $g_B = g_{t_0-1}^{(F)}$ and $g_C = 0$,

$$S_{t_0-1}^* \in \operatorname{argmin}\left\{\lambda_{\max,t_0-1}^{\{1,N,F\}} = \frac{(f+1)(n-2f-1)}{(n-f)^2}\|g_{t_0-1}^{(F)}\|_2^2,\right.$$
$$\left.\lambda_{\max,t_0-1}^{\{1,N,E\}} = \frac{f(n-2f-1)}{(n-f)^2}\|g_{t_0-1}^{(E)}\|_2^2 + \frac{(n-2f-1)}{(n-f)^2}\|g_{t_0-1}^{(F)}\|_2^2 + \frac{f}{(n-f)^2}\left(\|g_{t_0-1}^{(E)}\|_2 + \|g_{t_0-1}^{(F)}\|_2\right)^2\right\}.$$

Hence,

$$S_{t_0-1}^* = \{1, N, F\} \Leftrightarrow (n-2f-1)\left(\|g_{t_0-1}^{(F)}\|_2^2 - \|g_{t_0-1}^{(E)}\|_2^2\right) < \left(\|g_{t_0-1}^{(F)}\|_2 + \|g_{t_0-1}^{(E)}\|_2\right)^2$$
$$\Leftrightarrow (n-2f-1)\left(\|g_{t_0-1}^{(F)}\|_2 - \|g_{t_0-1}^{(E)}\|_2\right) < \|g_{t_0-1}^{(F)}\|_2 + \|g_{t_0-1}^{(E)}\|_2$$
$$\Leftrightarrow \psi\|g_{t_0-1}^{(F)}\|_2 < \|g_{t_0-1}^{(E)}\|_2.$$

which holds by construction of our initialization, $\psi\|g_{t_0-1}^{(F)}\|_2 < \|g_{t_0-1}^{(E)}\|_2 < \|g_{t_0-1}^{(F)}\|_2$.

We then prove that for the next step, $S_{t_0}^* = \{1, N, F\}$ in any case. This is sufficient to prove by induction that for all $t \geq t_0$, $S_t^* = \{1, N, F\}$ and the perturbed run converges to the minimizer of the subgroup $F$. In fact, for $t \geq t_0$, $\|g_t^{(F)}\|_2$ decreases and $\|g_t^{(E)}\|_2$ increases.

(1) with probability $\frac{1}{m}$, the differing samples are again drawn. As we already ensured the conditions in the previous step, we have $\psi\|g_{t_0}^{(F)}\|_2 \leq \|g_{t_0-1}^{(F)}\|_2 < g_{t_0-1}^{(E)}\|_2 \leq \|g_{t_0}^{(E)}\|_2$, then $S_{t_0}^* = \{1, N, F\}$.

(2) with probability $1 - \frac{1}{m}$ we have to ensure $S_{t_0}^* = \{1, N, F\} \underset{\text{SMEA}}{\Longleftrightarrow} \|g_{t_0}^{(F)}\|_2 \leq \|g_{t_0}^{(E)}\|_2$ with $\theta_{t_0} = -\gamma\frac{f+1}{n-f}g_{t_0-1}^{(F)} = \gamma\beta\frac{f+1}{n-f}v$, hence

$$\|g_{t_0}^{(F)}\|_2 = -\beta\left(1 - \gamma\frac{f+1}{n-f}Lb^2\right)v,$$

$$\|g_{t_0}^{(E)}\|_2 = \max\{\beta, \beta\left(1 - \epsilon + \gamma L\frac{f+1}{n-f}\right)\}v = \beta v$$

We have proven that, for $t \geq t_0 - 1$, $S_{t_0-1}^* = \{1, N, F\}$. Thus,

$$\mathbb{E}[\theta_t | T_0 = t_0] = \begin{cases} 0 & \text{if } t \leq t_0 - 1 \\ \gamma\beta\frac{f+1}{n-f}v & \text{for } t = t_0 \\ \mathbb{E}[\theta_{t-1} | T_0 = t_0] - \gamma p\mathbb{E}[g_{t-1}^{(F)} | T_0 = t_0] & \text{otherwise.} \end{cases}$$

with $\theta_t = \lambda_t v$, $p = \frac{f + \frac{1}{m}}{n - f}$, the recurrence for $t > t_0$ is given by

$$\mathbb{E}[\lambda_t | T_0 = t_0] = \mathbb{E}[\lambda_{t-1} | T_0 = t_0] + p\gamma(\beta - \mathbb{E}[\lambda_{t-1} | T_0 = t_0] Lb^2)$$
$$= (1 - p\gamma Lb^2) \, \mathbb{E}[\lambda_{t-1} | T_0 = t_0] + p\gamma\beta.$$

Hence

$$\mathbb{E}[\theta_t | T_0 = t_0] = \begin{cases} 0 & \text{if } t \leq t_0 - 1 \\ \frac{f+1}{f + \frac{1}{m}} \frac{\beta}{b^2 L} \left(1 - (1 - p\gamma Lb^2)^{t - t_0 + 1}\right) v & \text{otherwise.} \end{cases}$$

Taking the total expectation, we obtain

$$\mathbb{E}[\theta_T] = \sum_{t_0 = 1}^{T} \mathbb{E}[\theta_T | T_0 = t_0] \mathbb{P}(T_0 = t_0)$$
$$= \frac{CT \frac{f+1}{f + \frac{1}{m}}}{Lm} \sum_{t_0 = 1}^{T} \left(1 - \left(1 - \frac{p\gamma L}{T}\right)^{T - t_0 + 1}\right) \left(1 - \frac{1}{m}\right)^{t_0 - 1} \frac{v}{\sqrt{L}}$$

*(iii)* In the remainder of the proof, we bound $\mathbb{E}[\theta_T]$. Specifically, we leverage its analytical expression to establish a concise lower bound on uniform argument stability.

$$\sum_{t_0 = 1}^{T} \left(1 - (1 - p\gamma Lb^2)^{T - t_0 + 1}\right) \left(1 - \frac{1}{m}\right)^{t_0 - 1} \geq \sum_{t_0 = 1}^{T} \left(1 - \frac{1}{1 + p\gamma Lb^2(T - t_0 + 1)}\right) \left(1 - \frac{1}{m}\right)^{t_0 - 1}$$
$$\geq \sum_{t_0 = 1}^{T} \frac{p\gamma L \frac{T - t_0 + 1}{T}}{1 + p\gamma L \frac{T - t_0 + 1}{T}} \left(1 - \frac{1}{m}\right)^{t_0 - 1}$$

Where we used that $(1 - p\gamma Lb^2)^{T - t_0 + 1} \leq \frac{1}{1 + p\gamma Lb^2(T - t_0 + 1)}$ for $T - t_0 + 1 \geq 0$ and $-1 < -p\gamma Lb^2 < 0$. In fact using $\ln(1 + x) \leq x$ for $x > -1$,

$$\ln\left[(1 - p\gamma Lb^2)^{T - t_0 + 1}(1 + p\gamma Lb^2(T - t_0 + 1))\right] = (T - t_0 + 1)\ln(1 - p\gamma Lb^2) + \ln(1 + p\gamma Lb^2(T - t_0 + 1))$$
$$\leq -p\gamma Lb^2(T - t_0 + 1) + p\gamma Lb^2(T - t_0 + 1) = 0.$$

Also, as $\frac{1}{1 + p\gamma L \frac{T - t_0 + 1}{T}} \geq \frac{1}{1 + \gamma Lp} \geq \frac{1}{1 + p} \geq \frac{1}{2}$, we have

$$\sum_{t_0 = 1}^{T} \left(1 - (1 - p\gamma Lb^2)^{T - t_0 + 1}\right) \left(1 - \frac{1}{m}\right)^{t_0 - 1}$$
$$\geq \frac{p\gamma L}{2} \sum_{t_0 = 1}^{T} \left(1 - \frac{t_0 - 1}{T}\right) \left(1 - \frac{1}{m}\right)^{t_0 - 1}$$
$$= \frac{p\gamma Lm}{2} \left(1 - \left(1 - \frac{1}{m}\right)^T\right) - \frac{p\gamma L}{2T} \sum_{t_0 = 1}^{T} (t_0 - 1)\left(1 - \frac{1}{m}\right)^{t_0 - 1}$$
$$= \frac{p\gamma Lm}{2} \left(1 - \left(1 - \frac{1}{m}\right)^T\right) - \frac{p\gamma L}{2T} \left[m(m - 1) - m^2\left(1 - \frac{1}{m}\right)^T\left(1 + \frac{T - 1}{m}\right)\right]$$
$$= \frac{p\gamma Lm}{2} \left[1 - \frac{m - 1}{T}\left(1 - \left(1 - \frac{1}{m}\right)^T\right)\right],$$

where we used the finite polylogarithmic sum formula in the second equality above.

If we additionally assume there exist a constant $\tau \geq c > 0$ for an arbitrary c, such that $T \geq \tau m$, we then can prove that $K(T,m) = \left[1 - \frac{m-1}{T}\left(1 - \left(1 - \frac{1}{m}\right)^T\right)\right] \in \Omega(1)$. In fact this is trivial for $m = 1$, then we prove that the quantity $K(T,m)$ increases with $T$, for $T \geq 1$ and $m \geq 2$. We prove by induction that $K(T+1) - K(T) = \frac{m-1}{T(T+1)}\left[1 - \left(1 - \frac{1}{m}\right)^T(1 + \frac{T}{m})\right] > 0$ which is equivalent to $\left(1 - \frac{1}{m}\right)^T(1 + \frac{T}{m}) < 1$. For $T = 1$, $(1 - \frac{1}{m})(1 + \frac{1}{m}) = 1 - \frac{1}{m^2} < 1$. For $T + 1 > 2$, $\frac{\left(1 - \frac{1}{m}\right)^{T+1}(1 + \frac{T+1}{m})}{\left(1 - \frac{1}{m}\right)^T(1 + \frac{T}{m})} = \left(1 - \frac{1}{m}\right)\frac{1 + \frac{T+1}{m}}{1 + \frac{T}{m}} < 1$. Hence

$$1 - \frac{m-1}{T}\left(1 - \left(1 - \frac{1}{m}\right)^T\right) \geq 1 - \frac{m-1}{\tau m}\left(1 - \left(1 - \frac{1}{m}\right)^{\tau m}\right)$$

$$\underset{(1)}{\geq} 1 - \frac{1}{\tau}(1 - \frac{1}{m})\left(1 - e^{-\tau}(1 - \frac{\tau}{m})\right)$$

$$\underset{(2)}{\geq} 1 - \frac{1 - e^{-\tau}}{\tau}$$

$$\geq 1 - \frac{1 - e^{-c}}{c} > 0,$$

where we used in the second inequality (1) that $\left(1 - \frac{1}{m}\right)^{\tau m} \geq e^{-\tau}(1 - \frac{\tau}{m})$ (we can prove it by taking the logarithm and compare their Taylor expansion). We also used in the third inequality (2) that $1 - \frac{1}{\tau}(1 - \frac{1}{m})\left(1 - e^{-\tau}(1 - \frac{\tau}{m})\right) - 1 - \frac{1 - e^{-\tau}}{\tau} = \frac{1}{\tau m}\left(1 - (1 + \tau)e^{-\tau} + \frac{\tau e^{-\tau}}{m}\right) \geq \frac{e^{-\tau}}{m^2} \geq 0$. Wrapping-up yields

$$\mathbb{E}[\|\theta_T - \theta'_T\|_2] \geq \|\mathbb{E}[\theta_T]\|_2 \geq \frac{1}{2}(1 - \frac{1 - e^{-c}}{c})\frac{f+1}{f + \frac{1}{m}}p\gamma CT \in \Omega(p\gamma CT).$$

Building on this lower bound for uniform argument stability, we then establish a connection to uniform stability, allowing us to complete the comparison with the upper bound from Theorem 3.3. Indeed,

$$\mathbb{E}\left[\ell(\theta_T; (v, -\frac{C}{\sqrt{L}})) - \ell(\theta'_T; (v, -\frac{C}{\sqrt{L}}))\right] = C\sqrt{L}\mathbb{E}\lambda_T + \frac{C^2}{2L} - \frac{C^2}{2L} \in \Omega\left(pC^2\gamma T\right). \qquad \square$$

## E. Stability Discrepancy Between the Threat Models in Nonconvex Learning

This section presents additional comparisons in nonconvex settings that were omitted from the main text. Specifically, we compare the stability bounds obtained in Theorem E.3, using the SMEA aggregation rule under data poisoning—$\varepsilon_{\text{SMEA}}^{\text{poisoning}}$ from (25)—against those under Byzantine failures—$\varepsilon^{\text{Byzantine}}$ from (24). This comparison yields the following ratio

$$\varepsilon^{\text{Byzantine}}/\varepsilon_{\text{SMEA}}^{\text{poisoning}} = \left(\frac{\frac{1}{(n-f)m} + \sqrt{\kappa}}{\frac{1}{(n-f)m} + \frac{f}{n-f}}\right)^{\frac{1}{c+1}}. \tag{23}$$

Here, we recover an analogous discrepancy to the convex learning case: we improve the dependency from $\sqrt{\kappa}$ to $\frac{f}{n-f}$. A rigorous derivation of this comparison is provided below.

Unlike GD—where the differing samples affect every iteration—in SGD, it may not be sampled immediately. This distinction enables our analysis to yield better stability upper bound with SGD, mimicking the proof technique of Hardt et al. (2016). Below, we state a lemma reasoning about the first occurrence of the differing samples.

**Lemma E.1.** *Consider the setting described in Section 2, under either Byzantine failures or data poisoning. We assume for all $z \in \mathcal{Z}$, $\ell(\cdot; z)$ nonnegative and uniformly bounded by $\ell_\infty \in \mathbb{R}_+$. Let $\mathcal{S}$ and $\mathcal{S}'$ being two neighboring dataset, we denote $\theta_T = \mathcal{A}(\mathcal{S})$, $\theta'_T = \mathcal{A}(\mathcal{S}')$ and intermediate state of the algorithm $\theta_t$, $\theta'_t$ and $\delta_t = \|\theta_t - \theta'_t\|_2$ for any $t \in \{0, \ldots, T\}$. Then for every $z \in \mathcal{Z}$ and every $t_0 \in \{0, \ldots, \min(m,T)\}$, the following holds for SGD with sampling with replacement*

$$\mathbb{E}_{\mathcal{A}}[|\ell(\theta_T; z) - \ell(\theta'_T; z)|] \leq \min\{\ell_\infty, \frac{t_0}{m}\ell_\infty + \mathbb{E}_{\mathcal{A}}[|\ell(\theta_T; z) - \ell(\theta'_T; z)| \mid \delta_{t_0} = 0]\}$$

*Proof.* Let $z \in \mathcal{Z}$ and $t_0 \in \{0, \dots, \min(m, T)\}$, we have

$$
\begin{aligned}
\mathbb{E}_{\mathcal{A}}[|\ell(\theta_T; z) - \ell(\theta'_T; z)|] = \quad & \mathbb{E}_{\mathcal{A}}[|\ell(\theta_T; z) - \ell(\theta'_T; z)| | \delta_{t_0} = 0] \mathbb{P}(\delta_{t_0} = 0) \\
& + \mathbb{E}_{\mathcal{A}}[|\ell(\theta_T; z) - \ell(\theta'_T; z)| | \delta_{t_0} \neq 0] \mathbb{P}(\delta_{t_0} \neq 0) \\
\leq \quad & \mathbb{E}_{\mathcal{A}}[|\ell(\theta_T; z) - \ell(\theta'_T; z)| | \delta_{t_0} = 0] + \mathbb{P}(\delta_{t_0} \neq 0) \ell_\infty.
\end{aligned}
$$

To bound $\mathbb{P}(\delta_{t_0} \neq 0)$, we consider $T_0$, the random variable of the first step the differing samples are drawn. We have

$$
\mathbb{P}(\delta_{t_0} \neq 0) \leq \mathbb{P}(T_0 \leq t_0) \leq \min\left(1, \frac{t_0}{m}\right).
$$

For the first inequality, we used the fact that under identical conditions (i.e., the same seed and initial state), the algorithm operating on two neighboring datasets will produce the same intermediate states until the differing samples are drawn. For the second inequality, we use the union bound to obtain $\mathbb{P}(T_0 \leq t_0) \leq \sum_{t=1}^{t_0} \mathbb{P}(T_0 = t) = \min(1, \frac{t_0}{m})$. $\qquad \square$

The following supporting result will be used to derive readable bounds.

**Lemma E.2.** *Let $c > 0$, $T \in \mathbb{N}^*$ and $t_0 \in \{0, \dots, T-1\}$. Then*

$$
\sum_{t=t_0}^{T-1} \frac{1}{t} \prod_{s=t+1}^{T-1} e^{\frac{c}{s}} \leq \frac{1}{c} \left(\frac{T}{t_0}\right)^c.
$$

*Proof.*

$$
\sum_{t=t_0}^{T-1} \frac{1}{t} \prod_{s=t+1}^{T-1} e^{\frac{c}{s}} = \sum_{t=t_0}^{T-1} \frac{1}{t} e^{c \sum_{s=t+1}^{T-1} \frac{1}{s}} \leq \sum_{t=t_0}^{T-1} \frac{e^{c \log(\frac{T}{t})}}{t} = T^c \sum_{t=t_0}^{T-1} t^{-c-1} \leq \frac{1}{c} \left(\frac{T}{t_0}\right)^c.
$$

Where we used the following sum-integral inequality: for any non-increasing and integrable function $f$, and for $a, b \in \mathbb{N}$, it holds that $\sum_{t=a}^{b} f(t) \leq \sum_{t=a}^{b} \int_{t-1}^{t} f(s) ds = \int_{a-1}^{b} f(s) ds$. Hence, with $\alpha \in \mathbb{R}_+^*$,

$$
\sum_{s=t+1}^{T-1} \frac{1}{s} \leq \int_{t}^{T-1} \frac{1}{s} ds = \ln\left(\frac{T-1}{t}\right) \leq \ln\left(\frac{T}{t}\right),
$$

and

$$
\sum_{t=t_0}^{T-1} t^{-\alpha-1} \leq \int_{t_0}^{T} t^{-\alpha-1} dt \leq -\frac{1}{\alpha}(T^{-\alpha} - t_0^{-\alpha}) \leq \frac{1}{\alpha} t_0^{-\alpha}. \qquad \square
$$

Building on the above lemmas, we now state uniform stability bounds for SGD with nonconvex objective. Our focus is SGD, since GD lacks meaningful uniform stability guarantees in the nonconvex regime issue illustrated in Hardt et al. (2016, Figure 10) and demonstrated in Charles & Papailiopoulos (2018, Section 6). This instability arises because a single differing sample affects *every* step of GD. In contrast, SGD only incorporates the differing sample when it is sampled, allowing us to reason about the first such occurrence. This distinction enables the following bounds.

**Theorem E.3.** *Consider the setting described in Section 2 under Byzantine failures. Let $\mathcal{A} = $ SGD, with a $(f, \kappa)$-robust aggregation rule $F$. Suppose $\mathcal{A}$ is run for $T \in \mathbb{N}^*$ iterations. Assume $\forall z \in \mathcal{Z}$, $\ell(\cdot; z)$ bounded by $\ell_\infty$, nonconvex, $C$-Lipschitz and $L$-smooth. Then, with a monotonically non-increasing learning rate $\gamma_t \leq \frac{c}{Lt}$ with $t \in \{0, \dots, T-1\}$ and $c > 0$, we have for any neighboring datasets $\mathcal{S}, \mathcal{S}'$*

$$
\sup_{z \in \mathcal{Z}} \mathbb{E}_{\mathcal{A}}[|\ell(\mathcal{A}(\mathcal{S}); z) - \ell(\mathcal{A}(\mathcal{S}'); z)|] \leq 2 \left(\frac{2C^2}{L}\right)^{\frac{1}{c+1}} \left(\sqrt{\kappa} + \frac{1}{(n-f)m}\right)^{\frac{1}{c+1}} \left(\frac{\ell_\infty T}{m}\right)^{\frac{c}{c+1}}. \tag{24}
$$

*Alternatively, if we analyze the same setting under data poisoning with the SMEA aggregation rule, it holds that for any neighboring datasets $\mathcal{S}, \mathcal{S}'$*

$$
\sup_{z \in \mathcal{Z}} \mathbb{E}_{\mathcal{A}}[|\ell(\mathcal{A}(\mathcal{S}); z) - \ell(\mathcal{A}(\mathcal{S}'); z)|] \leq 2 \left(\frac{2C^2}{L}\right)^{\frac{1}{c+1}} \left(\frac{f}{n-f} + \frac{1}{(n-f)m}\right)^{\frac{1}{c+1}} \left(\frac{\ell_\infty T}{m}\right)^{\frac{c}{c+1}}. \tag{25}
$$

*Proof.* The reasoning behind (16), as detailed in the proof of Theorem 3.1 in Appendix C.2, applies equally here. We recall the recursion, for $t \in \{0, \ldots, T-1\}$,

$$\mathbb{E}_{\mathcal{A}}[\|\theta_{t+1} - \theta'_{t+1}\|_2] \leq \eta_{G_{\gamma_t}^{\mathrm{SGD}}} \mathbb{E}_{\mathcal{A}}[\|\theta_t - \theta'_t\|_2] + \gamma_t \sigma_{\mathrm{SGD}}^F,$$

where $\eta_{G_{\gamma_t}}^{\mathrm{SGD}} \leq 1 + \gamma_t L$ for nonconvex and $L-$smooth functions (cf. Lemma B.4), and $\sigma_{\mathrm{SGD}}^F = 2C\left(\frac{1}{(n-f)m} + \sqrt{\kappa}\right)$. Invoking Lemma B.4 and considering $t_0 \in \{0, \ldots, T-1\}$, we have, for all $t \in \{t_0, \ldots, T-1\}$,

$$\mathbb{E}_{\mathcal{A}}[\|\theta_{t+1} - \theta'_{t+1}\|_2 | \|\theta_{t_0} - \theta'_{t_0}\|_2 = 0] \leq (1 + \gamma_t L)\mathbb{E}_{\mathcal{A}}[\|\theta_t - \theta'_t\|_2 | \|\theta_{t_0} - \theta'_{t_0}\|_2 = 0] + \gamma_t \sigma_{\mathrm{SGD}}^F$$
$$\leq e^{\gamma_t L}\mathbb{E}_{\mathcal{A}}[\|\theta_t - \theta'_t\|_2 | \|\theta_{t_0} - \theta'_{t_0}\|_2 = 0] + \gamma_t \sigma_{\mathrm{SGD}}^F, \tag{26}$$

where we used $1 + \gamma_t L \leq e^{\gamma_t L} = e^{c/t}$. By summing (26), and then invoking Lemma E.2, we obtain

$$\mathbb{E}_{\mathcal{A}}[\|\theta_T - \theta'_T\|_2 | \|\theta_{t_0} - \theta'_{t_0}\|_2 = 0] \leq \frac{c\sigma_{\mathrm{SGD}}^F}{L} \sum_{t=t_0}^{T-1} \frac{1}{t} \prod_{s=t+1}^{T-1} e^{\frac{c}{s}} \leq \frac{\sigma_{\mathrm{SGD}}^F}{L}\left(\frac{T}{t_0}\right)^c.$$

Substituting our derivation into the bound from Lemma E.1 yields

$$\mathbb{E}_{\mathcal{A}}[|\ell(\theta_T; z) - \ell(\theta'_T; z)|] \leq \frac{t_0 \ell_\infty}{m} + \frac{\sigma_{\mathrm{SGD}}^F C}{L}\left(\frac{T}{t_0}\right)^c.$$

We approximately minimize the expression with respect to $t_0$ by balancing the two terms (i.e., by equating them),

$$0 \leq \widetilde{t_0} = \left(\frac{m\sigma_{\mathrm{SGD}}^F C}{L\ell_\infty}\right)^{\frac{1}{c+1}} T^{\frac{c}{c+1}} \underset{\text{for sufficiently large } T \geq \frac{m\sigma_{\mathrm{SGD}}^F C}{L\ell_\infty}}{\leq} T.$$

Finally, we obtain the result by directly substituting $\widetilde{t_0}$ into the bound

$$\mathbb{E}_{\mathcal{A}}[|\ell(\theta_T; z) - \ell(\theta'_T; z)|] \leq 2\left(\frac{\sigma_{\mathrm{SGD}}^F C}{L}\right)^{\frac{1}{c+1}}\left(\frac{\ell_\infty T}{m}\right)^{\frac{c}{c+1}}.$$

Alternatively, we analyze the same setting under data poisoning with the SMEA aggregation rule The proof under data poisoning is analogous. Indeed, the reasoning behind (22), as detailed in the proof of Theorem 3.3 in Appendix D.1, applies equally here. We recall the recursion, with $\eta_{G_{\gamma_t | S_t^* \cap S_t^{*'}}}^{\mathrm{SGD}} \leq (1 + \gamma_t L)$ for nonconvex and smooth functions Lemma B.4, considering $t_0 \in \{0, \ldots, T-1\}$, for $t \in \{t_0, \ldots, T-1\}$,

$$\mathbb{E}_{\mathcal{A}}[\delta_{t+1} | \delta_{t_0} = 0] \leq (1 + \gamma_t L)\mathbb{E}_{\mathcal{A}}[\|\theta_t - \theta'_t\|_2 | \|\theta_{t_0} - \theta'_{t_0}\|_2 = 0] + 2\gamma_t C\frac{f + \frac{1}{m}}{n - f}.$$

Following the same derivation as in the proof of Theorem E.3 above yields (25). $\qquad \square$

## F. Deferred Proofs and Additional Details from Section 4

### F.1. Proof of Lemma 4.1 and Theorem 4.2

Below we compute the generalization error of a learning algorithm in the setting explicited in Section 4.

**Lemma 4.1.** Consider the setting described in Section 2, with $m = 1$, regardless of the assumed threat model. There exist $\ell \in \mathbb{R}^{\Theta \times \mathcal{Z}}$ such that $\forall z \in \mathcal{Z}, \ell(\cdot; z)$ is $C$-Lipschitz, $L$-smooth and convex, data distributions $\{p_i\}_{i \in \mathcal{H}}$ over $\mathcal{Z}$, such that for any distributed algorithm $\mathcal{A}$, we have

$$\left|\mathbb{E}_{\mathcal{A}, \mathcal{S} \sim \otimes_{i \in \mathcal{H}}(p_i^{\otimes m})}\left[R_{\mathcal{H}}(\mathcal{A}(\mathcal{S})) - \widehat{R}_{\mathcal{H}}(\mathcal{A}(\mathcal{S}))\right]\right| = \frac{1}{4(n-f)} \sup_{z \in \mathcal{Z}} \mathbb{E}_{\mathcal{A}}\left[\ell(\mathcal{A}(S); z) - \ell(\mathcal{A}(S'); z)\right], \tag{27}$$

where the dependence on the threat model arises solely through the stability term.

*Proof.* We consider a linear loss function $z \in [-C, C], \theta \in \mathbb{R} \mapsto z\theta$, and we assume every honest local distribution is a Dirac (regardless of the value where the singularity is), except for a "pivot" honest worker (indexed 1 without loss of generality) whose distribution is a mixture of Dirac $p_1 = \frac{1}{2}\delta_0 + \frac{1}{2}\delta_{-C}$. In this case, when the number of local samples is fixed to one ($m = 1$), the two possible datasets are always neighboring. Denote them $S^{(-C)}$ and $S^{(0)}$. We denote the parameters resulting from $\mathcal{A}$ on the two different datasets by $\mathcal{A}(S^{(0)}) = \theta_T^{(0)}$ and $\mathcal{A}(S^{(-C)}) = \theta_T^{(-C)}$.

In this setup, we can compute the generalization error exactly as follows

$$
\begin{aligned}
\mathbb{E}_{\mathcal{S}}\left[R_{\mathcal{H}}(\mathcal{A}(\mathcal{S})) - \widehat{R}_{\mathcal{H}}(\mathcal{A}(\mathcal{S}))\right] &= \frac{1}{|\mathcal{H}|}\mathbb{E}_{\mathcal{S}}\left[\mathbb{E}_{z \sim p_1}[\theta_T^{(z_1)}z] - \theta_T^{(z_1)}z_1\right] \\
&= \frac{1}{|\mathcal{H}|}\mathbb{E}_{\mathcal{S}}\left[\frac{1}{2}\theta_T^{(z_1)} \times 0 + \frac{1}{2}\theta_T^{(z_1)} \times (-C) - \theta_T^{(z_1)}z_1\right] \\
&= \frac{1}{|\mathcal{H}|}\left(-\frac{C}{2}\mathbb{E}_{\mathcal{S}}\theta_T^{(z_1)} - \mathbb{E}_{\mathcal{S}}\left[\theta_T^{(z_1)}z_1\right]\right) \\
&= \frac{1}{|\mathcal{H}|}\left(-\frac{C}{2}\left(\frac{1}{2}\theta_T^{(0)} + \frac{1}{2}\theta_T^{(-C)}\right) - \frac{1}{2}\theta_T^{(0)} \times 0 - \frac{1}{2}\theta_T^{(-C)} \times (-C)\right) \\
&= \frac{C}{|\mathcal{H}|}\left(-\frac{1}{4}\theta_T^{(0)} - \frac{1}{4}\theta_T^{(-C)} + \frac{1}{2}\theta_T^{(-C)}\right) \\
&= \frac{C\left(\theta_T^{(-C)} - \theta_T^{(0)}\right)}{4(n-f)}.
\end{aligned}
\tag{28}
$$

As the above computation is independent of the assumed threat model, we have

$$
\left|\mathbb{E}_{\mathcal{A},\mathcal{S}}\left[R_{\mathcal{H}}(\mathcal{A}(\mathcal{S})) - \widehat{R}_{\mathcal{H}}(\mathcal{A}(\mathcal{S}))\right]\right| = \frac{1}{4(n-f)}\sup_{z \in [-C,C]}\mathbb{E}_{\mathcal{A}}\left|\ell(\mathcal{A}(S^{(-C)}); z) - \ell(\mathcal{A}(S^{(0)}); z)\right|. \qquad \square
$$

Building upon the theorem above, we leverage our stability analysis to demonstrate that the Byzantine failure and data poisoning threat models differ fundamentally in their capacity to affect the generalization error of distributed learning algorithms.

**Theorem 4.2.** Consider the setting of Lemma 4.1, $\frac{n}{3} \leq f < \frac{n}{2}$, $\mathcal{A} = $ GD with SMEA. There exist $\ell \in \mathbb{R}^{\Theta \times \mathcal{Z}}$ such that $\forall z \in \mathcal{Z}, \ell(\cdot; z)$ is $C$-Lipschitz, $L$-smooth and convex, data distributions $\{p_i\}_{i \in \mathcal{H}}$ over $\mathcal{Z}$ and a Byzantine attack such that for any data poisoning attacks, we have

$$
\frac{\mathcal{E}_{gen}^{\text{byz}}}{\mathcal{E}_{gen}^{\text{pois}}} \in \Omega\left(\frac{n-f}{\sqrt{f(n-2f)}}\right),
\tag{29}
$$

where $\mathcal{E}_{gen}^{\text{byz}}$ and $\mathcal{E}_{gen}^{\text{pois}}$ denote generalization errors under the Byzantine and data poisoning attacks, respectively.

*Proof.* We specialize Lemma 4.1 to $\mathcal{A} = $ GD with SMEA. If we assume every honest local distribution is a Dirac with singularity at the value defined in the paragraph *(i)* of the proof of Theorem D.4—except for a "pivot" honest worker, indexed 1, whose distribution is a mixture $p_1 = \frac{1}{2}\delta_0 + \frac{1}{2}\delta_{-C}$, then the only two possible datasets for honest workers align with the worst-case Byzantine and data-poisoning scenarios considered in our lower bounds (see Appendices C.3 and D.2.1 for details of the datasets used). Hence, using the attacks described therein demonstrates that the generalization error exhibits the same intrinsic gap between the two threat models, consistent with the uniform stability analysis, as illustrated in Figure 1. That is $\mathcal{E}_{gen}^{\text{byz}}$ scales according to (5) and $\mathcal{E}_{gen}^{\text{pois}}$ scales according to (10) (both scaled by $\frac{1}{4(n-f)}$), as revealed by (28). $\square$

### F.2. Numerical Analysis Additional Details

We describe here the details of our numerical experiment presented in Section 4 into two points: *(i)* the problem instantiation; *(ii)* our tailored Byzantine attack.

*(i)* We instantiate the setting from the proof of Theorem D.4 (see full details therein). That is, we consider $\mathcal{A} = $ GD under a linear loss function, using the parameters $C = 1, \gamma = 1, T = 5, n = 15, m = 1$. We make vary $f$ from 1 to 7, and for each value instantiate the dataset presented in the proof of Theorem D.4.

*(ii)* In our tailored Byzantine failures, the identities of the Byzantine workers remain fixed throughout the optimization process, but vary depending on the number of Byzantine workers $f$. The set of Byzantine identities is defined as $\mathcal{B} := \{1, \ldots, n\} \setminus \mathcal{H}$, and is summarized in the table below. As we will see, we optimize the values sent by Byzantine workers so that they are selected while also biasing the gradient statistics—either toward positive or negative values—depending on an event that triggers them to switch their communicated value. To construct a maximally impactful scenario, we choose the identities of the Byzantine workers to maximize the minimal variance over all subsets of size $n - f$. This increases their biasing power, as it hinges on the smallest variance among these subsets. For instance, when $f = 2$, the Byzantine workers can be hidden in the dominant subset $N$; however, when $f = 6$, they must be distributed across both competing subgroups $E$ and $F$ to maintain influence.

$$
\begin{aligned}
f &= 1 : \mathcal{B} = \{2\} &\quad f &= 4 : \mathcal{B} = \{2, 3, 4, 5\} \\
f &= 2 : \mathcal{B} = \{2, 3\} &\quad f &= 5 : \mathcal{B} = \{2, 3, 4, 5, 6\} \\
f &= 3 : \mathcal{B} = \{2, 3, 4\} &\quad f &= 6 : \mathcal{B} = \{7, 8, 9, 10, 11, 12\} \\
& &\quad f &= 7 : \mathcal{B} = \{6, 7, 8, 9, 10, 11, 12\}
\end{aligned}
$$

For instance, if a single sample from worker 1 (without loss of generality) is changed from 0 to $-C$ (defining a neighboring dataset), the server receives a different average $-\frac{1}{m}C < 0$ that triggers the Byzantine workers to amplify the positive or negative parameter direction by adaptively crafting updates that remain within the aggregation rule's selection range. Specifically, their algorithm is as follows.

– If $g_0^{(1)} \geq 0$, all Byzantine workers send the value $\alpha_f$, where $\alpha_f$ is numerically chosen to be as small as possible under the constraint that the SMEA aggregation rule continues to select the values sent by Byzantine workers in subsequent steps.

– If $g_0^{(1)} < 0$, all Byzantine workers send the value $\beta_f$, where $\beta_f$ is numerically chosen to be as large as possible under the constraint that the SMEA aggregation rule continues to select the values sent by Byzantine workers in subsequent steps.

The following pseudocode summarizes the behavior of the Byzantine workers. Let $\alpha \in \mathbb{R}$ and

$$
S_\alpha^* \in \underset{|S|=n-f}{\arg\min} \operatorname{Var}\left(\{g_k\}_{k \in S \cap \mathcal{H}} \cup \{\alpha\}_{k \in S \cap \mathcal{B}}\right).
$$

---

**Algorithm 3** Byzantine worker $i \notin \mathcal{H}$ behavior, $\epsilon = 10^{-3}$

**if** $g_0^{(1)} \geq 0$ **then**
    **return** $\alpha_f = \inf\left\{\alpha; i \in S_\alpha^*\right\} + \epsilon$   # amplifies the initial direction
**else**
    **return** $\beta_f = \sup\left\{\beta; i \in S_\beta^*\right\} - \epsilon$   # amplifies the initial direction
**end if**

---

## G. Refined Bounds Under Bounded Heterogeneity and Bounded Variance Assumptions

For clarity and simplicity, we chose to emphasize our results under the broad bounded gradient assumption (i.e., Lipschitz-continuity of the loss function) in the main text. In this section, we present results derived under refined assumptions—namely, bounded heterogeneity and bounded variance—in place of the more general bounded gradient condition. These assumptions are more commonly used in the optimization literature (as opposed to the generalization literature).

We formally define these assumptions below.

**Assumption G.1** (Bounded heterogeneity). *There exists $G < \infty$ such that $\forall \theta \in \Theta$, $\frac{1}{|\mathcal{H}|} \sum_{i \in \mathcal{H}} \|\nabla \widehat{R}_i(\theta) - \nabla \widehat{R}_{\mathcal{H}}(\theta)\|_2^2 \leq G^2$.*

**Assumption G.2** (Bounded variance). *There exists $\sigma < \infty$ such that $\forall i \in \mathcal{H}$, $\forall \theta \in \Theta$, $\frac{1}{m} \sum_{x \in \mathcal{D}_i} \|\nabla \ell(\theta; x) - \nabla \widehat{R}_i(\theta)\|_2^2 \leq \sigma^2$.*

As shown in the following, our analysis techniques can yield tighter bounds when assuming bounded heterogeneity and bounded variance. However, we note that these bounds do not provide additional conceptual insights within the scope of

our discussion. Indeed, within the uniform stability framework, if we assume only bounded gradients and nothing more, there exist distributed learning settings where $G$ is a constant multiple of the Lipschitz constant $C$. For instance, consider a scenario where half of the honest workers produce gradients with norm $C$, and the other half produce gradients with norm $-C$; in this case, $G$ would be $C$.

As an example, we derive bounds on the expected spectral norm of the empirical covariance matrix of honest workers' gradients that are expressed in terms of $G$ and $\sigma$. These estimates can be directly used in Theorem 3.1 to provide tighter bounds under additional assumptions of bounded heterogeneity and bounded variance.

**Lemma G.3.** *Consider the setting described in Section 2 under Byzantine attacks with Assumptions G.1 and G.2. Let $\mathcal{A} \in \{\mathrm{GD}, \mathrm{SGD}\}$, with a $(f, \kappa)$-robust aggregation rule, for $T \in \mathbb{N}^*$ iterations. We have, for every $t \in \{1, \ldots, T-1\}$,*

$$\mathbb{E}_{\mathcal{A}} \|\Sigma_{\mathcal{H},t}\|_{\mathrm{sp}} = \mathbb{E}_{\mathcal{A}}[\lambda_{\max}(\frac{1}{|\mathcal{H}|}\sum_{i \in \mathcal{H}}(g_t^{(i)} - \overline{g}_t)(g_t^{(i)} - \overline{g}_t)^{\mathsf{T}})] \leq \sigma^2 + G^2.$$

*Proof.* In the following, we derive a bound for the stochastic case $\mathcal{A} = \mathrm{SGD}$; the result naturally extends to the deterministic setting $\mathcal{A} = \mathrm{GD}$ as well. The proof of the refined upper bound leverages the ability to compute the expectation of the controllable noise—specifically, the noise resulting from random sample selection $J_t^{(i)}$, $i \in \mathcal{H}$, $t \in \{0, \ldots, T-1\}$—while conditioning on the uncontrollable noise introduced by Byzantine workers. Since the bound is independent of the latter, we establish the result by applying the law of total expectation. Let $t \in \{0, \ldots, T-1\}$,

$$\Delta_t = \lambda_{\max}(\frac{1}{|\mathcal{H}|}\sum_{i \in \mathcal{H}}(g_t^{(i)} - \overline{g}_t)(g_t^{(i)} - \overline{g}_t)^{\mathsf{T}}) = \sup_{\|v\|_2 \leq 1} \frac{1}{|\mathcal{H}|}\sum_{i \in \mathcal{H}}\langle v, g_t^{(i)} - \overline{g}_t\rangle^2.$$

We decompose,

$$g_t^{(i)} - \overline{g}_t = g_t^{(i)} - \nabla\widehat{R}_i(\theta_t) + \nabla\widehat{R}_i(\theta_t) - \nabla\widehat{R}_{\mathcal{H}}(\theta_t) + \nabla\widehat{R}_{\mathcal{H}}(\theta_t) - \overline{g}_t,$$

so that we have

$$\langle v, g_t^{(i)} - \overline{g}_t\rangle^2 = \langle v, \nabla\widehat{R}_i(\theta_t) - \nabla\widehat{R}_{\mathcal{H}}(\theta_t)\rangle^2 + \langle v, g_t^{(i)} - \nabla\widehat{R}_i(\theta_t) + \nabla\widehat{R}_{\mathcal{H}}(\theta_t) - \overline{g}_t\rangle^2$$
$$+ 2\langle v, g_t^{(i)} - \nabla\widehat{R}_i(\theta_t) + \nabla\widehat{R}_{\mathcal{H}}(\theta_t) - \overline{g}_t\rangle\langle v, \nabla\widehat{R}_i(\theta_t) - \nabla\widehat{R}_{\mathcal{H}}(\theta_t)\rangle.$$

Upon averaging, taking the supremum over the unit ball, and then taking total expectations, we get

$$\mathbb{E}\left[\sup_{\|v\|_2 \leq 1} \frac{1}{|\mathcal{H}|}\sum_{i \in \mathcal{H}}\langle v, g_t^{(i)} - \overline{g}_t\rangle^2\right] \leq \mathbb{E}\left[\sup_{\|v\|_2 \leq 1} \frac{1}{|\mathcal{H}|}\sum_{i \in \mathcal{H}}\langle v, \nabla\widehat{R}_i(\theta_t) - \nabla\widehat{R}_{\mathcal{H}}(\theta_t)\rangle^2\right]$$
$$+ \mathbb{E}\left[\sup_{\|v\|_2 \leq 1} \frac{1}{|\mathcal{H}|}\sum_{i \in \mathcal{H}}\langle v, g_t^{(i)} - \nabla\widehat{R}_i(\theta_t) + \nabla\widehat{R}_{\mathcal{H}}(\theta_t) - \overline{g}_t\rangle^2\right]$$
$$+ 2\mathbb{E}\left[\sup_{\|v\|_2 \leq 1} \frac{1}{|\mathcal{H}|}\sum_{i \in \mathcal{H}}\langle v, g_t^{(i)} - \nabla\widehat{R}_i(\theta_t) + \nabla\widehat{R}_{\mathcal{H}}(\theta_t) - \overline{g}_t\rangle\langle v, \nabla\widehat{R}_i(\theta_t) - \nabla\widehat{R}_{\mathcal{H}}(\theta_t)\rangle\right]. \quad (30)$$

We show that the last term in (30) is non-positive. In fact, with $M = N + N^{\mathsf{T}}$,

$$N = \sum_{i \in \mathcal{H}}(g_t^{(i)} - \nabla\widehat{R}_i(\theta_t) + \nabla\widehat{R}_{\mathcal{H}}(\theta_t) - \overline{g}_t)(\nabla\widehat{R}_i(\theta_t) - \nabla\widehat{R}_{\mathcal{H}}(\theta_t))^{\mathsf{T}},$$

and using Lemma D.4 from Allouah et al. (2023b),

$$\mathbb{E}\left[\sup_{\|v\|_2 \leq 1} 2\sum_{i \in \mathcal{H}}\langle v, g_t^{(i)} - \nabla\widehat{R}_i(\theta_t) + \nabla\widehat{R}_{\mathcal{H}}(\theta_t) - \overline{g}_t\rangle\langle v, \nabla\widehat{R}_i(\theta_t) - \nabla\widehat{R}_{\mathcal{H}}(\theta_t)\rangle\right]$$
$$= \mathbb{E}\left[\sup_{\|v\|_2 \leq 1}\langle v, Mv\rangle\right] \leq 9^d \sup_{\|v\|_2 \leq 1}\mathbb{E}[\langle v, Mv\rangle]$$

$$= 9^d \sup_{\|v\|_2 \leq 1} 2\mathbb{E} \sum_{i \in \mathcal{H}} \langle v, \underbrace{\mathbb{E}[g_t^{(i)} - \nabla\widehat{R}_i(\theta_t) + \nabla\widehat{R}_\mathcal{H}(\theta_t) - \overline{g}_t | \theta_t]}_{=0} \rangle \langle v, \nabla\widehat{R}_i(\theta_t) - \nabla\widehat{R}_\mathcal{H}(\theta_t) \rangle.$$

We now bound the second term in (30),

$$\mathbb{E}\left[ \sup_{\|v\|_2 \leq 1} \frac{1}{|\mathcal{H}|} \sum_{i \in \mathcal{H}} \langle v, g_t^{(i)} - \nabla\widehat{R}_i(\theta_t) + \nabla\widehat{R}_\mathcal{H}(\theta_t) - \overline{g}_t \rangle^2 \right] \leq \mathbb{E}\left[ \sup_{\|v\|_2 \leq 1} \frac{1}{|\mathcal{H}|} \sum_{i \in \mathcal{H}} \langle v, g_t^{(i)} - \nabla\widehat{R}_i(\theta_t) \rangle^2 \right]$$

$$\leq \mathbb{E}\left[ \frac{1}{|\mathcal{H}|} \sum_{i \in \mathcal{H}} \underbrace{\mathbb{E}_{J_t^{(i)}}[\|g_t^{(i)} - \nabla\widehat{R}_i(\theta_t)\|_2^2]}_{\leq \sigma^2} \right].$$

Finally, we bound the first term in (30) with Assumption G.1,

$$\mathbb{E}\sup_{\|v\|_2 \leq 1} \frac{1}{|\mathcal{H}|} \sum_{i \in \mathcal{H}} \langle v, \nabla\widehat{R}_i(\theta_t) - \nabla\widehat{R}_\mathcal{H}(\theta_t) \rangle^2 \leq \sup_{\theta \in \Theta} \sup_{\|v\|_2 \leq 1} \frac{1}{|\mathcal{H}|} \sum_{i \in \mathcal{H}} \langle v, \nabla\widehat{R}_i(\theta) - \nabla\widehat{R}_\mathcal{H}(\theta) \rangle^2$$

$$= \sup_{\theta \in \Theta} \frac{1}{|\mathcal{H}|} \sum_{i \in \mathcal{H}} \|\nabla\widehat{R}_i(\theta) - \nabla\widehat{R}_\mathcal{H}(\theta)\|_2^2 \leq G^2. \qquad \square$$

## H. High-Probability Excess Risk Bounds for Robust Distributed Learning

In this section, we establish high-probability excess risk bounds (cf. inequality (1)) under smooth and strongly convex assumptions for robust distributed learning algorithms, providing insights into their generalization performance. Specifically, we combine a deterministic optimization error bound—adapted directly from prior work from Farhadkhani et al. (2024b)—with our high-probability stability bound.

High probability bound under uniform stability framework are derived as follows. The change of a single training sample has a small effect on a uniformly stable algorithm. This broadly implies low variance in the difference between empirical and population risk. If this difference also has low expectation, the empirical risk should accurately approximate the population risk (Bousquet & Elisseeff, 2002). In fact, we directly adapt the results from Bousquet et al. (2020) to our framework and state the following high-probability bound.

**Lemma H.1.** *Consider the setting described in Section 2, under either Byzantine failures or data poisoning. Let $\mathcal{A}$ an $\varepsilon$-uniformly stable. Assume $\forall z \in \mathcal{Z}, \ell(\cdot, z) \leq \ell_\infty$, then we have that for any $0 < \delta < 1$, with probability at least $1 - \delta$,*

$$|R_\mathcal{H}(\mathcal{A}(\mathcal{S})) - \widehat{R}_\mathcal{H}(\mathcal{A}(\mathcal{S}))| \in \mathcal{O}\left( \min\left\{ \varepsilon \ln(|\mathcal{H}|m) \ln(\frac{1}{\delta}) + \frac{\ell_\infty}{\sqrt{|\mathcal{H}|m}} \sqrt{\ln(\frac{1}{\delta})}, \left( \sqrt{\varepsilon\ell_\infty} + \frac{\ell_\infty}{\sqrt{|\mathcal{H}|m}} \right) \sqrt{\ln(\frac{1}{\delta})} \right\} \right)$$

**Data poisoning.** By adapting the optimization bound technique from Farhadkhani et al. (2024b, Theorem 4) to GD algorithm with SMEA under data poisoning, we show that for a constant number of steps—specifically, when the step count satisfies $T \geq |\ln(4\frac{f+1/m}{n-f} + \kappa)/\ln(1 - \frac{\mu}{L})| \in \mathcal{O}(1)$—the expected excess risk is of the order of

$$\frac{C^2}{\mu}\mathcal{O}\left( \frac{f}{n - 2f} + \frac{1}{m(n - 2f)} \right).$$

Additionally, we provide a high-probability bound on the excess risk using Lemma H.1. Specifically, for any $0 < \delta < 1$, with probability at least $1 - \delta$ the excess risk lies within

$$\mathcal{O}\left( \min\left\{ \frac{f}{n - 2f}C^2 + \left( \frac{f}{n - 2f} + \frac{1}{m(n - 2f)} \right) C^2 \ln((n - f)m) \ln(\frac{1}{\delta}) + \frac{\ell_\infty}{\sqrt{(n - f)m}} \sqrt{\ln(\frac{1}{\delta})}, \right.\right.$$

$$\left.\left. \frac{f}{n - 2f}C^2 + \left( C\sqrt{\ell_\infty \left( \frac{f}{n - 2f} + \frac{1}{m(n - 2f)} \right)} + \frac{\ell_\infty}{\sqrt{(n - f)m}} \right) \sqrt{\ln(\frac{1}{\delta})} \right\} \right)$$

In the absence of a minimax statistical lower bound for the heterogeneous case—and acknowledging that the comparison is imperfect due to differing assumptions—it remains informative to compare our resulting bound with those from Yin et al. (2018); Zhu et al. (2023), as well as Allouah et al. (2023a, Proposition 1).

**Byzantine failures.** Under Byzantine failures, the derivation proceeds similarly and yields an analogous bound, with a dependence on $\sqrt{\kappa}$ replacing the $\frac{f}{n-2f}$ term in the uniform stability bound.

# I. Additional Discussions

**On the relevance of worst-case bounds to practical learning scenarios.** Although variations in attack strength and implementation may contribute to the observed gap in generalization performance between the two threat models, our results indicate that this gap cannot be explained solely by the fact that data poisoning attacks used in practice are often empirically weak (suboptimal within their threat model), whereas Byzantine attacks used in practice tend to be empirically strong (near-optimal). Rather, the gap also reflects an inherent difference in harm capacity (Section 4).

We recall that a gap in test accuracy has been consistently reported in prior work on realistic datasets, further motivating our investigation. We list below examples of such work.

*(i)* Table 2 from Allouah et al. (2023a) and Figure 2, 4 and 6 from Allouah et al. (2023b), where ALIE, FOE, and SF simulate Byzantine failures, and LF or Mimic simulate data poisoning.

*(ii)* Figure 1 from Karimireddy et al. (2022), where LF simulates data poisoning attacks and IPM, ALIE, and BF simulate Byzantine attacks.

*(iii)* Table 2 and 5 from Fang et al. (2020), where LabelFlip, SingleWorker and Uniform simulate data poisoning attacks and Partial and Full simulate Byzantine attacks.

*(vi)* Figure 2 from Allouah et al. (2025b), where LabelFlipping simulates data poisoning attacks and SignFlipping simulates Byzantine attacks.

Before our work, it was unclear whether this empirically observed gap reflects a fundamental difference in adversarial power or merely the suboptimality of practical attacks. Prior theory failed to explain this observed gap (Farhadkhani et al., 2024b). Although designing new benchmarks and stronger attacks is a worthwhile direction, relying solely on empirical validation leads to an endless cycle of scenario-specific attacks without establishing a universal distinction. This underscores the need for a formal theoretical framework. We address this gap by developing a refined theory that incorporates generalization error, providing a principled explanation for the observed phenomenon.

**About the regime $f < \frac{n}{3}$.** We derived a lower bound of $\Omega\left(\frac{f}{n-2f}\right)$ for the Byzantine setting when $f < \frac{n}{3}$ (see Theorem 3.2 part (ii) as written in Appendix C.3). Although this differs from the $\Omega\left(\frac{f}{n-f}\right)$ bound obtained under data poisoning, the two bounds are within a constant factor. For clarity and brevity, we omitted this result from the main text.

We do not prove the tightness of this bound, but we conjecture that it is tight for GD with SMEA. We provide the following intuition, partially outlined in Appendix F.2 point (ii). The SMEA aggregation rule selects the subset of size $n - f$ with minimal variance. Therefore, Byzantine workers' capacity to skew the output is capped by this minimal variance. To achieve the largest possible lower bound, we have to maximize the variance of the honest subsets. The maximal variance for a set of $n - f$ values bounded by $C > 0$ is achieved when half the values are $C$ and the other half are $-C$ (yielding a variance of $C^2$). Yet, when $f < \frac{n}{3}$, the inequality $\frac{n-f}{2} + f < n - f$ dictates that any selected subset containing the $f$ Byzantine updates *must* inevitably include a mix of both positive and negative honest values. Because of this forced mixture, Byzantine workers must limit the magnitude of their attacks to avoid being filtered out by SMEA, in contrast to the case $f \geq \frac{n}{3}$, where no such forced mixture exists.

**Does our conclusions would fundamentally change for other popular first-order optimization methods (e.g., momentum, variance reduction, multiple local steps)?** We believe our proof techniques are broadly applicable to robust first-order optimization, so the fundamental results should remain consistent. Quantitatively characterizing this phenomenon for specific algorithms is an interesting direction for future work. More concretely, similar recursions can be derived for stochastic heavy-ball momentum, using the proof techniques of Appendix C.2 (under Byzantine failures), and the refined approach in the proof of Theorem 3.3 (under data poisoning). These derivations reveal a similar gap as in robust distributed GD and SGD: a $\sqrt{\kappa}$ dependency appears under Byzantine failures, while the tighter bound $\frac{f}{n-f}$ arises under data poisoning. However, fully analyzing these more complex algorithms introduces additional technical challenges. To maintain clarity and focus, we chose to highlight the GD/SGD case, which already illustrates the core insights.

Many practical algorithms perform multiple local SGD steps to reduce communication overhead (e.g., FedAvg). However, our focus in this work is not on practical communication efficiency, but rather on understanding the generalization behavior in the presence of misbehaving agents in the most direct way. We thus chose to analyze the setting where each worker performs a single gradient step which provides a clearer and more focused framework to present and understand the fundamental algorithmic stability properties. We believe our insights extend to the multi-step case, and see no reason why the general conclusion of our paper would be impacted. Exploring this formally is an interesting direction for future work.

**Can we relax the convexity, Lipschitz smoothness or Lipschitz continuity assumptions?**

*Relaxing the convex assumption*: First, the $\mu$-strongly convex assumption can be relaxed to the $\mu$-Polyak–Łojasiewicz (Karimi et al., 2016) under Byzantine failures using techniques from Charles & Papailiopoulos (2018). Then, it is possible to derive uniform algorithmic stability upper-bounds for the nonconvex setting under both threat models through our proof techniques (cf. Appendix E).

*Relaxing the L-Lipschitz smoothness assumption*: This is indeed an interesting question. Bassily et al. (2020) provided a tight analysis of the uniform algorithmic stability of (non robust) GD/SGD under nonsmooth and convex loss functions. Combining their proof techniques with ours (cf. Appendix C.2 under Byzantine failures and the arguments in Theorem 3.3 under data poisoning) should be sufficient to extend our results.

*Relaxing the C-Lipschitz continuous assumption*: It can be relaxed by analyzing a data-dependent notion of stability known as *on-average algorithmic stability* (Lei & Ying, 2020). Although our focus is on *worst-case* relative harmfulness under the two threat models, on-average algorithmic stability offers a promising direction for future work. Our results highlight this path by showing that algorithmic stability is a meaningful tool for analyzing robust distributed algorithms.

**Extending our results to other aggregation rules.** We have focused on SMEA as it is an optimal high-dimensional robust aggregation rule for the optimization of training errors (Allouah et al., 2023b). Nevertheless, we discuss the possibility of extending our results to other rules.

- *Extending upper bounds:* Theorem 3.1 holds for any $(f, \kappa)$-robust aggregation rule. Under data poisoning, while the specific analytical technique depends on the aggregation rule, our proof strategy remains broadly applicable. As noted in the paragraph after Theorem 3.3, it holds for any rule that averages $n - f$ gradients, independent of the selection mechanism. As an other example, we have derived a similar upper bound using CWTM, but omitted it from the manuscript for brevity.

- *Extending lower bounds:* The datasets, loss functions, and attacks that we have constructed for our lower bounds are tailored for SMEA. While extending it to other rules is non-trivial, we believe our proof techniques are directly applicable to the related Covariance bound-Agnostic Filter (Allouah et al., 2025b), as both rules select vectors according to a variance minimization criterion. Designing novel attacks for fundamentally different rules, such as CWTM, remains a challenging but promising direction for future work motivated by our findings.

Together, these results indicate that our theoretical framework is informative and generalizable beyond SMEA. Detailed extensions of the constructions to other aggregation rules are left for future work.

**Practical implications.** *Cryptographic tools as a mitigation.* As explained in our conclusion, zero-knowledge proofs (ZKPs) mitigate Byzantine failures by verifying that each worker's input data lies within a valid range and that model updates are correctly computed from their committed dataset. In effect, ZKPs reduce Byzantine attacks to data poisoning. Since our theoretical analysis shows that algorithmic stability improves under data poisoning, this formally implies better generalization when ZKPs are deployed against Byzantine failures. To our knowledge, our work provides the first theoretical justification for using ZKPs in Byzantine-robust learning. While implementing ZKPs is beyond the scope of this paper, our theory offers strong evidence supporting their practical use.

*Design of new robust aggregations.* By proving that algorithmic stability (and, in turn, generalization) fundamentally differs between the two threat models, we provide the first theoretical justification for tailoring defenses to specific threats.

*(i) If the system is susceptible to Byzantine failures*: Our proposed Byzantine attack causes the ouptut of SMEA to exceed the Lipschitz constant of the loss function, a scenario not possible under data poisoning. This highlights the relevance of

strategies that cap the output norm, such as Adaptive Gradient Clipping (Allouah et al., 2025a), to improve algorithmic stability under Byzantine attacks. Analyzing the stability impact of such algorithmic modifications is nontrivial and represents a promising direction for future work. Alternatively, when computationally feasible, zero-knowledge proofs (discussed above) provide an off-the-shelf mitigation.

*(ii) If the system is only susceptible to data poisoning*: Future work should target robust aggregation rules that preserve input gradient regularity, further widening the stability gap.

**Discussion on other threat model formulations.**   Other threat models could be investigated in future work, building upon the proof techniques developed in our work. To offer some preliminary context, we briefly discuss two relevant threat model formulations below.

*1. Locally-poisonous threat model.* The recent work by Farhadkhani et al. (2024b), which studies the locally-poisonous threat model, indicates that this threat model tends to have a fundamentally lower impact on optimization error compared to the Byzantine failures and data poisoning threat models considered in our paper. Studying stability and generalization under locally-poisonous threat model is an interesting future work.

*2. Backdoor (or targeted) attacks.* While important, this threat model differs in goals and evaluation from our study of generalization error under untargeted attacks. That said, they can be seen as special cases of Byzantine failures and data poisoning. Extending our theory to handle targeted attacks like backdoors remains an interesting direction for future work.

