# OpenReview forum: "Tight Stability Bounds for Robust Distributed Learning: Byzantine Failures Hurt Generalization More than Data Poisoning"
_ICML.cc/2026/Conference — ICML 2026 regular_

### Official Review · Reviewer_zrHh · 2026-02-26

**Soundness:** 2
**Presentation:** 2
**Significance:** 1
**Originality:** 2
**Overall Recommendation:** 2
**Confidence:** 5

**Summary:**

This paper examines the performance of server-assisted distributed learning under two types of attacks: Byzantine attacks and data-poisoning attacks. It concludes that Byzantine attacks cause greater degradation of generalization performance.

**Compliance With Llm Reviewing Policy:**

Affirmed.

**Final Justification:**

I remain unconvinced by the authors’ response to my concerns regarding the motivation and the restricted scope of the results.

**Key Questions For Authors:**

see weakness

**Limitations:**

see weakness

**Strengths And Weaknesses:**

Strength:
1. Attack resilience is an important and timely research topic.

Weakness:

1. Without knowing whether the bound provides a necessary and sufficient condition for uniform stability, simply comparing the upper bounds does not offer a convincing assessment of performance under the two types of attacks.

2. In fact, Byzantine attacks include data poisoning attacks as a special case. Therefore, it is both logical and theoretically predictable that the generalization performance would be more severely compromised under Byzantine attacks than under data poisoning attacks. This raises questions about the significance of research that merely confirms this expected outcome.

3. The considered setting assumes the existence of a server, which significantly limits its applicability. In practice, when a server is present, one can employ detection mechanisms to identify and remove attacking nodes, rather than relying solely on so-called Byzantine-resilient strategies, often achieving substantially better results.

---

> ### Author Rebuttal · Authors · 2026-03-31
>
> The review appears to reflect several misunderstandings of the scope and contributions of our work. We therefore clarify below three foundational aspects, and respectfully ask the reviewer to reconsider their assessment in light of these clarifications.
>
> W1. **Our stability bounds are tight**
>
> There appears to be a misunderstanding here. We do not "*simply compare the upper bounds*"; rather, we establish **tight stability bounds**, which crucially rely on matching lower bounds (Theorems 3.2, 3.4). In addition, we are unclear about the intended meaning of "*necessary and sufficient condition for uniform stability*". Our results provide the **tight quantitative characterizations** of uniform stability for the considered algorithm under both threat models, which translate into a **fundamental generalization gap** (Theorem 4.2).
>
> W2. **About the significance of a theory accounting for this 'logical' performance difference**
>
> While it is intuitive for Byzantine failures to be more harmful than data poisoning, prior work has shown that **this intuition is not valid when it comes to the optimization (or training) error**. Indeed, the worst-case attack matching Byzantine upper bounds is derived under data poisoning (Farhadkhani *et al.*, 2024b).
>
> As discussed in the second paragraph of the Introduction, until now, the community has lacked a theoretical demonstration of this hierarchy, relying solely on numerical analysis. Our work fills this gap, showing that these two threat models have fundamentally different capabilities when it comes to harming the **generalization (or test) error**. Specifically, we derive the first Byzantine attack that achieves matching upper and lower bounds with a magnitude strictly greater than those under data poisoning. This distinction is critical: without it, there is no formal justification for designing defenses against the stronger capabilities of Byzantine workers.
>
> W3. **About the usefulness of Byzantine-resilient strategies**
>
> We respectfully disagree with the reviewer. The assumption of a central server captures a wide range of practical settings (e.g., federated learning and distributed computing). That said, extending our techniques to decentralized robust learning is feasible and constitutes a natural direction for future work.
>
> Without additional clarification, it is unclear what the reviewer means by "*mechanisms to identify and remove attacking nodes*". Moreover, the claim that "*detection mechanisms*" render Byzantine-resilient strategies obsolete appears to be at odds with the core motivation of the Byzantine-robust distributed learning literature (e.g., G. Baruch et al., 2019; P. Blanchard, 2017; R. Guerraoui et al., 2024). To the best of our knowledge, $(f, \kappa)$-robustness yields optimal optimization guarantees under Byzantine attacks. We would appreciate references supporting the reviewer's claim.

---

> > ### Author Rebuttal · Reviewer_zrHh · 2026-04-01
> >
> > Thank you for your response. However, a bound can be considered tight only if (1) it provides an upper bound on the error for all cases, and (2) there exists at least one instance in which the error exactly matches this bound. Could you clarify whether the bound in (5) satisfies both of these properties, and if so, provide a proof?
> >
> > The response to W2 is also somewhat confusing. It states that “this intuition is not valid when it comes to the optimization.” However, it is both logical and theoretically expected that Byzantine attacks would cause more severe damage than data poisoning attacks. I am unclear why this needs to be validated.

---

> > > ### Author Response · Authors · 2026-04-02
> > >
> > > We are glad to have resolved some of your concerns. Thanks for clarifying your remaining concerns in the follow-up, which we hope to resolve in the following.
> > >
> > > **W1.** We are unsure what may have misled the reviewer, as achieving tight bounds (i.e., matching upper and lower bounds) is the central focus of our paper and is discussed in detail throughout. Specifically, Theorem 3.2 presents an instance where the lower bound in Equation (5) matches the upper bound given from Theorem 3.1 in Equation (3), which holds for all instances. Thus, we explicitly demonstrate a case with matching upper and lower bounds. For further clarification, please see the paragraph immediately following the proof sketch of Theorem 3.2 (lines 174–180, second column). A complete summary of all proven tight relationships is also provided in Table 1. We hope this answers the reviewer’s concern on the tightness of our result.
> > >
> > > **W2.** We believe the following reasoning, which integrates both prior work and our contribution, will elucidate our previous response highlighting the non-triviality and importance of our result.
> > > - **Intuition**: Byzantine failures should be more harmful than data poisoning.
> > > - **Prior work**: Farhadkhani *et al.* (2024b) theoretically show that optimization error guarantees are identical under data poisoning or under Byzantine failures, contradicting this intuition. For further details, we refer the reviewer to Theorems 1 and 2 in Farhadkhani et al. (2024b), as well as Sections 1.3 and 1.4 for a comprehensive discussion.
> > > - **Our work**: We prove the generalization error can be order-greater under Byzantine failures compared to data poisoning (cf. Theorem 4.2), thus restoring the intuitive expectation. This result is both new and critical: providing the first formal justification for designing defenses against the stronger capabilities of Byzantine workers.
> > > - **Intuition on why the behavior in optimization and generalization is different under the two threat models**: see final paragraph of Section 4.

---

### Official Review · Reviewer_JVHc · 2026-03-12

**Soundness:** 3
**Presentation:** 3
**Significance:** 2
**Originality:** 2
**Overall Recommendation:** 4
**Confidence:** 4

**Summary:**

This paper studies the impact of different adversarial threat models on the generalization performance of robust distributed learning algorithms. In particular, the paper analyzes the difference between Byzantine failures (where workers may send arbitrary updates) and data poisoning attacks (where corruption is restricted to local datasets). The authors use algorithmic stability as the analytical framework to quantify generalization degradation under these two threat models. The paper derives stability upper and lower bounds for distributed (S)GD with robust aggregation rules such as SMEA. The paper then connects algorithmic stability to generalization error and proves that Byzantine failures can cause larger generalization error than data poisoning.

**Compliance With Llm Reviewing Policy:**

Affirmed.

**Final Justification:**

The rebuttal addressed my main concerns.

**Key Questions For Authors:**

1. What are the concrete practical implications of the derived stability bounds?
2. What insights do the bounds provide beyond the intuitive fact that Byzantine failures are stronger than data poisoning?
3. Can the analysis be extended to other robust aggregation rules?

**Limitations:**

The paper briefly includes an impact statement but does not adequately discuss the limitations of the theoretical framework, in particular, the practical implications of the theory.

**Strengths And Weaknesses:**

**Strength**
1. Understanding the differences between Byzantine failures and data poisoning is a relevant problem for distributed and federated learning security.
2. The paper provides detailed mathematical derivations of stability bounds using the algorithmic stability framework, which is a well-established method for analyzing generalization.
3. The analysis systematically compares two adversarial settings and derives different stability bounds, offering a structured perspective on their theoretical differences.

**Weakness**
1. While the paper derives stability and generalization bounds, the practical implications of these theoretical results remain unclear. It is not obvious how the derived bounds should influence the design of new robust aggregation algorithms. The paper briefly mentions cryptographic tools such as zero-knowledge proofs as a potential mitigation direction, but the connection between the theoretical stability bounds and such mitigation strategies is not clearly established.
2. A central claim of the paper is that Byzantine failures degrade generalization more severely than data poisoning. However, this conclusion appears largely intuitive. Byzantine failures assume a strictly stronger adversarial model, where attackers can send arbitrary and adaptive updates, while data poisoning only affects the training data. Given this stronger capability, it is not surprising that Byzantine failures would cause greater damage. Because of this, the novelty and necessity of deriving a precise theoretical bound are somewhat unclear. The paper does not sufficiently explain what new insight the derived bounds provide beyond confirming the intuitive ordering of the two threat models.
3. The paper is almost entirely theoretical and lacks empirical experiments evaluating the proposed findings. Although the paper includes a numerical illustration for a specific constructed example, it does not include experiments on real datasets or practical distributed learning setups.

---

> ### Author Rebuttal · Authors · 2026-03-31
>
> We thank the reviewer for their feedback. We address their concerns in detail below, starting with Q2, as it provides a natural foundation for responding to the remaining points.
>
> **Q2.** We agree that it is intuitive for Byzantine failures to be more harmful than data poisoning, since the former subsumes the latter. Yet, prior work has shown that this intuition is not valid when it comes to the **optimization error**. Indeed, the worst-case attack matching Byzantine upper bounds is derived under data poisoning (Farhadkhani *et al.*, 2024b).
>
> As discussed in the the second paragraph of the Introduction, until now, the community has lacked a theoretical demonstration of this hierarchy, relying solely on numerical analysis. Our work fills this gap, showing that these two threat models have fundamentally different capabilities when it comes to harming the **generalization error**. Specifically, we derive the first Byzantine attack that achieves matching upper and lower bounds with a magnitude strictly greater than those under data poisoning. This distinction is critical: without it, there is no formal justification for designing defenses against the stronger Byzantine workers.
>
> For **insights on why differences in threat models impact generalization but not optimization**, see the final paragraph of Section 4.
>
> **Q1.** Practical implications.
>
> **Cryptographic tools as a mitigation.** As explained in our conclusion, zero-knowledge proofs (ZKPs) effectively reduce Byzantine attacks to data poisoning. Since our theoretical analysis shows that algorithmic stability improves under data poisoning, this formally implies better generalization when ZKPs are deployed against Byzantine failures. To our knowledge, our work provides the first theoretical justification for using ZKPs in Byzantine-robust learning.
>
> **Design of new robust aggregations.** By proving that algorithmic stability fundamentally differs between the two threat models, we provide the first theoretical justification for tailoring defenses to specific threats:
> - *(i) If the system is susceptible to Byzantine failures*: Our proposed Byzantine attack causes the ouptut of SMEA to exceed the Lipschitz constant of the loss function, a scenario not possible under data poisoning. This highlights the relevance of strategies that cap the output norm, such as Adaptive Gradient Clipping [1], to improve stability under Byzantine attacks. Analyzing the stability impact of such algorithm is nontrivial and represents a promising direction for future work. Alternatively, ZKPs (discussed above) provide an off-the-shelf mitigation.
> - *(ii) If the system is only susceptible to data poisoning*: Future work should target robust aggregation rules that preserve input gradient regularity. For convex, smooth learning, preserving the co-coercivity inequality remains an open problem. While it may be impossible in general, identifying specific settings where it holds would directly improve guarantees. For smooth non-convex learning, preserving the smoothness inequality improves stability, which can be achieved using known rules like CWTM.
>
> **Q3.** We have focused on SMEA as it is optimal for the optimization of training errors (Allouah *et al.* 2023b). Nevertheless, we discuss the possibility of extending our results to other rules.
>
> - **Extending upper bounds**: Theorem 3.1 holds for any $(f,\kappa)$-robust aggregation rule. Under data poisoning, while the specific analytical technique depends on the aggregation rule, our proof strategy remains broadly applicable. As noted in lines 265–268, Theorem 3.3 holds for any rule that averages $n-f$ gradients, independent of the selection mechanism. As an other example, we have derived a similar upper bound using CWTM, but omitted it from the manuscript for brevity.
>
> - **Extending lower bounds:** The datasets, loss functions, and attacks that we have constructed for our lower bounds are tailored for SMEA. While extending it to other rules is non-trivial, we believe our proof techniques are directly applicable to the related Covariance bound-Agnostic Filter (Y. Allouah *et al.*, 2025), as both rules select vectors according to a variance minimization criterion. Designing novel attacks for different rules, such as CWTM, remains a challenging but promising direction for future work.
>
> Together, these results indicate that our theoretical framework is informative and generalizable beyond SMEA. Detailed extensions of the constructions to other aggregation rules are left for future work.
>
> **W3.** Regarding your weakness section, we would like to clarify that our theoretical investigation is directly motivated by prior empirical findings on realistic datasets, which consistently report a gap in test accuracy between Byzantine and data poisoning attacks. Please refer to our detailed response to Reviewer 5hm3 (Question 2) for more details.
>
> ---
> [1] Y. Allouah et al., Adaptive Gradient Clipping for Robust Federated Learning, ICLR 2025.

---

> > ### Author Rebuttal · Reviewer_JVHc · 2026-04-07
> >
> > Thank you author for the rebuttal, which has addressed most of my questions.

---

### Official Review · Reviewer_Yw9L · 2026-03-12

**Soundness:** 3
**Presentation:** 4
**Significance:** 3
**Originality:** 3
**Overall Recommendation:** 5
**Confidence:** 4

**Summary:**

The paper presents an analysis of the generalization term of the expected excess population risk for two failure modes: (1) Byzantine attacks and (2) poisoned data. Working under the assumption that the number of Byzantine clients $f$ constitutes at least one third and at most one half of the total number of devices $n$, the authors demonstrate that the generalization term in case (1) degrades more severely than in case (2). The authors relate the generalization term to the stability of SGD with the SMEA aggregation rule and show the existence of a data distribution and a Byzantine attack such that the resulting generalization strictly exceeds the worst-case data poisoning scenario. The authors also provide a detailed discussion of the obtained results and outline directions for future work. The theoretical findings are supported by empirical observations known from the literature as well as by an experiment conducted by the authors.

**Compliance With Llm Reviewing Policy:**

Affirmed.

**Final Justification:**

After rebuttal, I do not have any concerns regarding the paper. I believe the paper should be accepted.

**Key Questions For Authors:**

1) SMEA is the optimal aggregation rule in the case $f < \frac{n}{2}$. Nevertheless, there exist robust aggregation rules that provide theoretical guarantees for $f \geq \frac{n}{2}$. For example, methods that use a trust sample (see Algorithm 2 in [1]). Could the authors comment on whether schemes of this kind can be analyzed within their framework?

2) Could the authors explain why it was not possible to consider SGD in Theorem 3.2 and why it was not possible to construct an example in Theorem 4.2? Do the authors have any conjectures regarding what additional assumptions would be required in the problem formulation to obtain a lower bound on the generalization bound for robust SGD under Byzantine failures?

---

[1] Molodtsov, Gleb, et al. "Bant: Byzantine Antidote via Trial Function and Trust Scores." arXiv preprint arXiv:2505.07614 (2025).

**Limitations:**

Yes

**Strengths And Weaknesses:**

Strengths:

1) The analyzed phenomenon indeed occurs in practice. Thus, the work is of significant importance for theoretical research on robust distributed learning.

2) Authors derive $\Theta(\sqrt{\frac{f}{n-f}})$ for Byzantine failures and $\Theta(\frac{f}{n-f})$ for data poisoning attacks in the convex case in the range $\frac{n}{3} \leq f < \frac{n}{2}$ for SGD with SMEA aggregation. This part of the work is accompanied by proof sketches and a detailed discussion of the results.

3) The authors justify the assumption $f < \frac{n}{2}$ by the impossibility of the standard notion of $(f, \rho, \text{empirical})$-resilience.

4) The authors justify the connection between the generalization bound and the stability of the algorithm.

5) The strict separability of Byzantine and data failures is illustrated for deterministic GD with SMEA aggregation.

Weaknesses:

1) My major concern regarding this work is the assumption $f \geq \frac{n}{3}$. Although the authors explain why their theoretical framework does not allow them to derive a lower bound on the generalization bound for Byzantine failures, this is an important practical question that should be addressed in future work.

2) Theorems 3.1, 3.3, and 3.4 assume SGD as one of the possible options. However, Theorem 3.2 (the lower bound for Byzantine failures) only allows GD. The paper lacks an explanation of what prevents the construction of a lower bound in the stochastic case.

---

> ### Author Rebuttal · Authors · 2026-03-31
>
> We sincerely thank the reviewer for their positive feedback and address their specific concerns below. We provide detailed responses to the questions raised, as we believe this additional discussion will strengthen the paper.
>
> Q1. / W2. **Extension of Theorems 3.2 and 4.2 to SGD.**
>
> - *For Theorem 3.2*: We can prove the same lower bound for SGD in Theorem 3.2, provided that there exists $k>0$ such that $T \geq k m$. This mild assumption is necessary to ensure that the differing sample has non-negligeable probability to be drawn at least once. As the differing sample might not be drawn immediately, we have to adapt Algorithm 1 (i.e., the Byzantine workers' behavior) so that the attacker's trigger depends on all previously observed gradients. Concretely, at step $t$ (using the notation of Appendix C.3), $$\mathbf{if} \ \forall s \in [t], g^{(1)}_s < C \ \mathbf{then} \ \text{return an arbitrarily large value;} \ \mathbf{else} \ \text{return} \ \beta.$$ Under the condition $T \geq km$, the probability of sampling the differing sample is lower bounded by a constant ($1 - e^{-k}$). This allows us to recover the same divergence as in Theorem 3.2 over a constant fraction of the $T$ iterations, yielding the desired lower bound. We will include this extension in the appendix. Thank you for raising this point.
>
> - *For Theorem 4.2*: Using the lower bound construction above, we can directly apply Lemma 4.1 to extend Theorem 4.2 to the SGD case.
>
> W1. **Clarification on the reviewer's concern about the regime $f < \frac{n}{3}$.**
>
> We did derive a lower bound of $\Omega(\frac{f}{n-2f})$ for the Byzantine setting when $f < \frac{n}{3}$ (see Theorem 3.2 part (ii) in Appendix C.3). Although this differs from the $\Omega(\frac{f}{n-f})$ bound obtained under data poisoning, the two bounds are within a constant factor. For clarity and brevity, we omitted this result from the main text.
>
> We do not prove the tightness of this bound, but we conjecture that it is tight for GD with SMEA. We provide the following intuition, partially outlined in Appendix F.2 point (ii). The SMEA aggregation rule selects the subset of size $n-f$ with minimal variance. Therefore, Byzantine workers' capacity to skew the output is capped by this minimal variance. To achieve the largest possible lower bound, we have to maximize the variance of the honest subsets. The maximal variance for a set of $n-f$ values bounded by $C > 0$ is achieved when half the values are $C$ and the other half are $-C$ (yielding a variance of $C^2$). Yet, when $f < \frac{n}{3}$, the inequality $\frac{n-f}{2} + f < n-f$ dictates that any selected subset containing the $f$ Byzantine updates *must* inevitably include a mix of both positive and negative honest values. Because of this forced mixture, Byzantine workers must limit the magnitude of their attacks to avoid being filtered out by SMEA, in contrast to the case $f\geq\frac{n}{3}$, where no such forced mixture exists.
>
> Q2. **On the extension of our framework to the regime $f \geq \frac{n}{2}$.**
>
> This is an interesting point, although it is out of the scope of the present work. Indeed, prior methods that achieve guarantees for $f\geq \frac{n}{2}$ all rely on additional verified data at the server, which is not assumed in our setting. As noted in your strength paragraph, it is theoretically impossible to handle $f \geq n/2$ in our framework (cf. [1]).
>
> With that said, and based on a brief review of the referenced paper, we offer a few broader remarks:
> - **The "`AutoBant`" aggregation**: The operation on Line 11 in Algorithm 2 [2] can be interpreted as an aggregation step. However, because it explicitly depends on the current weight parameter (Line 10), it cannot be defined as a function $(\mathbb{R}^d)^n \to \mathbb{R}^d$. Therefore, it does not satisfy our definition of a valid aggregation rule (cf. Definition 2.2), and our theory cannot be directly applied, even in the case $f < n/2$.
> - Nevertheless, under Byzantine failures, the core proof technique of Theorem 3.1 (outlined in the Appendix C.2 proof sketch) relies on comparing the effective update to a standard SGD update. If one can establish a quantitative inequality analogous to the $(f, \kappa)-$robustness property for such a method, it may be possible to derive a stability upper bound in their setting.
>
> ---
>
> [1] Y. Allouah *et al.*, Robust Distributed Learning: Tight Error Bounds and Breakdown Point under Data Heterogeneity, NeurIPS 2023.
> [2] Molodtsov, Gleb, et al. "Bant: Byzantine Antidote via Trial Function and Trust Scores." arXiv preprint arXiv:2505.07614 (2025).

---

> > ### Author Rebuttal · Reviewer_Yw9L · 2026-04-03
> >
> > Thanks to the authors for the detailed clarification. I have no further questions and will maintain my positive score.

---

### Official Review · Reviewer_5hm3 · 2026-03-16

**Soundness:** 3
**Presentation:** 3
**Significance:** 3
**Originality:** 3
**Overall Recommendation:** 4
**Confidence:** 3

**Summary:**

The paper studies a significant gap in generalization guarantees between the two threat models of Byzantine failures and data poisoning. They show a separation in stability rates for data poisoning and Byzantine failures, that Byzantine failures is about an order of square root worse. And they show that this yields a generalization gap by constructing a setting where generalization is proportional to stability, and they back this up with numerical experiments.

**Compliance With Llm Reviewing Policy:**

Affirmed.

**Final Justification:**

I don't have further concerns, I will keep my positive score.

**Key Questions For Authors:**

The paper makes a non-obvious theoretical point. I like the paper in general and I have the following questions for the authors to answer:

- For the regime of $f < n / 3$, do you expect the Byzantine and poisoning gap to remain order-wise, or is it more likely to be only a constant factor gap?
- Do you have results on more realistic benchmarks showing that the predicted generalization gap appears beyond the proof-inspired synthetic setup in Figure 1?
- You highlight co-coercivity preservation as an open direction. Do you have candidate robust aggregation rules that might preserve it, or evidence suggesting this is impossible?

**Limitations:**

I don't see negative societal impacts.

**Strengths And Weaknesses:**

### Strengths:
- The paper cleanly separates optimization from generalization and gives tight or near-tight stability bounds, especially for data poisoning and for Byzantine failures when $f \ge n/3$.
- The main insight is important and non-obvious: two threat models that can look similar from an optimization perspective can behave quite differently for generalization.

### Weaknesses:
- The Byzantine lower bound is strongest only for $f \ge n / 3$. For smaller corruption levels, the order-wise gap is not fully resolved, so the tightness claim is not uniform across all regimes.
- The empirical section is fairly narrow. The experiments are mainly synthetic and closely tied to the lower-bound construction, rather than standard federated benchmarks.

---

> ### Author Rebuttal · Authors · 2026-03-31
>
> We thank the reviewer for their feedback and for acknowledging that the main insight is important and non-obvious.
>
> Q1. ***"For $f < \frac{n}{3}$, do you expect the Byzantine and poisoning gap to remain order-wise, or is it more likely to be only a constant factor gap?"***
>
> It is more likely to be only a constant factor gap when $f < \frac{n}{3}$. To support this conjecture, we provide the following intuition, partially outlined in Appendix F.2 point (ii) and Theorem 3.2 part (ii) in Appendix C.3. The SMEA aggregation rule selects the subset of size $n-f$ with minimal variance. Therefore, Byzantine workers' capacity to skew the output is capped by this minimal variance. To achieve the largest possible lower bound, we have to maximize the variance of the honest subsets. The maximal variance for a set of $n-f$ values bounded by $C > 0$ is achieved when half the values are $C$ and the other half are $-C$ (yielding a variance of $C^2$). Yet, when $f < \frac{n}{3}$, the inequality $\frac{n-f}{2} + f < n-f$ dictates that any selected subset containing the $f$ Byzantine updates *must* inevitably include a mix of both positive and negative honest values. Because of this forced mixture, Byzantine workers must limit the magnitude of their attacks to avoid being filtered out by SMEA, in contrast to the case $f\geq\frac{n}{3}$, where no such forced mixture exists.
>
> Q2. ***"Do you have results on more realistic benchmarks showing that the predicted generalization gap appears beyond the proof-inspired synthetic setup in Figure 1?"***
>
> Our theoretical investigation is directly motivated by prior empirical findings on realistic datasets, which consistently report a gap in test accuracy between Byzantine and data poisoning attacks. Below, we list examples of such results:
>
> * Figures 2, 4, and 6 from Allouah et al. (2023b), where ALIE, FOE, and SF simulate Byzantine failures, while LF or Mimic are data poisoning attacks. Table 2 from Allouah et al. (2023a) is also relevant.
> * Figure 2 from Allouah et al. (2025), where Label Flipping is a data poisoning attack while (Gradient) Sign Flipping simulates Byzantine failures.
> * Tables 2 and 5 from Fang et al. (2020), where LabelFlip, SingleWorker, and Uniform are data poisoning attacks and Partial and Full simulate Byzantine failures.
> * Figure 1 from Karimireddy et al. (2022), where LF is a data poisoning attack and IPM, ALIE, and BF simulate Byzantine failures.
>
> While the above list is currently provided in Appendix I, we are happy to include it in the introduction if the reviewer believes it would better emphasize the empirical motivation behind our theoretical study. Additionally, we can incorporate an illustrative numerical experiment in the appendix if desired.
>
> Before our work, it was unclear whether this empirically observed gap reflects a fundamental difference in adversarial power or merely the suboptimality of practical attacks. Prior theory failed to explain this observed gap (Farhadkhani *et al.*, 2024b). Although designing new benchmarks and stronger attacks is a worthwhile direction, relying solely on empirical validation leads to an endless cycle of scenario-specific attacks without establishing a universal distinction. This underscores the need for a formal theoretical framework. We address this gap by developing a refined theory that incorporates generalization error, providing a principled explanation for the observed phenomenon.
>
> Q3. ***"You highlight co-coercivity preservation as an open direction. Do you have candidate robust aggregation rules that might preserve it, or evidence suggesting this is impossible?"***
>
> We have a follow-up work that establishes that, in general, it is impossible to preserve this property under robust aggregation. In fact, we show that only affine aggregation rules preserve co-coercivity. The proof is technically involved, and the result has implications beyond robust distributed learning. We present this as a separate contribution in a dedicated paper, which is currently under review.
>
> That said, identifying specific settings in which co-coercivity can still be preserved remains an interesting research direction, as it would lead to improved guarantees in those cases. We refer to [1] for an example of such a setting.
>
> ---
>
> [1] S. Song *et al.*, Evading the curse of dimensionality in unconstrained private glms, AISTATS 2021

---

> > ### Author Rebuttal · Reviewer_5hm3 · 2026-04-02
> >
> > Thanks for the rebuttal. I don't have further concerns. I'll keep my score positive. I will be beneficial to include the new discussions in the revision.

---

### Decision · Program_Chairs · 2026-04-30

**Decision:**

Accept (regular)

**Comment:**

This paper studies the generalization error of (stochastic) gradient descent with SMEA in a distributed learning setting. Specifically, the authors consider two types of adversaries: Byzantine failures, where misbehaving workers may send arbitrary gradients, and data poisoning, where a subset of workers holds corrupted data. The authors show that these two settings exhibit fundamentally different generalization behavior by deriving uniform stability–based upper bounds of $\sqrt{f/(n - f)}$ for the Byzantine failure case and $f/(n - f)$ for the data poisoning case. They further establish matching lower bounds for both the uniform stability and the resulting generalization gap.

While three reviewers recommended acceptance, one reviewer raised concerns about the motivation and scope of the work. In particular, since the Byzantine failure model is strictly stronger than the data poisoning model, a worse generalization bound might be expected in that setting. However, as the authors point out, such a distinction does not appear in the optimization error. The other reviewers also appreciated the paper’s analysis, which highlights a nontrivial gap in generalization between the two settings. I find this perspective convincing and recommend acceptance.

If accepted, the authors should clarify this point and incorporate a more explicit discussion, as outlined in their rebuttal.